# Nanoparticles targeting mutant p53 overcome chemoresistance and tumor recurrence in non-small cell lung cancer

Yu-Yang Bi[1], Qiu Chen[1], Ming-Yuan Yang[1], Lei Xing[1,2] & Hu-Lin Jiang [1,2,3] ✉

Non-small cell lung cancer (NSCLC) shows high drug resistance and leads to low survival due to the high level of mutated Tumor Protein p53 (*TP53*). Cisplatin is a first-line treatment option for NSCLC, and the p53 mutation is a major factor in chemoresistance. We demonstrate that cisplatin chemotherapy increases the risk of *TP53* mutations, further contributing to cisplatin resistance. Encouragingly, we find that the combination of cisplatin and fluvastatin can alleviate this problem. Therefore, we synthesize Fluplatin, a prodrug consisting of cisplatin and fluvastatin. Then, Fluplatin self-assembles and is further encapsulated with poly-(ethylene glycol)–phosphoethanolamine (PEG–PE), we obtain Fluplatin@PEG–PE nanoparticles (FP NPs). FP NPs can degrade mutant p53 (mutp53) and efficiently trigger endoplasmic reticulum stress (ERS). In this study, we show that FP NPs relieve the inhibition of cisplatin chemotherapy caused by mutp53, exhibiting highly effective tumor suppression and improving the poor NSCLC prognosis.

Patients with NSCLC are often diagnosed at an advanced stage, leading to high mortality and a poor prognosis[1,2]. Patients with stage I-II disease can undergo surgery combined with chemotherapy, while patients with disease in stage III or higher can only be treated with drugs[2–4]. As the first-line treatment option for patients with NSCLC, cisplatin has been widely used in the clinic[5–7]. Cisplatin remains an irreplaceable treatment for NSCLC due to its broad spectrum of action and acceptable financial burden[4]. However, in the clinical use of cisplatin, clinical treatment often failures due to chemoresistance[8–10].

Because p53 is an essential factor through which cisplatin exerts its antitumor effects, the successful activation of p53 and its downstream pathways is crucial for cisplatin to exert its antitumor effects[11]. Notably, the p53 mutation rate in NSCLC is 46% for adenocarcinoma and 90% for squamous carcinoma[3]. In NSCLC, missense mutations are the most common and often lead to inactivation of p53[12]. P53 mutations can directly lead to the inactivation of cisplatin targets, which is an important factor leading to cisplatin resistance[12,13]. In addition, mutp53 shows a "gain of function (GOF)" effect that can further lead to cisplatin resistance[14–16]. Moreover, cisplatin as a chemotherapeutic

agent also carries the risk of causing mutations[17,18]. This suggests that chemotherapy with cisplatin may lead to *TP53* mutations in patients. If p53 mutations accumulate during cisplatin chemotherapy, this will also further affect the survival of patients after chemotherapy[14,15]. Despite all the basic research performed to date, it has not been conducted on strategies used to target mutp53 to reverse the cisplatin treatment bottleneck caused by cisplatin-induced mutations.

Strategies have been adopted to develop small-molecule compounds that can target mutp53 for effective anticancer therapy[19,20]. Statins are considered to have some potential for antitumor clinical applications as drugs that directly degrade mutp53[19,21,22]; however, the use of statins is currently limited by the clinical treatment approval process[22], and increasing evidence has demonstrated that modulation of mutp53 is only the first step to effective treatment, with a combination of chemotherapeutic agents being the next essential treatment[19]. In addition, statins combined with chemotherapeutic agents still have drawbacks because even wild-type p53 (wtp53) expression continues to be stabilized after cisplatin treatment, eventually leading to a protumor effect[23]. Therefore, it is important to

[1]State Key Laboratory of Natural Medicines, China Pharmaceutical University, Nanjing 210009, China. [2]Jiangsu Key Laboratory of Druggability of Biopharmaceuticals, China Pharmaceutical University, Nanjing 210009, China. [3]College of Pharmacy, Yanbian University, No.977, Gongyan Road, Yanji 133000, China. ✉e-mail: jianghulin3@gmail.com

develop an antitumor regimen that is independent of the p53 pathway. The proapoptotic pathway of ERS can act independently of the p53 pathway[19]. Importantly, this approach is flawed because mutp53 inhibits ERS-induced apoptosis[19]; however, when mutp53-specific degradation is used in combination with ERS, efficient tumor treatment independent of p53 may be achieved.

In this work, we validate some possible mechanisms underlying the progressive chemoresistance to cisplatin during NSCLC treatment. The antitumor effects of cisplatin are accompanied by an increased risk of *TP53* mutations, further contributing to cisplatin resistance. Fluvastatin is shown to mitigate the bottleneck of cisplatin therapy (Fig. 1). Next, we design a prodrug on the basis of the coordinated interaction between dechlorinated cisplatin and fluvastatin. Fluplatin can self-assemble into stable homogeneous nanoparticles. Next, we coat PEG−PE on its surface, and the final formulation FP NPs are obtained (Supplementary Fig. 1a). Based on the cell membrane insertion ability of PEG−PE, nanoparticles can be more accessible into cells[24], these FP NPs eventually reach the ER to degrade mutp53 and cause ERS[25,26]. Moreover, the FP NPs are activated in the nucleus where they trigger apoptosis through mismatched base pairs during the DNA damage repair response[27] (Supplementary Fig. 1b). FP NPs break the restriction of p53 via such a proapoptotic mechanism and exert efficient antitumor effects (Supplementary Fig. 1c). In addition, FP NPs ultimately slow the progression of NSCLC, illustrating their potential as a therapeutic approach to improving poor prognosis.

## Results

### The vicious interaction of cisplatin and p53

In NSCLC, mutations in p53 are significantly associated with patient survival (Fig. 1a, Supplementary Fig. 2, 3 and Supplementary Table 1). Both reactive oxygen species (ROS) production and DNA damage are factors in gene mutations[28,29]. Therefore, we performed confocal laser scanning microscopy (CLSM) to examine the ROS levels and DNA damage in wtp53-expressing A549 NSCLC cells treated with cisplatin (Supplementary Figs. 4a, b, 5). The ROS levels and DNA damage were positively correlated with the cisplatin dose. Next, we set up three dose groups of cisplatin to induce wtp53-expressing cells for 1 month (Fig. 1b). Dose selection was based on $IC_{50}$ and $IC_{30}$ values of cisplatin (Supplementary Fig. 6). First, we used Sanger sequencing to detect the mutation status of *TP53*, and the results showed that the number of mutations was dose-dependently (Fig. 1c and Supplementary Table 2). In addition, we examined the p53 expression levels after treatment (Supplementary Fig. 7). We also performed RNA sequencing (RNA-seq) of A549 cells after treatment to observe the changes in p53 target genes. The downregulation of p53 target genes increased with increasing dose (Fig. 1c). Next, we examined $IC_{50}$ and caspase activity after treatment (Supplementary Fig. 8). Whereas, ROS content, DNA damage, and total cholesterol (TC) content showed dose-dependent and time-dependent increases during the treatment (Fig. 1d−f).

To emphasize the necessity of fluvastatin addition, we performed RNA-seq on fluvastatin treated cells (Fig. 1g). The results showed that after treatment with cisplatin, the expression of cancer suppressor genes in the downstream pathway of p53 was downregulated and oncogenes were upregulated, while the opposite result was observed after fluvastatin treatment. (Fig. 1h). We analyzed the p53 downstream pathway using Gene Ontology (GO) enrichment analysis, and the results showed that the difference in the p53 downstream pathway between fluvastatin treated and wild-type cells was much smaller (Fig. 1i). We also analyzed the cholesterol pathway by Gene Set Enrichment Analysis (GESA), and the results showed that fluvastatin treatment downregulated the mevalonate pathway in mutant cells (Fig. 1j, k and Supplementary Fig. 10). We next determined the $IC_{50}$ values of cisplatin on H1975 cells, A549 cells, and A549/DDP cells via Methyl thiazolyl tetrazolium (MTT) cytotoxicity assay (Supplementary Fig. 4f). The $IC_{50}$ values of the mutant cells and drug-resistant cells

were much higher than that of the wtp53-expressing cells. We subsequently incubated A549 cells with cholesterol and next determined the cisplatin $IC_{50}$ values (Supplementary Fig. 4g). The results showed that high-cholesterol treatment resulted in increased cisplatin $IC_{50}$ values, and the addition of fluvastatin to these cells mitigated this trend[30,31]. Finally, we performed CLSM to examine the levels of p53 in A549 cells treated with cisplatin and H1975 cells treated with fluvastatin (Supplementary Fig. 9). These findings suggested that subsequent cisplatin resistance caused by cisplatin-induced p53 mutations was mitigated by fluvastatin treatment.

Cisplatin exerts its antitumor effect mainly through DNA damage to further activate the p53 pathway and c-Abl pathway[11]. First, we examined the proapoptotic effects of cisplatin on mutp53- and wtp53-expressing cells after inhibition of the p53 signaling pathway or c-Abl signaling pathway, respectively. STI571 was used as the c-Abl inhibitor, and pifithrin-α was used as the p53 inhibitor[32,33]. We constructed H1299 cells transfected with vector, expression constructs containing wtp53 or the R273H variant (Supplementary Fig. 11). In H1299 cells transfected with wtp53, the apoptogenic effects of cisplatin were identified in both pathways, and the effect on one pathway was not mitigated by the inhibited activation of the other pathway (Supplementary Fig. 12a). In H1299 cells transfected with vector and the R273H variant, the p53 pathway was inactivated, and inhibition of the c-Abl pathway was stronger in cells expressing mutp53 (Supplementary Fig. 12b, c). The same conclusion was obtained in H1975 cells and A549 cells (Supplementary Figs. 4, 13). This could also reflect the effect of p53 mutations on cisplatin resistance (Fig. 1l).

### Synthesis of Fluplatin and preparation of FP NPs

First, we wanted to synthesize a prodrug consisting of cisplatin and fluvastatin via a coordination reaction. To confirm the agent combination ratio of the prodrug, we investigated the combination indexes of cisplatin and fluvastatin (Fig. 2a and Supplementary Fig. 14). The results showed that fluvastatin and cisplatin showed synergistic effects, and the combination index was similar when the fluvastatin:cisplatin ratio was 2:1 and 4:1[34]. To maximize the synthesis of Fluplatin, a fluvastatin:cisplatin ratio of 2:1 was used. As shown in Supplementary Fig. 15, the final product was obtained in only two steps with a final yield of 64.18%. Then, Fluplatin was characterized by ultraviolet–visible spectroscopy, high-performance liquid chromatography, mass spectrometry (MS), and $^1H$, $^{13}C$, and $^{195}Pt$ nuclear magnetic resonance spectroscopy (NMR), and the final purity was 94.26% (Supplementary Fig. 16−22).

Fluplatin could self-assemble into nanoparticles with a particle size of $11.05 \pm 0.13$ (Fig. 2b), which are called Fluplatin NPs (F NPs). Next, as shown in Fig. 2c, a Fluplatin:PEG−PE mass ratio of 5:1 was used for self-assembling the final products, that is, the FP NPs. The encapsulation efficiency of FP NPs was $96.15 \pm 1.98\%$. The details of the process used to prepare FP NPs are shown in Supplementary Fig. 23. According to the dynamic light scattering (DLS) data, the average particle size of the FP NPs was $101.55 \pm 0.65$ nm, and the zeta potential was $-4.54 \pm 0.18$ (Fig. 2d and Supplementary Fig. 24). We examined the stability of F NPs and FP NPs in 10% plasma over 72 h, the PEG−PE coated FP NPs showed a slower release capacity (Fig. 2e). To further examine the response of coordination bond in nanoparticles to pH, we examined the release ability of nanoparticles in PBS solution containing 1% Tween 80 at different pH values. Release increased with decreasing pH, as shown in Fig. 2f. Therefore, we next investigated the self-assembly force of the F NPs and FP NPs (Fig. 2g). The results showed that the F NPs mainly depended on the hydrophobic force and the effects of coordination bonds. Whereas the self-assembly force of FP NPs is mainly a hydrophobic force, the force of coordination bonds is reduced. These results showed the successful encapsulation of F NPs by PEG−PE. The element distribution in the FP NPs is shown in Fig. 2h and Supplementary Fig. 25; notably, elemental F, Pt, C, N, O, and Cl in

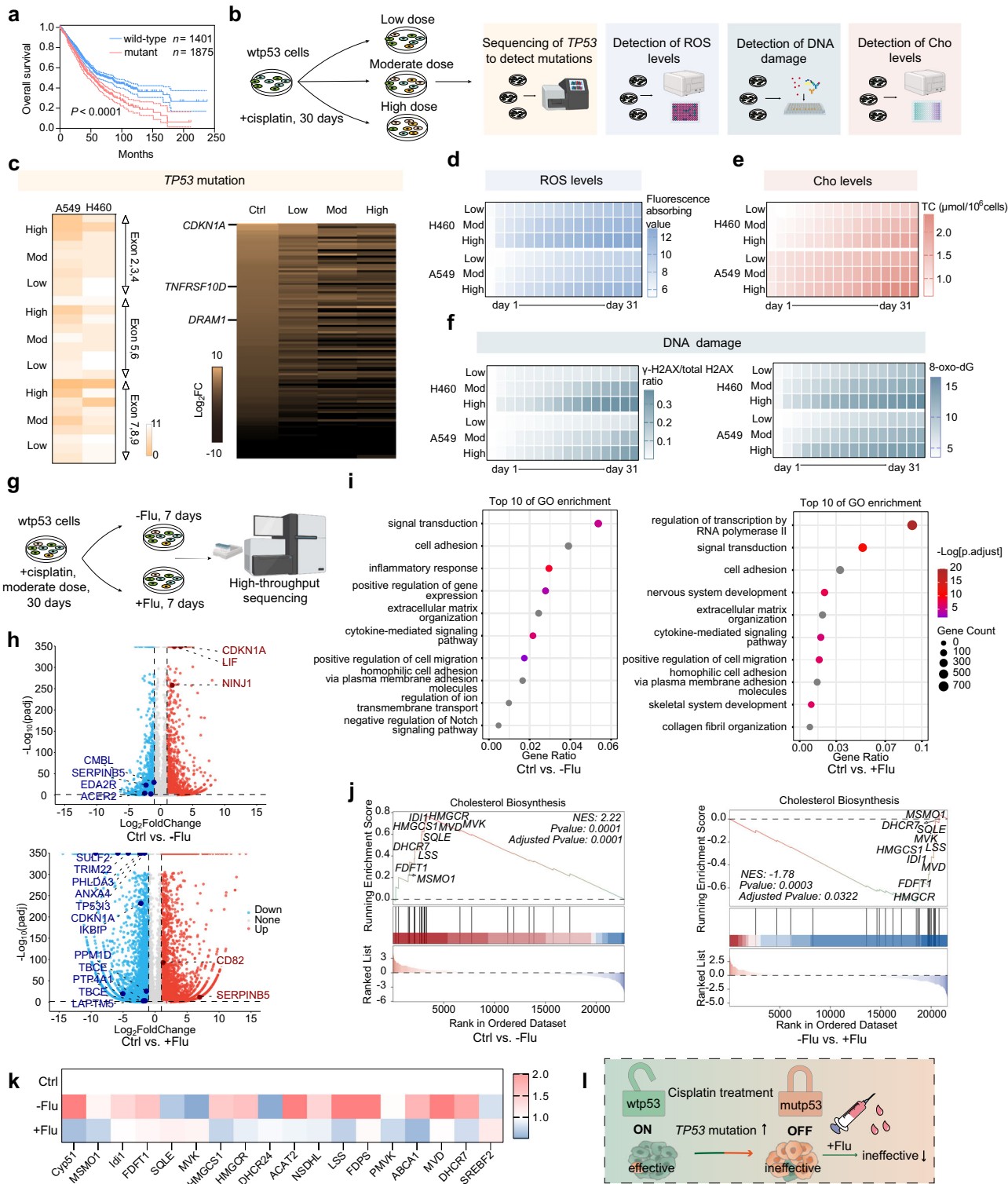

**Fig. 1 | Mechanistic validation of a vicious accumulation between cisplatin and p53. a** Kaplan-Meier plot of the correlation between the mutation of p53 and the survival of patients with NSCLC (Log-rank Mantel–Cox test). Cisplatin (Cis), fluvastatin (Flu). **b** Outline of the assays of cisplatin treatment (low dose, 10 μM; moderate (mod) dose, 25 μM; high dose, 50 μM). Cholesterol (Cho). Heatmap of the number of mutations and RNA-seq analysis (**c**), ROS levels (**d**), Cho levels (**e**), and DNA damage (**f**) for wtp53-expressing cells treated with cisplatin for 30 days. **g** High-throughput sequencing before and after fluvastatin sodium (4 μM) treatment in A549 cells. **h** Genome-wide analysis. The volcano plot depicts the significance and magnitude of difference (Fold Change). The dashed line indicates the threshold of the Fold Change > 2 and adjusted $p < 0.05$. Some of the cancer related genes are labeled by dark colors. **i** GO enrichment analysis of differentially expressed genes (DEGs). The advanced bubble chart shows GO enrichment of DEGs in signaling pathways. The x-axis label represents the gene ratio, and the y-axis label represents GO terms. **j** GSEA analysis. The normalized enrichment scores (NES) and $p$ values are indicated in each plot. **k** Heatmap analysis of mevalonate pathway genes from RNA-seq data. The color scale indicates the fold change in genes expression. **l** Schematic illustration of treatment with cisplatin. **b**, **g** created with BioRender.com. Data are shown as the mean ± SD; n.s. = no significance. Source data are provided as a Source Data file.

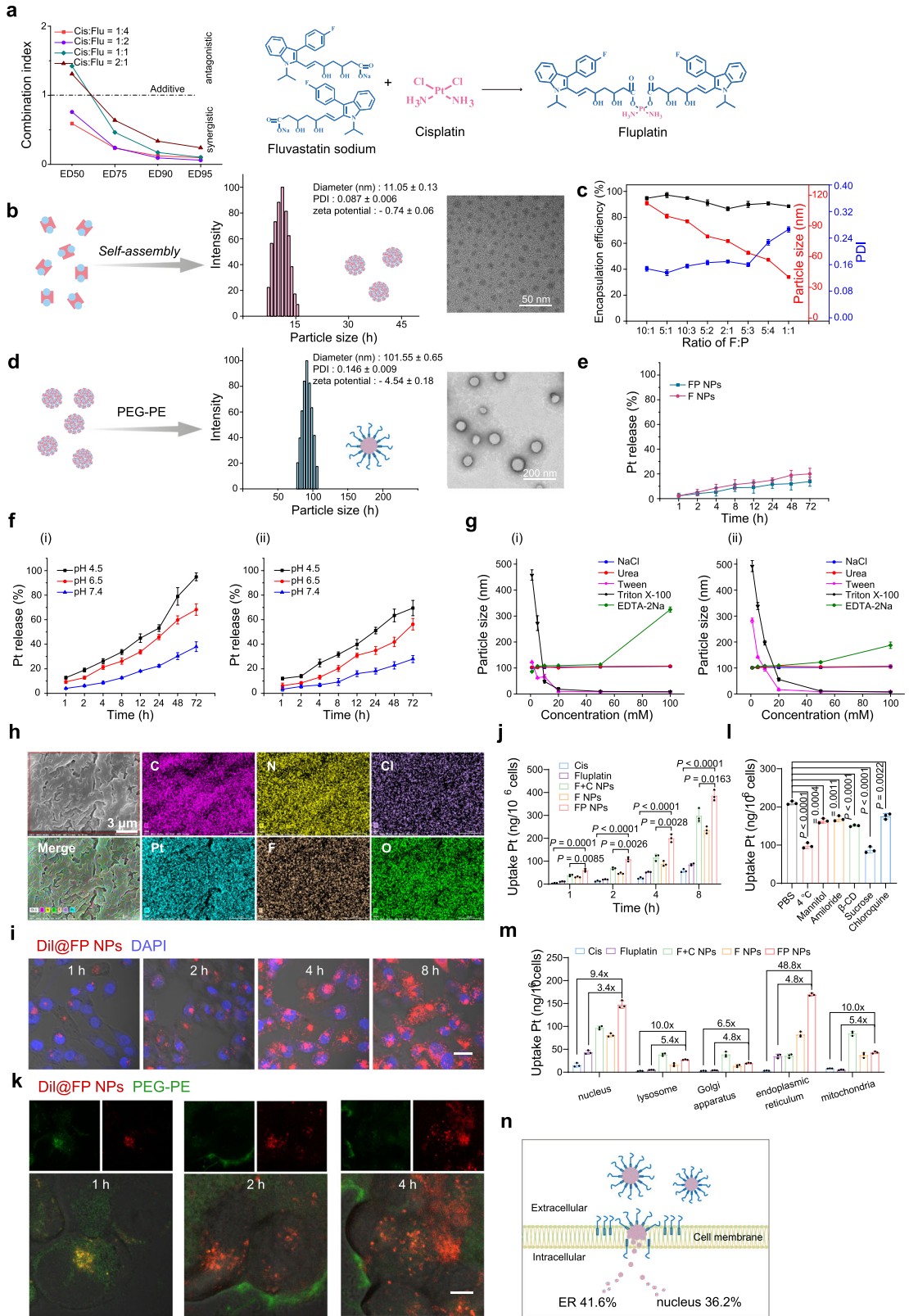

Fluplatin were uniformly distributed on the nanoparticles. Furthermore, to investigate the stability of the FP NPs, changes in particle size and polydispersity index (PDI) in four different solutions maintained for 7 days were investigated at room temperature and at 4 °C (Supplementary Fig. 26), the FP NPs had good stability. In addition, we performed hemolysis experiments with the FP NPs, FP NPs showed high biological safety (Supplementary Fig. 27).

We investigated the uptake of Dil@FP NPs (Dil encapsulated in F NPs). Dil@FP NPs by H1975 cells at different time points using CLSM (Fig. 2i and Supplementary Fig. 28), the results showed an increasing uptake with time. Next, to quantify the results, we investigated the uptake at the same time points using inductively coupled plasma–MS (ICP–MS) (Fig. 2j). Previous studies have demonstrated that micelles formed by PEG−PE can insert into the cell membrane without

**Fig. 2 | Design and characterization of FP NPs. a** CI value in different ratios of physical mixture, and the synthesis route of Fluplatin. **b** Particle size, zeta potential, PDI, and TEM image of the F NPs. Scale bars, 50 nm. **c** Particle size, PDI, and encapsulation efficiency of nanoparticles formed by different proportions of Fluplatin and PEG−PE ($n = 3$ independent samples). **d** Particle size, zeta potential, PDI, and TEM image of the FP NPs. Scale bars, 200 nm. **e** Release profile of the F NPs (i) and FP NPs (ii) in phosphate solution of 10% plasma at pH 7.4. **f** Release profile of the F NPs (i) and FP NPs (ii) in phosphate solution of 1% Tween 80 ($n = 3$ independent samples). **g** Mechanism for the F NPs (i) and FP NPs (ii) ($n = 3$ independent samples). **h** SEM image and EDS element mapping of FP NPs. Scale bars, 3 μm. ($n = 3$ independent samples) **i** Confocal images of H1975 cells after treatment with 2 μM Dil@FP NPs ($n = 3$ independent samples). Scale bars, 20 μm. **j** Uptake Pt in H1975 cells after treatment with 2 μM different formulations (dosage based on Fluplatin) detected by ICP−MS ($n = 3$ independent samples; one-way ANOVA followed by Tukey's HSD post hoc test). **k** Confocal images of Dil@FP NPs and PEG−PE in H1975 cells after treatment with 2 μM Dil@FP NPs. Scale bars, 5 μm. ($n = 3$ independent samples) **l** Uptake efficiency of 2 μM FP NPs treatment for 4 h in H1975 cells in the presence of various endocytosis inhibitors was detected by ICP−MS ($n = 3$ independent samples; one-way ANOVA followed by Tukey's HSD post hoc test). **m** Uptake of Pt in organelles after 8 h treatment with 2 μM different formulations (dosage based on Fluplatin; $n = 3$ independent samples; one-way ANOVA followed by Tukey's HSD post hoc test). **n** Major intracellular distribution of FP NPs. Data are shown as the mean ± SD. Source data are provided as a Source Data file.

disrupting membrane integrity, the fluidity of the cell membrane is then altered and the drug can enter the cell directly[24]. In addition, fluvastatin, a small-molecule inhibitor of 3-hydroxy-3-methyl-glutaryl-coenzyme A reductase (HMGR) located on ER endosomes has demonstrated the ability to specifically bind to ER endosomes[35], suggesting that FP NPs may have ER-targeting capabilities. To verify the mechanism of PEG−PE membrane insertion, we constructed FITC-Dil@FP NPs (PEG−PE was labeled by FITC, while Dil was assembled in F NPs) and observed their distribution in cells using CLSM. The results are shown in Fig. 2k that PEG−PE inserted into the cell membrane while Dil entered the cell, and a clear separation was observed at 2 h. In addition, to determine the interference of the pathway of PEG−PE endocytosis on the whole process, various endocytic pathway inhibitors were used to treat cells prior to FP NPs administration. As shown in Fig. 2l, the most important pathway was energy-dependent endocytosis. Finally, the platinum content in each organelle was quantified by ICP−MS (Fig. 2m). FP NPs were mainly distributed on the ER, and the Pt content was significantly higher than that of cisplatin or Fluplatin in each organelle (Fig. 2n).

## Tumor cell inhibitory effects of FP NPs
To verify the antitumor effect of the FP NPs, MTT cytotoxicity assays were performed with H1975, A549, and A549/DDP cells. For these three cell lines, the $IC_{50}$ values of the FP NPs were $2.24 \pm 0.20$ μM, $4.57 \pm 0.18$ μM, and $4.51 \pm 0.21$ μM, respectively, and the $IC_{50}$ values of cisplatin alone were $59.65 \pm 2.51$ μM, $15.54 \pm 0.92$ μM, and $123.53 \pm 5.22$ μM, respectively (Fig. 3a). These results indicated that the FP NPs exerted a highly potent effect against these three cell lines with a much lower $IC_{50}$ value than that of cisplatin. Both F NPs and FP NPs had much stronger antitumor effects than nanoparticles of free drugs encapsulated by PEG−PE (F + C NPs). Here, we used a fold increase (FI) to exemplify the specificity of FP NPs relative to cisplatin for mutp53. The FI values of H1975 and A549/DDP cells, two cells with p53 mutations, were much larger than those of wild-type A549 cells (p53 mutations in A549/DDP cells were verified in Supplementary Fig. 29). These three cell lines were then double-stained with calcein acetoxymethyl ester (calcein AM)/propidium iodide (PI), and the results showed that the best tumor killing effect was achieved after FP NPs treatment (Fig. 3b). The antitumor effect of the FP NPs was not inhibited by p53 mutations and was maximized in all cell types. Next, the apoptosis-inducing and overall H1975 cells killing abilities of the FP NPs were confirmed via apoptosis assays (Fig. 3c and Supplementary Fig. 30). In addition, we also examined the enzyme activities of caspase-3 and caspase-9 after treatment in each group in each of the three cell lines, and ultimately the FP NPs showed the greatest enzymatic activity (Fig. 3d). Next, we used transfected H1299 cells to examine the percentage of p53 pathway and c-Abl pathway in the antitumor activity of FP NPs. The results showed that in cells expressing wtp53, inhibition of p53 and c-Abl slightly reduced the antitumor activity of FP NPs (Fig. 3e); in p53-null and mup53-expressing cells, inhibition of p53 had no significant effect (Fig. 3f, g). The addition of these two inhibitors did not significantly reduce the antitumor activity

of FP NPs, suggesting that there were different mechanisms in the antitumor mechanism of FP NPs that were not in the p53 pathway and c-Abl pathway.

## Antitumor mechanism study of FP NPs in vitro
Because of the ER-targeting ability and metal complex composition of the FP NPs, we investigated their ERS-inducing capabilities and related mechanisms. First, we performed a colocalization analysis with ER-Tracker Green using CLSM (Fig. 4b and Supplementary Fig. 31), and the results showed that the Dil@FP NPs continued to accumulate on the ER over time. To detect ROS production in the ER, we observed colocalized ROS and the ER for a period of time and found that the ROS levels in the ER increased with time (Supplementary Fig. 32). As shown in Fig. 4c, we found that the calcium sensor Fluo-4 AM emitted intense green fluorescence in cells treated with Dil@FP NPs, which was enhanced with increased uptake of Dil@FP NPs. These findings indicated a marked increase in $Ca^{2+}$ levels in the cytoplasm, further demonstrating increased mitochondrial damage. Next, we used Mito-Tracker Green to identify changes in mitochondrial morphology and the degree of Dil@FP NPs and mitochondrion colocalization (Fig. 4d and Supplementary Fig. 33). With time, the mitochondrial morphology gradually changed from organelles presenting an initial filamentous shape to a spherical shape. Consistent with our previously obtained ICP−MS results, no significant accumulation of Dil@FP NPs in the mitochondria was observed until 8 h after Dil@FP NPs treatment, indicating that the damage to the mitochondria was caused by ERS. Furthermore, we examined the membrane potential of mitochondria with JC-1, and among the preparations evaluated, the FP NPs induced the lowest membrane potential (Fig. 4e, j, and Supplementary Fig. 34). Finally, we verified the expression of phosphorylated-eIF2α, unphosphorylated eIF2α and its key downstream transcription factors, activating transcription factor 4 (ATF4) and C/EBP homologous protein (CHOP), in the ERS-activated proapoptotic signaling pathway[13], by immunoblotting (IB) in H1975 cells and A549 cells. The ratio of phosphorylated-eIF2α to eIF2α and the ATF4 and CHOP expression levels were highest in the FP NPs group after treatment with different formulations (Fig. 4f, g and Supplementary Figs. 35, 36). These expression levels increased in a dose-dependent manner after FP NPs treatment (Fig. 4h and Supplementary Fig. 37, 38). The results of RNA-seq were the same as those of IB, FP NPs maximally activated ERS. In addition, downstream genes of the non-ERS apoptotic pathway were also maximally activated (Supplementary Fig. 39). Because the proapoptotic effect of ERS is independent of the p53 pathway[19], the efficient activation of ERS by FP NPs is the key to the efficient antitumor effect of FP NPs in both mutp53/wtp53-expressing cells. Considering the previous results demonstrating that the Pt content in FP NPs was high in the nucleus, we performed γ-H2AX IF staining with H1975 cells after different treatments, and among these treatments, the FP NPs treatment showed the strongest DNA-damaging effect (Fig. 4i, k).

Next, we performed IB to examine the p53 protein levels in mutp53-expressing and wtp53-expressing cells treated with various agents or FP NPs. The FP NPs in mutp53-expressing cells led to a higher

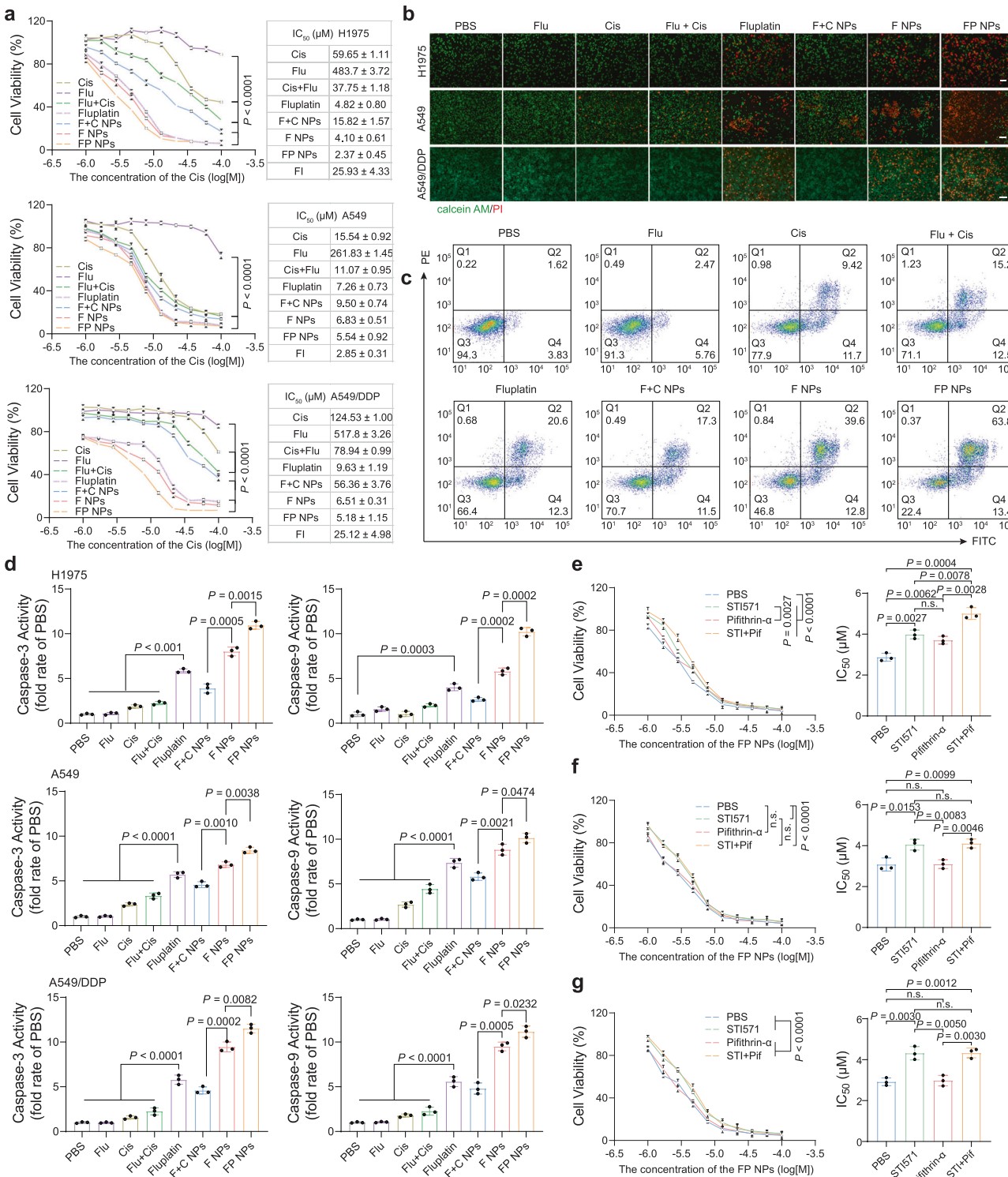

**Fig. 3 | Antitumor activity of FP NPs in vitro. a** Cytotoxicity and IC$_{50}$ of various formulations as determined by the MTT assay in H1975, A549, and A549/DDP cells after 24 h incubation ($n$ = 3 independent samples; one-way ANOVA followed by Tukey's HSD post hoc test). FI is defined as IC$_{50}$ (Cis)/IC$_{50}$ (FP NPs) **b** Inverted microscopy images of H1975, A549, and A549/DDP cells calcein AM/PI double staining after 6 h treatment with 4 μM different formulations (dosage based on Fluplatin; $n$ = 3 independent samples). Scale bars, 100 μm. **c** The Apoptosis rate of H1975 cells after incubation with 4 μM different formulations (dosage based on Fluplatin) for 6 h was determined by flow cytometry (FACS) analysis. **d** The fold rate of caspas3/9 enzymatic activities was detected after 6 h treatment with 4 μM different formulations (dosage based on Fluplatin; $n$ = 3 independent samples; one-way ANOVA followed by Tukey's HSD post hoc test). After Pifithrin-α and STI571 treatment for 24 h, the cytotoxicity and IC$_{50}$ of cisplatin treatment in H1299 cells transfected with expression constructs containing wtp53 (**e**), with vector (**f**), and expression constructs containing the R273H variant (**g**) were measured by the MTT method, $n$ = 3 independent samples; one-way ANOVA followed by Tukey's HSD post hoc test for (**e**, **f**, **g**). STI571 (STI), pifithrin-α (pif). Data are shown as the mean ± SD; n.s. no significance. Source data are provided as a Source Data file.

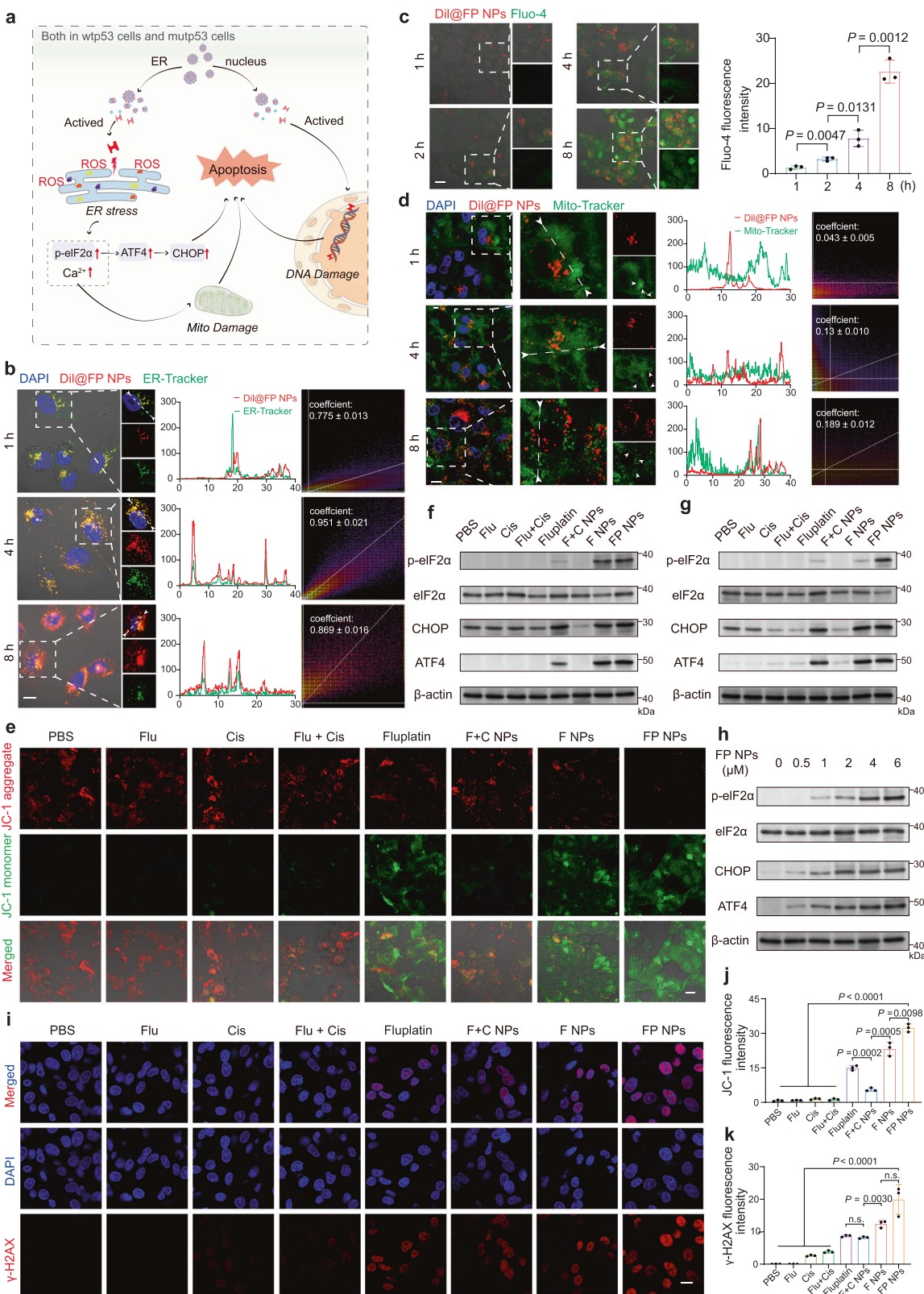

p53 degradation rate than fluvastatin (Fig. 5a–d), indicating that the FP NPs exhibited a stronger ability to degrade mutp53. In wtp53-expressing cells, FP NPs did not downregulate p53; in contrast, FP NPs upregulated it (Fig. 5a, e and Supplementary Fig. 40). In both H1975 and A549 cells, notably, the effect of FP NPs was dose-dependent (Fig. 5f–h). Additionally, we examined the ability of FP NPs to degrade mutp53 in other mutp53-expressing cells, and FP NPs

were most capable of degrading conformationally mutated p53 (Fig. 5i, j). Next, we verified by IB that the effect of FP NPs on mutp53 was mediated by ubiquitination rather than autophagy, whereas wtp53 was not affected by the inhibitor and its expression increases upon treatment with FP NPs (Fig. 5k–l). Next, we determined the half-life of mutp53 by cycloheximide (CHX) after treatment with FP NPs. The results showed that FP NPs significantly decreased the half-life of

**Fig. 4 | Mechanism of p53-independent antitumor activity of FP NPs in vitro.**
**a** Schematic summary of the p53-independent antitumor mechanism of FP NPs.
**b** Confocal images of Dil and ER-Tracker in H1975 cells treated with 2 μM Dil@FP NPs. Their colocalization determined by Pearson's correlation coefficient was quantified ($n = 3$ independent samples; one-way ANOVA followed by Tukey's HSD post hoc test). Scale bars, 10 μm. **c** Confocal images of Dil and Fluo-4 in H1975 cells treated with 2 μM Dil@FP NPs. Their fluorescence intensity was quantified ($n = 3$ independent samples; one-way ANOVA followed by Tukey's HSD post hoc test). Scale bars, 10 μm. **d** Confocal images of Dil and MitoTracker in H1975 cells treated with 2 μM Dil@FP NPs, and their colocalization determined by Pearson's correlation coefficient was quantified ($n = 3$ independent samples; one-way ANOVA followed by Tukey's HSD post hoc test). Scale bars, 10 μm. **e** Confocal images of JC-1 in H1975 cells treated with 4 μM different formulations (dosage based on Fluplatin)

for 6 h, and their fluorescence intensity was quantified (**j**), ($n = 3$ independent samples; one-way ANOVA followed by Tukey's HSD post hoc test). Scale bars, 10 μm. Western blotting analysis of p-elF2α, elF2α, CHOP, and ATF4 in H1975 cells (**f**) and A549 cells (**g**) after treatment with 4 μM of different formulations (dosage based on Fluplatin; $n = 3$ independent samples). **h** Western blotting analysis of p-elF2α, elF2α, CHOP, and ATF4 in H1975 cells after treatment with different concentration of FP NPs for 12 h ($n = 3$ independent samples). **i** Confocal images of IF staining against γ-H2AX in H1975 cells treated with 4 μM FP NPs for 6 h, and their fluorescence intensity was quantified (**k**), ($n = 3$ independent samples; one-way ANOVA followed by Tukey's HSD post hoc test). Scale bars, 10 μm. **a** Created with BioRender.com. Data are shown as the mean ± SD; n.s. no significance. Source data are provided as a Source Data file.

mutp53 within 16 h; whereas wtp53 itself had a short half-life and did not change significantly after FP NPs treatment (Fig. 5m–p). Finally, we verified again by immunoprecipitation that FP NPs are mediated by ubiquitination to degrade mutp53 and do not degrade wtp53 (Fig. 5q). In the next step of our analysis, we examined the HMGR activity in H1975 cells treated with fluvastatin, Fluplatin, F NPs or FP NPs. As shown in Fig. 5r, these treatments led to approximately the same levels of enzymatic activity, with FP NPs exerting a slightly lower effect than the other treatments, demonstrating that they exert the same inhibitory effect on HMGR as the other treatments. The results of the volcano plot showed that in H1975 cells, p53 downstream oncogenes were downregulated and oncogenes were upregulated compared to the control group (Fig. 5s and Supplementary Fig. 41). The specific effects of FP NPs on p53 are summarized in Fig. 5t. In conclusion, in mutp53-expressing cells, FP NPs were able to maximize the specific degradation of mutp53 by fluvastatin and further downregulate "GOF"; in wtp53-expressing cells, FP NPs were still able to exert the proapoptotic mechanism of cisplatin upregulation of wtp53 (Fig. 5t). The combination of the regulation of p53 and ERS by FP NP ultimately triggered apoptosis in both mutp53/wtp53-expressing tumor cells.

**Antitumor effects of the FP NPs in vivo**

Subsequently, we investigated the antitumor effects of FP NPs on mutant H1975 cells in vivo. First, we established an in vivo H1975 cell subcutaneous tumor model with nude mice. After treatment, the antitumor effect of cisplatin was not significant in p53-mutated models, and the combination of fluvastatin alleviated cisplatin resistance but was still limited. The FP NPs showed better tumor suppressive activity, as indicated by the smallest mean tumor volume of $107.81 \pm 10.69$ mm$^3$ (Fig. 6b). Figure 6c, d depicts the evolution of the tumor volume following the initial treatment. To further assess the apoptosis rate within the tumors of treated mice, we performed transferase-mediated dUTP nick end-labeling (TUNEL) fluorescence staining, ROS staining, hematoxylin and eosin (H&E) staining, and immunohistochemistry (IHC) to detect cleaved caspase-3 (Fig. 6f–i and Supplementary Fig. 42). The results showed that FP NPs upregulated the expression of cleaved caspase-3; caused the most accelerated cell death, as determined by H&E staining; led to a significant increase in the number of TUNEL-positive apoptotic cells, and generated a large amount of ROS in the tumor tissues. Given the above results, in p53-mutant NSCLC, FP NPs possessed the greatest antitumor efficacy. To exclude the effect of gender on the results of the experiment, we repeated the results of the change experiment in both female and male mice (Supplementary Fig. 43).

We examined the ability of the FP NPs to be passively targeted in vivo. The fluorescence intensity of the DiR@FP NPs reached the maximum value at 24 h, and their accumulation at tumor sites was greater than that at key organs (heart, liver, spleen, lung, and kidney) at any time point ex vivo, as shown in Fig. 7a. Next, to evaluate the precise effect, we quantified the Pt content of the tumors and different organs at different time points using ICP–MS. The results, which are displayed

in Fig. 7b, c, indicated that the accumulation of FP NPs was greatest at the tumor sites. The safety of the FP NPs was evaluated by measuring the changes in body weight of the mice, H&E staining of each organ after dissection, and serum biochemical indexes (Fig. 6e and Supplementary Fig. 44, 45). Next, to examine the ability of inhibition of the mevalonate pathway, the cholesterol levels of the tumor tissues and serum levels of TC, triglyceride (TG), low-density lipoprotein (LDL), and high-density lipoprotein (HDL) were measured. FP NPs had the greatest ability to reduce the cholesterol level in tumor tissues (Fig. 7d). ATF4 IF staining was performed with the tumor tissues of each group, and the FP NPs group showed the greatest ERS (Fig. 7e and Supplementary Fig. 46a). In addition, γ-H2AX IF staining demonstrated that, among the treatments, FP NPs damaged DNA to the greatest extent (Fig. 7f and Supplementary Fig. 46b). As shown in Fig. 7g, h, IB of the tumor tissue to measure p53 levels revealed that the FP NPs had the highest capacity to degrade mutp53 in vivo. We also verified this effect by performing p53 IHC with tumor tissues (Fig. 7i and Supplementary Fig. 46c).We next examined the survival and mean organ weight of each group, and the FP NPs showed the greatest positive effect on survival status (Fig. 7j, k). Ki67 expression is related to the degree of NSCLC cell differentiation and lymph node metastasis, and IHC assays were performed to measure the Ki67 level in each group (Fig. 7l and Supplementary Fig. 46d).

Similarly, we performed a pharmacodynamic examination of FP NPs in tumor-bearing mouse models of A549 and A549/DDP cells (Supplementary Fig. 47). To better reflect the process of tumor development and metastasis, we constructed H1975-luc, A549-luc, and A549/DDP-luc orthotopic lung tumors (Fig. 8a, h and Supplementary Fig. 48a). After successful orthotopic model construction, we performed in vivo imaging on days 0, 4, 8, 12, and 19 (Fig. 8b, h and Supplementary Fig. 48b). Besides, we recorded the final anatomical photographs and H&E staining intact lung lobes (Fig. 8c, i and Supplementary Fig. 48c), the number of tumor nodules in the lungs was also counted (Fig. 8d, j and Supplementary Fig. 48d). Finally, we quantified the bioluminescent image from each group (Fig. 8e, k and Supplementary Fig. 48e) and recorded the weight changes (Fig. 8f, l and Supplementary Fig. 48f). The orthotopic lung cancer model was consistent with the results of the subcutaneous tumor model, the antitumor effect of cisplatin was attenuated in p53-mutant and drug-resistant cells, whereas FP NPs were unaffected.

**The effect of FP NPs in preventing tumor recurrence**

Mutations in p53 are significantly associated with tumor development and poor prognosis in NSCLC[36–38]. Therefore, we wanted to examine the ability of FP NPs to inhibit tumor recurrence and metastasis. We examined recurrence in a subcutaneous tumor model of H1975 cells. The tumor tissue was completely excised after exposure to each dosing regimen (Fig. 9a). Changes in tumor volume and body weight of each mouse were recorded throughout the process (Fig. 9b–d). On the last day, minimal tumor recurrence was found in the FP NPs group (Fig. 9e). As determined by analyzing the photographs showing the

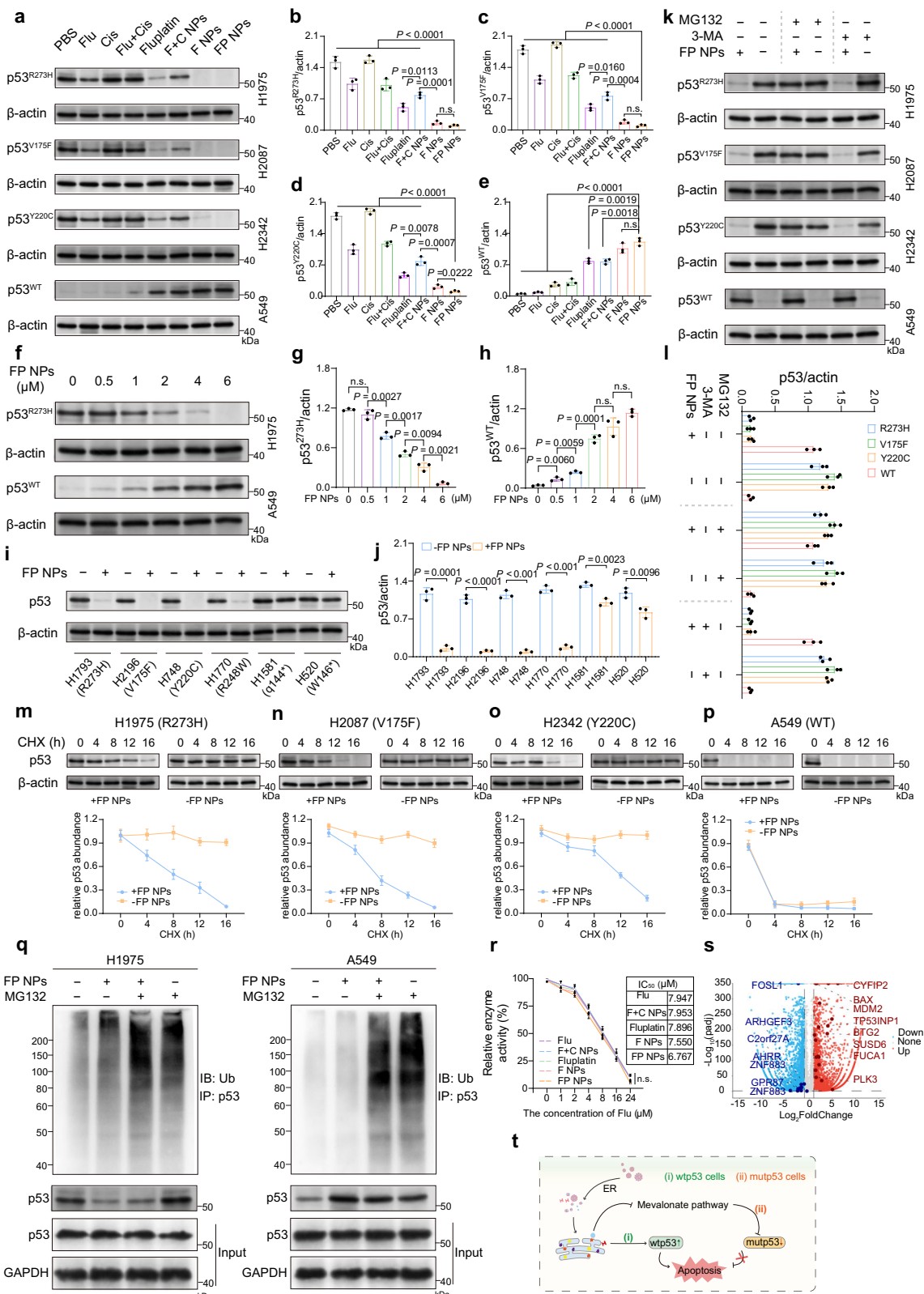

lungs on the last day and the H&E staining intact lung lobes, no visible lung metastases were identified in the FP NPs group (Fig. 9f). Furthermore, to better examine the expression of prognosis-related proteins at primary, recurrent, and metastatic sites after drug treatment, we separately measured the protein expression of p53, E-cadherin, matrix metallopeptidase 2 (MMP-2), matrix metallopeptidase 9 (MMP-9), and vimentin by IHC (Fig. 9g). In addition, the mice in the FP

NPs group had the lowest recrudescent tumor weight (Fig. 9h) and the highest long-term survival rate (Fig. 9i). The weights of the organs in each group were examined, and the spleen and lungs were found to be much heavier in the control group than in the FP NPs group (Fig. 9j). The greater spleen weight in the control group reflected the greater treatment effect of the FP NPs. Routine blood tests, which are common indicators of prognosis[39,40], were performed. The lymphocyte/

**Fig. 5 | Mechanism of p53-related antitumor activity of FP NPs in vitro.**
**a** Western blotting analysis of p53 in cells after treatment with 4 µM of different formulations (dosage based on Fluplatin) for 12 h. Their grayscale values were quantified (**b**–**e**) ($n = 3$ independent samples; one-way ANOVA followed by Tukey's HSD post hoc test). **f** Western blotting analysis of p53 in cells after treatment with different concentrations of FP NPs for 12 h. Their grayscale values were quantified (**g**, **h**) ($n = 3$ independent samples; one-way ANOVA followed by Tukey's HSD post hoc test). **i** Western blotting analysis of p53 in cells after treatment with 4 µM of FP NPs for 12 h. And their grayscale values were quantified (**j**) ($n = 3$ independent samples; one-way ANOVA followed by Tukey's HSD post hoc test). **k** Western blot of cells treated with 4 µM FP NPs or not for 12 h and cultured in medium containing 10 µM MG132 or 10 mM 3-MA. Their grayscale values were quantified (**l**) ($n = 3$

independent samples; one-way ANOVA followed by Tukey's HSD post hoc test). Western blot of H1975 (**m**), H2087 (**n**), H2342 (**o**), and A549 (**p**) cells treated with 2 µM FP NPs or not for 12 h and then treated with 100 µg/mL CHX at the indicated time points. $n = 3$ biologically independent samples, relative p53/actin ratios are shown. **q** Western blotting analysis of total ubiquitination (Ub) of immunoprecipitated (IP) p53 in cells after treatment with 2 µM FP NPs with or without MG132 ($n = 3$ independent samples). **r** HMGR inhibition assay under different formulations ($n = 3$ independent samples; one-way ANOVA followed by Tukey's HSD post hoc test). **s** The volcano plot shows differential expression of FP NPs treated and untreated H1975 cells. **t** Schematic summary of the p53-related antitumor mechanism of FP NPs. Data are shown as the mean ± SD; n.s. no significance. Source data are provided as a Source Data file.

monocyte ratio (LMR), neutrophil/lymphocyte ratio (NLR), and platelet/lymphocyte ratio (PLR) were included in our analysis. Among these measurements, an elevated NLR has been the most closely associated with poor prognosis in NSCLC patients[41], and all the factors in this study indicated greater effectiveness of the FP NPs compared to that of the other group (Fig. 9k). In addition, increased inflammation is a newly identified feature of tumors, and inflammatory factors are important for the prognostic evaluation of tumors[42]. Here, we examined tumor necrosis factor-α (TNF-α), interleukin 6 (IL-6), interleukin 10 (IL-10), and vascular endothelial growth factor (VEGF) protein expression, which was induced by mutp53 overexpression[43]. High levels of these inflammatory factors were significantly associated with malignant tumor progression, metastasis, and poor prognosis, and in our study, the levels of these inflammatory factors were significantly lower in the FP NPs group (Fig. 9l). Finally, we examined the recurrence models of A549 and A549/DDP cells (Supplementary Fig. 49a, l). The changes in tumor volume and the results of in vivo imaging showed that the FP NPs maximally fought the tumor and inhibited recurrence (Supplementary Fig. 49b, m). The H&E staining intact lung lobes, individual organ weights, routine blood tests, and Elisa results showed that the prognosis was worse in the p53-mutant or drug-resistant cell models, yet treatment with FP NPs reversed the deficiencies of cisplatin therapy (Supplementary Fig. 49c–k, n–v). These results suggest that FP NPs can improve postoperative recurrence in NSCLC patients and have potential for clinical application.

## Discussion

As the most common type of lung cancer, the vast majority of patients are in the middle or late stages when diagnosed; therefore, drug therapy remains the most important treatment[2–4]. Cisplatin is a first-line agent in the treatment of NSCLC, whose drug resistance is the primary consideration in its application[44]. In NSCLC, p53 has a high mutation rate, which directly leads to drug resistance[3,10]. In addition, chemotherapy is also thought to be a major cause of genetic mutations[9]. To clarify the changes of *TP53*, we chose wtp53-expressing A549 and H460 cells, which are widely used in preclinical studies[45]. Cisplatin increased the risk of *TP53* mutations with increasing dose and duration of chemotherapy (Fig. 1c–f). Fluvastatin, restored the antitumor ability of a portion of the p53 downstream pathway and downregulated the mevalonate pathway, which was upregulated due to the p53 mutations in mutated A549 cells (Fig. 1g–k).

FP NPs that we prepared can be used as a therapeutic agent for NSCLC (Fig. 2). FP NPs have a broad spectrum of action in vitro and can effectively overcome drug resistance due to p53 mutations (Fig. 3a–d). Experimental results have shown that FP NPs translocated to more ER than other cellular compartments (Fig. 2m, n). Interestingly, in mutp53-expressing cells FP NPs efficiently degraded mutp53 and triggered ERS (Figs. 4, 5), whereas, in wtp53-expressing cells, FP NPs still exerted cisplatin to activate wtp53 and efficiently triggered ERS (Fig. 5). Moreover, ERS-induced apoptosis was much higher than other antitumor mechanisms in both mtp53/wtp53-expressing cells (Fig. 3e–g). This process is also independent of whether p53 is

mutated, allowing the restriction of cisplatin to p53 to be broken. In the examination of the in vivo antitumor activity of FP NPs, we used two preclinical models, a subcutaneous tumor model and an orthotopic lung cancer model[46,47], to better compare the in vivo antitumor activity among mutant, wild-type, and drug-resistant cells (Figs. 6–8 and Supplementary Fig. 42, 43). In addition, mutp53 is thought to be closely associated with tumor recurrence and metastasis[43,48]. Moreover, suppression of the innate immune system by mutp53 may also exert an effect on tumor recurrence[5,49]. We established subcutaneous models of three cell types, H1975, A549, and A549/DDP, to examine the antitumor effect of FP NPs on survival and recurrent metastasis in mice. The results showed that FP NPs demonstrated the lowest recurrence and metastasis (Figs. 9 and Supplementary Fig. 48–53).

In conclusion, our results suggest that FP NPs may exert a profound therapeutic effect on p53-mutant NSCLC and prevent NSCLC tumor recurrence and resistance to cisplatin therapy (Figs. 7–9). Thus, our work may provide an effective treatment strategy for patients with p53-mutant NSCLC.

## Methods
### Ethics statement
All animal procedures were complied with the guidelines for care and use of the pharmic laboratory animal center of China Pharmaceutical University and experiments were approved by the animal ethics committee of China Pharmaceutical University (2022-08-009).

### Materials
Cisplatin was purchased from Shandong Boyuan Pharmaceutical Co., Ltd. Fluvastatin sodium was purchased from Shanghai Resuper Pharmaceutical Technology Co., Ltd. DSPE-mPEG2000 (N-(Carbonylmethoxypolyethyleneglycol2000)−1,2-distearoyl-sn-glycero-3-phosphoethanolamine, sodium salt) purchased from AVT (Shanghai) Pharmaceutical Tech Co., Ltd. Golgi Extraction Kit and Lysosomal Extraction Kit were purchased from Shanghai Bestbio Biotechnology Co., Ltd. Total Cholesterol Content Detection Kit was purchased from Shanghai Enzyme-linked Biotechnology Co., Ltd. TRIzol reagent, Mouse IL-6 ELISA KIT and Mouse IL-10 ELISA KIT were purchased from Bejing Solarbio Science & Technology Co., Ltd. Nuclei Isolation Kit, BCA Protein Quantitation Assay and Whole Cell Lysis Assay were purchased from Jiangsu Keygen Biotechnology Co., Ltd. ER Enrichment Kit and Mitochondrial Isolation Kit were purchased from Invent Biotechnologies, Inc. Pifithrin-α, NADPH, Enhanced Mitochondrial Membrane Potential Assay Kit with JC-1, Fluo-4 AM, Mito-Tracker Green, ER-Tracker Green, Caspase 3 Activity Assay Kit and Caspase 9 Activity Assay Kit were purchased from Beyotime Biotechnology Co., Ltd. Annexin V-FITC/PI Apoptosis Detection Kit were purchased from Yeasen Biotechnology (Shanghai) Co., Ltd. Reactive Oxygen Species (ROS) Fluorometric Assay Kit were purchased from Elabscience Biotechnology Co., Ltd. Plasmids (YOE-RP006-hTP53) were constructed by Ubigene Biosciences Co., Ltd., Guangzhou, China. Primary antibodies used for western blot experiments, IF and IHC staining: anti-p53 (Santa Cruz Biotechnology, sc-126; 1:1000 dilution), anti-VEGF (Santa

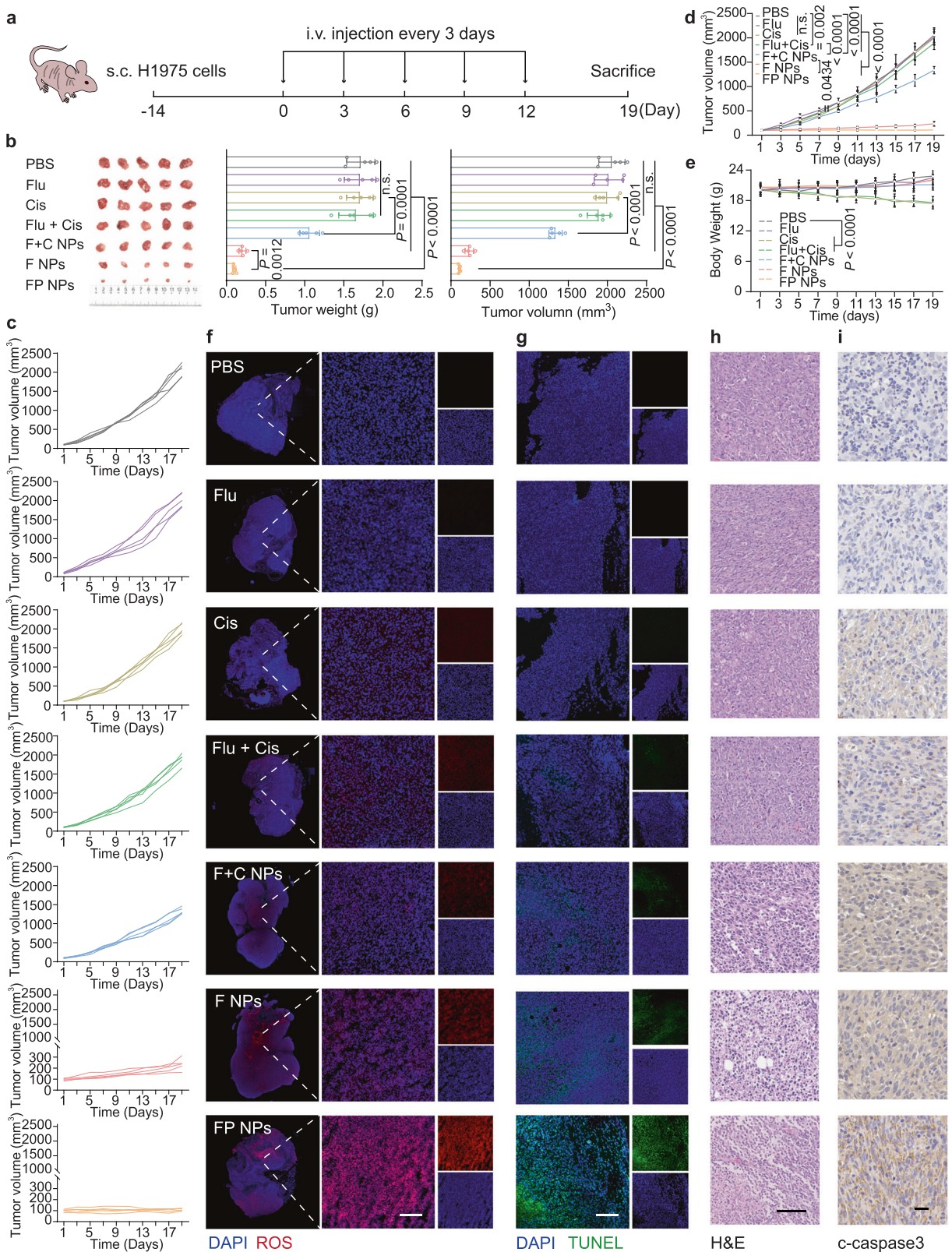

**DAPI ROS** **DAPI TUNEL** **H&E** **c-caspase3**

Cruz Biotechnology, sc-7269; 1:1000 dilution), anti-eIF2α (Cell Signaling Technology, #5324; 1:1000 dilution), anti-Phospho-eIF2α (Cell Signaling Technology, #3398; 1:1000 dilution), anti-Phospho-H2A.X (Servicebio, GB11181; 1:1000 dilution), anti-MMP2 (Servicebio, GB11130; 1:1000 dilution), anti-MMP9 (Servicebio, GB11132; 1:1000 dilution), anti-E-cadherin (Servicebio, GB12083; 1:1000 dilution), anti-Vimentin (Servicebio, GB111308; 1:1000 dilution), anti-ATF4 (Beyotime, AF2560; 1:1000 dilution), anti-DDIT3/CHOP (Beyotime, AF6684; 1:1000 dilution).

**Cell lines and animals**

The p53 mutant human NSCLC cell lines NCI-H1975 (American Type Culture Collection (ATCC), #CRL-5908), NCI-H2087 (ATCC, #CRL-5922), NCI-H2342 (ATCC, #CRL-5941), NCI-H1793 (ATCC, #CRL-5896),

**Fig. 6 | Antitumor efficacy of intravenously injected FP NPs in an H1975 tumor-bearing mouse model. a** Schematic illustration of the experimental design. **b** Images of tumor and tumor weight/volume on day 19 of each group ($n = 5$ mice per group; one-way ANOVA followed by Tukey's HSD post hoc test). **c** Tumor volume changes for each mouse in each group over 21 days of treatment ($n = 5$ mice per group). **d** The tumor volumes of mice during treatments with each group ($n = 5$ mice per group; two-way ANOVA followed by Tukey's multiple comparisons post test). **e** The body weight of mice during treatments with each group ($n = 5$ mice per group; two-way ANOVA followed by Tukey's multiple comparisons post test). Confocal images of ROS (**f**) and TUNEL (**g**) in the tumor tissues of mice after treatment with each group ($n = 3$ independent samples). Scale bars, 100 μm. IHC of cleaved caspase-3 (**h**) and H&E staining (**i**) in the tumor tissues after treatment with each group ($n = 3$ independent samples). Scale bars of cleaved caspase-3, 20 μm; Scale bars of H&E, 100 μm. Data are shown as the mean ± SD; n.s. no significance. Source data are provided as a Source Data file.

NCI-H2196 (ATCC, #CRL-5932), NCI-H748 (ATCC, #CRL-5841), NCI-H1770 (ATCC, #CRL-5893), NCI-H1591 (ATCC, #CRL-5878), and NCI-H520 (ATCC, #HTB-182) were purchased from ATCC. The p53 null human NSCLC cell line NCI-H1299 (ATCC, #CRL-5803) was purchased from ATCC. The p53 wild-type human NSCLC cell lines A549 (ATCC, #CCL-185) and NCI-H460 (ATCC, #HTB-177) were purchased from ATCC. The cisplatin-resistance human NSCLC cell line A549/DDP (Procell, #CL-0519) was purchased from Cell Resource Center (Procell Life Science & Technology Co., Ltd., China), obtained from cisplatin-induced A549 cells. The cell lines NCI-H1975-luc (Aoyinbio, SAc0273LG), A549-luc (Aoyinbio, SAc0023LG), and A549/DDP-luc (Aoyinbio, SAc0024LG) were purchased from Cell Resource Center (Aoyinbio Co., Ltd., China). The A549/DDP and A549/DDP-luc cells were cultured in RPMI-1640 media supplemented with 1% penicillin/streptomycin (Thermo-Fisher Scientific), 10% fetal bovine serum (FBS; Gibco) and 1–2 μg/mL cisplatin. The other cells were cultured in RPMI-1640 media supplemented with 1% penicillin/streptomycin (Thermo-Fisher Scientific) and 10% fetal bovine serum (FBS; Gibco). Authentication via STR profiling was performed by the commercial suppliers before purchase of the material. We also retested before the experiment was performed. All cell lines are tested in a regulatory basis (every 3 months) to rule out any mycoplasma contamination, all cells tested negative for mycoplasma contamination.

Female and male BALB/c nude mice, 4–5 weeks old, were purchased from Changzhou Cavens Laboratory Animal Co., Ltd. (Jiangsu, China). In order to avoid the biases associated with sex differences in the immune system and tumor growth, only females were selected in the main experiment. The mice were kept under observation for 1 week to confirm the survival environment (pathogen-free condition feed nude mice equipment, room temperature of 26–28 °C, humidity should be maintained at 40–60%, a light and dark cycle of 10 h of light and 14 h of no light should be maintained daily, food and water must be sterilized and freely available) and the health status of the nude mice.

### Synthesis of Fluplatin
Cisplatin (100 mg, 0.33 mmol) and silver nitrate (113.22 mg, 0.66 mmol) were dissolved in 5 mL DMF and stirred at room temperature in the dark for 24 h. The reaction solution was centrifuged until the silver chloride precipitate was removed and the supernatant was collected. Subsequently, fluvastatin sodium (571.56 mg, 1.32 mmol) was weighed and dissolved in the supernatant obtained in the previous step. The mixed solution was stirred at room temperature for 24 h in the dark, and then the solution was centrifuged and the yellow-brown supernatant was collected[50]. The product was then dissolved using ethyl acetate and extracted, concentrated and dropped into petroleum ether. The mixture was centrifuged, the precipitate was collected and dried in a fume hood to remove the residual petroleum ether, and the final product was obtained as a brown powder. The final yield is calculated after purification. Fluplatin was characterized by MS and NMR (Supplementary Fig. 16–18). ESI-MS: calculated for [M − H]⁻: 1048.37, found: 1048.6; ESI-HRMS: calculated for [M + Na]⁺: 1072.3598, found: 1072.36097; ¹H NMR (300 MHz, DMSO-$d_6$): δ ppm, 16-H: 7.64 (dd, $J = 8.2$, 2.8 Hz, 2H), 13,17,20-H: 7.43 (dt, $J = 9.1$, 3.0 Hz, 6H), 14,15-H: 7.24 (td, $J = 8.8$, 6.1 Hz, 4H), 19-H: 7.13 (dd, $J = 9.1$, 6.0 Hz, 2H), 18-H: 7.02 (dd, $J = 8.8$, 6.1 Hz, 2H), 9-H: 6.61 (dd, $J = 15.8$, 3.2 Hz, 2H), 8-H: 5.73 (ddd, $J = 16.1$, 5.8, 2.8 Hz, 2H), 6-H: 4.91 (p, $J = 6.9$ Hz, 2H), 4-H: 4.73 (s, 2H), 7-H: 4.59 (s, 2H), 2-H: 4.14 (m, 4H), 1-H: 4.82-2.71 (m, 6H), 5-H: 2.07 (m, 4H), 10,11,12-H: 1.59 (dd, $J = 7.0$, 2.5 Hz, 14H), 3-H: 1.29-1.24 (m, 2H).

### Preparation of FP NPs
The product Fluplatin was dissolved in DMF, then gradually dropped into water under stirring and stabilized by PEG−PE to finally obtain FP NPs. Finally, a 3500 D dialysis bag was used to dialyze for 6 h. Similarly, Dil@FP NPs and DiR@FP NPs were also produced by the same method. SEM analyses were performed on a FEI NOVA NANO450 by Eceshi; TEM analyses were performed on a FEI Talos F200S by Eceshi (www.eceshi.com).

### Combination index assay
The Synergy of cisplatin and fluvastatin sodium was performed on H1975 cells by MTT assay. Cells were seeded in 96-well plates at a density of $1 \times 10^4$ cells per well, and treated with a varying concentration of mixture of cisplatin and fluvastatin sodium in different ratio. After 24 h, cells were incubated with 200 μL serum-free media, and then 20 μL MTT solution (5 mg/mL) was added to each well in 4 h later. After that, each well was incubated with 150 μL of DMSO, and cells were determined at 490 nm. The combination index was measured by CompuSyn software (ComboSyn, Inc).

### Cell viability assay
The cytotoxicity of FP NPs in vitro was evaluated by MTT assays. Cells were seeded in 96-well plates at a density of $1 \times 10^4$ cells per well overnight. The cells were then incubated with a varying concentration of different formulations which were prepared with serum-free media, and the positive control group was untreated. After 24 h of treatment, 20 μL of MTT solution (5 mg/mL) was added to each well for 4 h. Then 150 μL of DMSO solution was added to the cells, and the absorbance was measured at 490 nm in a multifunctional microplate reader to calculate the viability of the cells.

### Calcein AM/PI double staining
The Live/Dead Cell Assay was conducted to evaluate the cell viability by using Calcein-AM/PI Double Stain Kit. Cells were seeded in 6-well plates at a density of $3 \times 10^5$ cells per well for 24 h. Then each well was incubated with drugs for 6 h. After that, the cells were stained with the Calcein-AM/PI Double Stain Kit for 20 min. After cell staining is complete, PBS was added to wash the cells 3 times, and the cells were examined simultaneously under an inverted fluorescence microscope (Nikon, Ti-S). Live cells were detected with a $490 \pm 10$ nm excitation filter, and dead cells were detected with a 545 nm emission filter.

### Apoptosis detection in vitro
We used an FITC Annexin V/PI apoptosis detection kit to detect apoptosis. $3 \times 10^5$ H1975 cells were seeded into 6-well plates. After treatment, all the attached cells were collected to disperse in 1× binding buffer solution (ice-cold) at a concentration of $3 \times 10^5$ cells. Then, the cells were incubated with a mixed cell suspension of FITC Annexin V and PI at room temperature for 20 min in darkness and analysis was performed using the FACS Calibur Flow Cytometer (BD Biosystems).

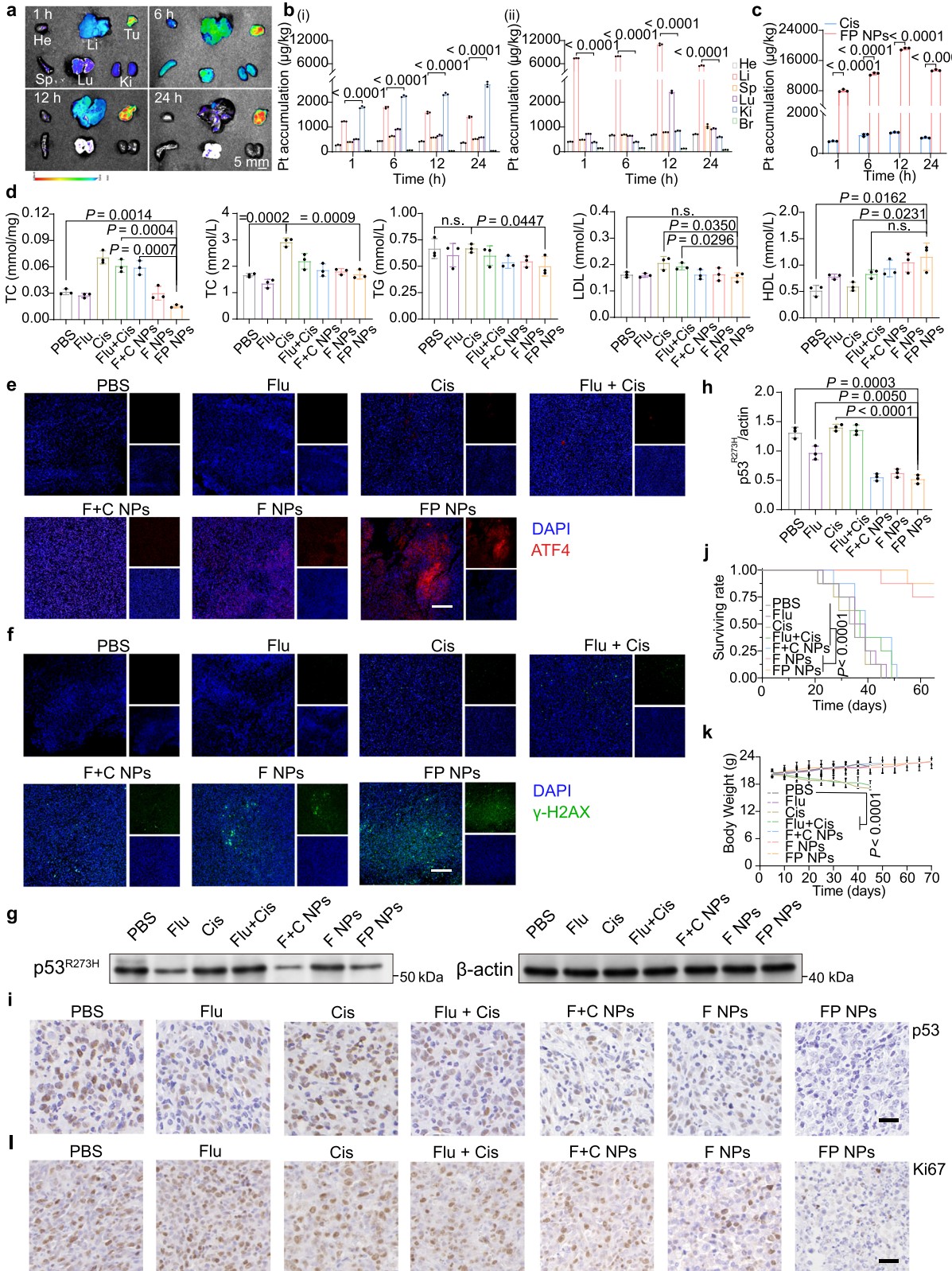

## Study of cellular uptake mechanisms

Endocytic pathway inhibitors were used to evaluate the cellular uptake mechanism of FP NPs, and untreated cells were set as controls. H1975 cells were seeded in 6-well plates at a density of $3 \times 10^5$ per well for 24 h. Cells were treated with various endocytic pathway inhibitors: clathin-mediated endocytosis inhibitors hypertonic sucrose (154 mg/mL) and chloroquine (0.032 mg/mL),

litteroprotein-mediated endocytosis inhibitors β-cyclodextrin (β-CD, 5 mM), macropinocytosis inhibitor amiloride (133.3 μg/mL), and energy-dependent endocytosis inhibitor mannitol (9.11 mg/mL). In addition, energy-dependent endocytosis inhibitor group was incubated with serum-free media at 4 °C. After 30 min of incubation, the cells were treated with 2 μM FP NPs, then the nanoparticle solution was removed in 4 h later. After that, the cells in each well were

**Fig. 7 | Biodistribution and anticancer mechanism of FP NPs in the H1975 tumor-bearing mouse model. a** Ex vivo biodistribution imaging of the main organs in mice bearing H1975 xenografts of the DiR@FP NPs at various time points post injection. Scale bars, 5 mm. **b** ICP–MS measurement of Pt accumulation in individual organs at 1, 6, 12, and 24 h after treatment with cisplatin (i) or FP NPs (ii) ($n = 3$ mice per group). **c** ICP–MS measurement of Pt accumulation in the tumor tissues at 1, 6, 12, and 24 h after treatment with cisplatin or FP NPs ($n = 3$ mice per group; one-way ANOVA followed by Tukey's HSD post hoc test). **d** TC levels in tumor tissues and TC, TG, HDL, LDL levels in serum after treatment with each group ($n = 3$ mice per group; one-way ANOVA followed by Tukey's HSD post hoc test). Confocal images of ATF4 (**e**) and γ-H2AX (**f**) in the tumor tissues of mice after

treatment with each group. Scale bars, 100 μm. **g** Western blotting analysis of p53$^{R273H}$ in tumor tissues after treatment with each group. Their grayscale values were quantified (**h**) ($n = 3$ independent samples; one-way ANOVA followed by Tukey's HSD post hoc test). **i** IHC of p53 in the tumor tissues after treatment with each group. Scale bars, 20 μm. **j** Kaplan-Meier survival curve of mice treated with each group over 70 days ($n = 8$ mice per group; Log-rank Mantel−Cox test). **k** The body weight of mice during treatments with each group over 70 days ($n = 8$ mice per group;two-way ANOVA followed by Tukey's multiple comparisons post test). **l** IHC of Ki67 in the tumor tissues after treatment with each group. Scale bars, 20 μm. Data are shown as the mean ± SD; n.s. no significance. Source data are provided as a Source Data file.

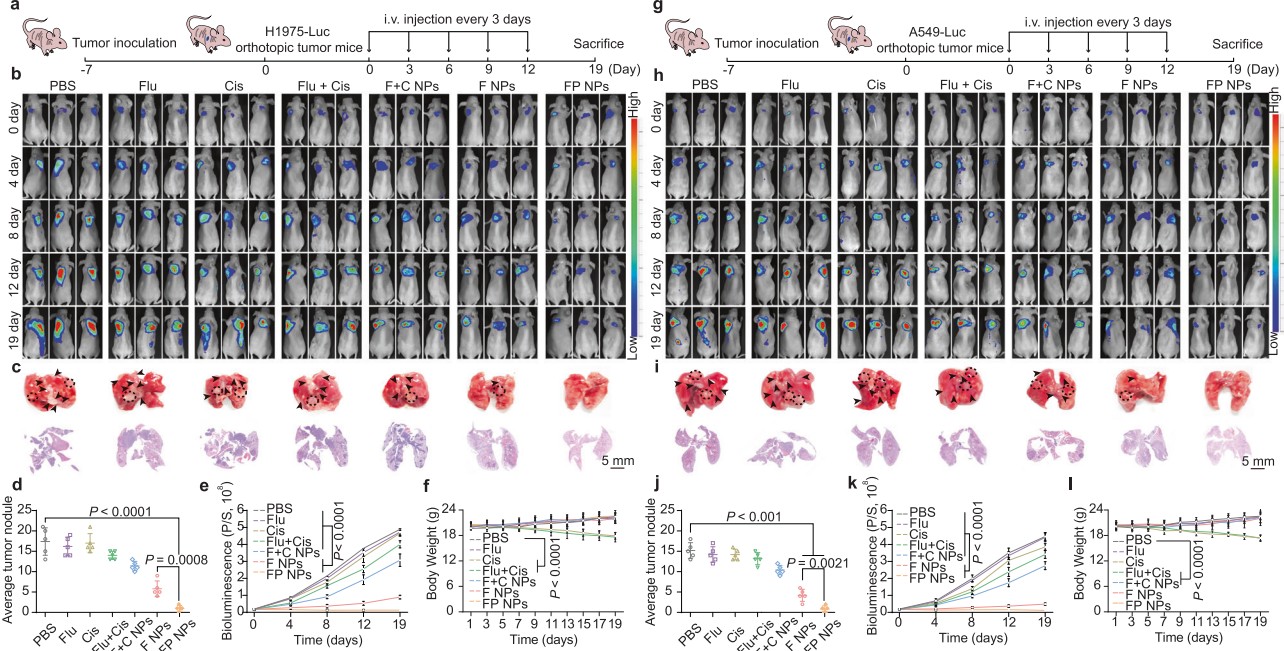

**Fig. 8 | Antitumor efficacy of intravenously injected FP NPs in H1975-luc and A549-luc orthotopic lung tumors. a, g** Schematic illustration of the experimental design. **b, h** The progression of orthotopic lung tumors ($n = 5$ mice per group) in BALB/c nude mice. A representative bioluminescent image from each group is shown. **c, i** The final anatomical picture and H&E staining of the lungs. Scale bars, 5 mm. **d, j** Average tumor nodule. Tumor nodules of 2–10 mm³ in volume were counted using harvested lungs from the control and treated groups, and the average number of tumor nodules was determined. Each dot represents a tumor

from an individual mouse. The tumor nodules in the lungs are indicated by arrows (one-way ANOVA followed by Tukey's HSD post hoc test). **e, k** Quantitative bioluminescence analysis in mice bearing orthotopic lung tumors; P/S = photons/second (two-way ANOVA followed by Tukey's multiple comparisons post test). **f, l** Mouse body weight in the orthotopic lung tumor mouse model (two-way ANOVA followed by Tukey's multiple comparisons post test). Data are shown as the mean ± SD; n.s. no significance. Source data are provided as a Source Data file.

counted and collected to measure intracellular Pt accumulation by ICP−MS.

## Sanger sequencing

Each sample was $1 \times 10^6$ cells before DNA extraction to ensure enough tissue to be processed. The primer pairs were: *TP53*–1,5′-GTCCC TCTCTGATTGTCTTTTCC-3′ and 5′-ACTGACAGGAAGCCAAAGGG-3′; *TPS3*–2,5′-TGTTTGTTTCTTTGCTGCCG-3′ and 5′-CAAATAAGCAGCA GGAGAAAGC-3′; *TP53*–3,5′-GAGACCATCCTGGCTAACGG-3′ and 5′-A CACCATCGTAAGTCAAGTAGCATC-3′. DNA was isolated by an Ezup Column Animal Genomic DNA Purification Kit. Samples were resuspended in buffer CL, proteinase K, CW1 Solution, CW2 Solution, and CE Buffer for further purification and PCR. After DNA purity and concentration were determined, 1 μL template DNA, 2 μL primer sequences, and 2.5 μL of Taq Buffer (with MgCl₂) (10×) were mixed together for the PCR cycle. The PCR conditions were set to 95 °C for 5 min followed by 40 cycles of 95 °C for 30 sec, 58 °C for 30 sec, and 72 °C for 30 sec with a final extension at 72 °C for 10 min. PCR products of 5 μL of each were analyzed in 1% agarose gel with 500 and 1000 bp ladder DNA

markers. The resulting PCR products were purified and sequenced in both directions using Sanger sequencing (Sangon Biotech Co., Ltd., Shanghai, China).

## Acquisition of liver hepatocellular carcinoma (LIHC) data

The database is publicly open-access and available, and therefore, there was no need to obtain approval from the local ethics committee[51]. The data of lung adenocarcinoma (LUAD) and lung squamous cell carcinoma (LUSC) patients with the expression of RNA-Seq and matching clinical pathologic information were obtained from the TCGA database (https://tcga.xenahubs.net). Maftools was used to identify the top 20 highly mutated genes and visualize mutations and their frequencies in all samples[51]. We obtained the mutation status of *TP53* through the cBioPortal for Cancer Genomics (http://www.cbioportal.org/)[52,53].

## GO and GESA analysis

The GO (http://geneontology.org/) enrichment of DEGs was implemented by the hypergeometric test, in which *p*-value was calculated

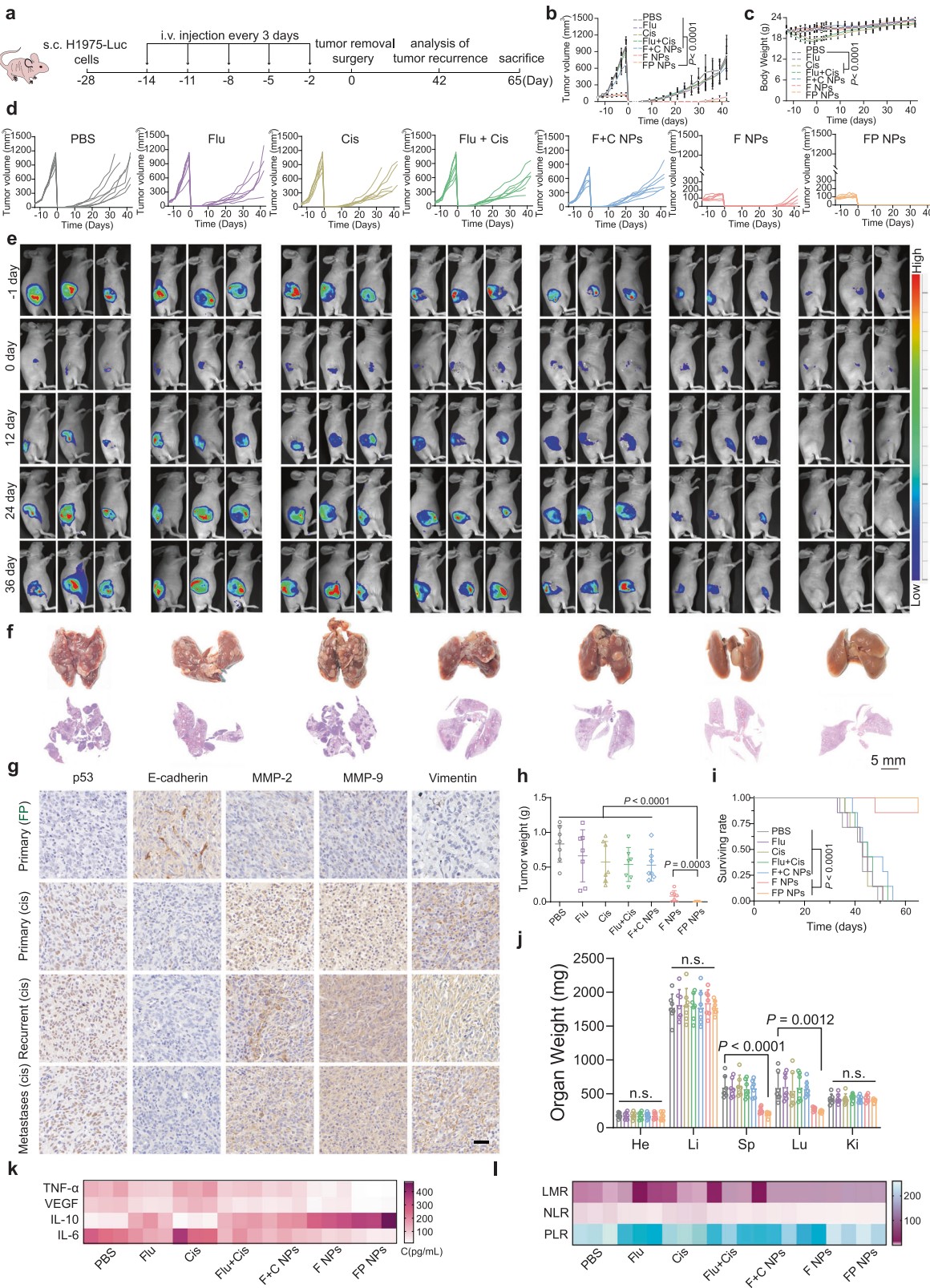

and adjusted, and the data background was genes in the whole genome. GO terms with adjusted $p$-value < 0.05 were considered to be significantly enriched. GO enrichment analysis could exhibit the biological functions of the DEGs. The significance of DEGs was determined by the threshold with an adjusted $p < 0.05$ and cut-off of $|Log2FC| \geq 1$. GSEA was ranked by their differential expression using ClusterProfiler (version 4.8.2) and GseaVis package

in R (https://github.com/junjunlab/GseaVis). Gene sets were considered to be significantly enriched with an alpha or $p$-value < 0.05 and a false discovery rate (FDR) < 0.25. The significance of gene sets from the GSEA was based on the normalized enrichment score (NES), $p$ value and adjusted p-value to determine the probability that a gene set with a given NES represents a false-positive finding.

**Fig. 9 | The tumor recurrence and metastasis inhibition of FP NPs in H1975-luc tumor-bearing mice. a** Schematic illustration of the experimental design. **b** The tumor volumes of mice during treatments with each group ($n = 7$ mice per group; two-way ANOVA followed by Tukey's multiple comparisons post test). **c** The body weight of mice during treatments with each group ($n = 7$ mice per group; two-way ANOVA followed by Tukey's multiple comparisons post test). **d** Tumor volume changes for each mouse in each group ($n = 7$ mice per group). **e** In vivo bioluminescence imaging of tumor-bearing mice receiving various treatments after surgery. Three representative mice in each treatment group are shown. Images of day 0 were taken on the day of surgery. **f** The final anatomical picture and H&E staining of the lungs ($n = 3$ independent samples; Scale bars, 5 mm). **g** IHC of p53, E-cadherin, MMP-2, MMP-9, Vimentin in the primary tumor tissues of cisplatin or FP NPs treatment, and recurrent and metastatic tumor tissues of cisplatin treatment ($n = 3$ independent samples). Scale bars, 20 μm. **h** Recurrence tumor weights of different groups on day 42 after surgery ($n = 7$ mice per group; one-way ANOVA followed by Tukey's HSD post hoc test). **i** Kaplan-Meier survival curve of mice treated with each group over 65 days ($n = 7$ mice per group; Log-rank Mantel–Cox test). **j** Final weights of the heart, liver, spleen, lungs, and kidneys ($n = 7$ mice per group; two-tailed unpaired t test). **k** Heatmap of TNF-α, VEGF, IL-10 and IL-6 expression profiles in serum ($n = 3$ mice per group). **l** Indicators of routine blood examination ($n = 3$ mice per group; two-tailed unpaired t test) of mice. Data are shown as the mean ± SD; n.s. no significance. Source data are provided as a Source Data file.

## RNA sequencing analysis

The RNA sequencing of cells was completed in Annoroad Gene Technology Corporation (Beijing, China). Total RNA was extracted from the cells using TRIzol reagent according to the standard protocol. RNA purity was evaluated using a NanoPhotometer® (IMPLEN, CA, USA). RNA integrity and concentration were assessed using the RNA Nano 6000 Assay Kit of the Bioanalyzer 2100 system (Agilent Technologies, CA, USA). After the total RNA samples were tested to be qualified, the MGIEasy RNA library preparation kit was used for library construction. Magnetic beads with oligo (DT) were used for enrichment and purification. The samples were sequenced on DNBSEQ-T7 for sequencing, and 150 bp double ended sequencing reads were obtained. After primary sequencing data passed quality control, the raw reads were filtered to obtain clean reads, which were aligned to the reference genome using HISAT2 v.2.1.0. RNA-seq data have been deposited in NCBI's Gene Expression Omnibus.

## Antitumor assay in vivo

Subcutaneous tumor model: The mice housing in SPF environment were subcutaneously injected with 100 μL H1975 cells ($1.0 \times 10^7$/mL). When the tumors grew to -100 mm³, the mice were randomly divided according to the experimental needs. Orthotopic lung cancer model: Cells were labeled with luciferase and mixed 1:1 with matrix gel to a final concentration of $1.0 \times 10^7$/mL. Mice were anesthetized with isoflurane (flow rate of 500–1000 mL/min), with the vaporizer set at 5%, until signs of recumbency were present. Mice were placed in the right lateral recumbent position, and a 1 cm incision was made after disinfection. A 1 mL syringe was used to puncture vertically at the upper edge of the ribs for about 3 mm, slowly injecting about 50 μL of cell suspension staying for 3–5 seconds, quickly withdrawing the needle, disinfecting, and suturing with iodine povidone. The wounds could be healed in 4–6 days, so as to establish the orthotopic grafted tumor model of lung cancer. Tumor-bearing mice (randomly divided into seven groups ($n = 5$)) were tail vein injected with 0.1 mM PBS, cisplatin (3 mg/kg), fluvastatin sodium (8.67 mg/kg), a mixed solution of cisplatin and fluvastatin sodium, F + C NPs, F NPs and FP NPs (Fluplatin 10.49 mg/kg). Each group was dosed once every 3 days five times, and the tumor volume and body weight of the mice were recorded daily. Euthanasia was performed after observation until day 19, when blood was obtained by retro-orbital sinus/plexus sampling (using isoflurane, flow rate 500–1000 mL/min). The survival status of the mice was recorded daily until 65 days later. In our animal study, the maximum permissible tumor size (20 mm in diameter) was not exceeded, the weight loss due to chemotherapy was less than 20%, and the nude mice were in good condition throughout the procedure (according to the requirements of the Animal Welfare and Ethics Committee of China Pharmaceutical University).

## Biodistribution in vivo

Tumor-bearing mice were randomly divided into DiR@FP NPs groups ($n = 12$). Three mice were sacrificed by tail vein injection of DiR@FP NPs (Fluplatin 10.49 mg/kg), and after 1 h, 6 h, 12 h, and 24 h, major organs (heart, liver, spleen, lung, and kidney) and tumor tissues were dissected for in vivo imaging observation.

Tumor-bearing mice were randomly divided into two groups of cisplatin and FP NPs ($n = 12$). Cisplatin (3 mg/kg) or FP NPs (Fluplatin 10.49 mg/kg) were injected via the tail vein, and after 1 h, 6 h, 12 h, and 24 h, 3 mice were sacrificed in each group and the major organs (heart, liver, spleen, lung, and kidney) and tumor tissues were dissected. The tissues and organs were well ground and the biodistribution of the drug in vivo was evaluated by ICP–MS.

## Tumor recurrence and metastasis assay

The tumor-bearing mice were randomly divided into two groups ($n = 7$) of each group. First, each group was injected via the tail vein once every 3 days five times. Two days after the end of the drug treatment, mice were injected subcutaneously at 100 μL with 0.1 mg/kg buprenorphine (30 min before) and anesthetized with isoflurane (4% in 100% oxygen at a flow rate of 2 L/min). After ensuring that the nude mice entered complete anesthesia, the tumors were completely removed surgically by an experienced operator over surgery to remove the tumors. Briefly, after the animal was anesthetized, a small incision is made with a scalpel at a distance from the tumor and then most of the tumor was removed under the microscope. The scalpel was used under the microscope to further remove the visible tumor residue without destroying the connective tissue, and finally the wound was closed using biologic tissue glue and surgical sutures (the anesthetic status and surgical asepsis need to be ensured throughout). After anesthesia, the health of the nude mice was confirmed and the mice were warmed.

## Statistics and reproducibility

All studies were performed at least in triplicate unless otherwise stated. Statistical analysis was performed using GraphPad Prism 7.0 software (GraphPad Software, CA). For comparison of multiple groups, one-way analysis of variane (ANOVA) or two-way ANOVA was used, followed by Tukey's honest significant difference post hoc test or Bonferroni's multiple comparisons post-test, except where otherwise noted. Comparison between the two groups was performed using an unpaired two tailed Student's $t$ test. For survival studies, Log-rank Mant–Cox tests were performed to determine significance. For RNA sequencing analysis, wald tests were performed and corrected via BH method. For GO enrichment analysis, fisher tests were performed and corrected via BH method. For GSEA analysis, permutation tests were performed and corrected via BH method. Results are expressed as means ± SD. $P < 0.05$ was considered statistically significant. The Image J software (National Institutes of Health, USA) was applied for quantitative analysis in fluorescence intensity for confocal images. For all studies, samples were randomly assigned to various experimental groups. No data were excluded from the analyses. Measurements were taken from distinct samples rather than repeated measuring the same sample. All data were obtained from the results of at least three independent experiments.

**Reporting summary**

Further information on research design is available in the Nature Portfolio Reporting Summary linked to this article.

## Data availability

The RNA sequencing data used in this study are available at National Center for Biotechnology Information Sequence Read Archive (SRA) database with accession code PRJNA1047938. The authors declare that all other data supporting the findings of this study are available within the article and its Supplementary Information files. Should any data files be needed in another format they are available from the corresponding author upon reasonable request. Source data are provided with this paper.

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

## Acknowledgements

This work was financially supported by the National Natural Science Foundation of China (82020108029, 82073398). This work was also supported by the Project of State Key Laboratory of Natural Medicines, China Pharmaceutical University (SKLNMZZ202021), Double First-class University Projects (CPU2018GY06).

## Author contributions

Conceptualization: Y.Y.B. Methodology: Y.Y.B., Q.C. Investigation: Y.Y.B., Q.C., M.Y.Y. Visualization: Y.Y.B., M.Y.Y. Funding acquisition: H.L.J. Project administration: L.X., H.L.J. Writing: Y.Y.B., L.X., H.L.J.

## Competing interests

The authors declare no competing interests.
