## [Peer Review File · Nature Communications]

Reviewers' comments:

Reviewer #1 - cisplatin prodrugs, nanoparticles - (Remarks to the Author):

The present research manuscript titled “Mutant-p53-targeting- therapeutic nanoparticles resolves chemoresistance and tumor recurrence in non-small cell lung cancer” by Bi et al is a well-written manuscript with apparent logic. The majority of conclusions are supported by a significant amount of data. The authors think out of the box and synthesized a novel prodrug of fluvastatin with cisplatin by taking p53 mutation, a major problem of cisplatin resistance, as an entry point. At the same time, the prodrug can also self-assemble into homogeneous nanoparticles, and the encapsulation by PEG-PE can cleverly combine the endoplasmic reticulum (ER) targeting ability of the prodrug to further amplify the ER stress which is independent of p53. Notably, this formulation breaks the restriction of p53 mutation and remains apoptogenic via p53 in wild-type cells, while once mutated, it can effectively degrade mutated p53 and cause apoptosis through a p53-independent pathway. In addition, this therapeutic strategy, not only achieved superior anti-tumor effects in vitro and in vivo, but also improved the poor prognosis of NSCLC, which is also of great clinical significance. As a first-line drug for NSCLC, chemoresistance is a major limitation in its application, and it is meaningful that this article can significantly address this issue by targeting its key action on p53. I would like to recommend this remarkable manuscript to be published in Nature Communications with minor revisions. My comments are as follows.

- 1) It would be more informative to give a more specific experimental procedure for the induction of p53 mutation after cisplatin treatment presented in Fig. 2c.
- 2) The yield of Fluplatin is 64% as mentioned in the manuscript, can a more precise value be given here? And can the exact yield at which step be reflected in the synthesis method?
- 3) In Fig. 4i, the platinum content of different organelles was measured by ICP–MS. Is it possible to directly show the ratio of the FP NPs to the cisplatin content of cisplatin and Fluplatin to better reflect the advantage of FP NPs?
- 4) It should have been clearer in the presentation in lines 207-209. It should directly reflect that PEG-PE is labeled by FITC, while Dil is assembled in Fluplatin self-assembled nanoparticles.
- 5) Notice that Fig. 5c needs to be added to the scale. I think the labeling of the positions in the confocal images corresponding to the fluorescence intensities of Dil and ER/Mito-tracker in Fig. 5b,d should be more clear.
- 6) In Supplementary Fig. 27 you provide the fluorescence intensity of ROS at different times, in addition to that, the values of the coefficient of ROS and ER need to be provided.
- 7) Regarding your examination of recurrence in mice, could you give more specific steps to the procedure? In addition, I would also like to know some links between the examined indicators in recurrence and p53 mutations and the link with recurrence?

Reviewer #2 - targeting P53, therapy (Remarks to the Author):

This is an interesting study aiming at the development of nanoparticles targeting p53 mutations to reverse cisplatin resistance. Although this project is not particularly innovative, it is of significance and has a high translational potential.

1. The lack of novelty of this project is of concern. It is not true that no strategies to target mutated p53 for reversing cisplatin resistance. A thorough literature search and review is needed to make such a statement (Line 49-50). For example, similar findings shown in Figure 2 have been well reported. No evidence for increase in p53 mutations is provided (Fig 2).
2. The clinical relevance of the mutations in A549 cells is not clear and clinical evidence is needed. The biological significance of the cisplatin-induced p53 mutations in A549 cells is overly simplified by showing the changes in cholesterol levels in the cells. The clinical relevance of this type experiments is not clear (Figs 2c, d, e). The mechanisms responsible for p53 mutation, cholesterol production/metabolism and cell cytotoxicity are not demonstrated. The authors made their conclusions based on correlation analysis, not mechanistic studies. In addition, the study design in mechanistic studies is flawed by switching cell line among different assays (Figs 2g-k)
3. The selection of Fluvastatin is not well justified.
4. The purity of FPNPs was not clearly demonstrated.
5. Fig. 4: The relative values of IC50s (the ratios of cisplatin vs FP NPs) among the three cell lines are similar, indicating that there are no specific effects on drug resistant cells and there is no direct relation to p53 mutation.
6. Fig. 5: No direct evidence for FPNPs' specific effects on p53 mutations.
7. Figs. 6,7, and 8: Only one mutated cell line is used. P53 wt and p53 mt and drug resistant cells should be used to demonstrate the specific effects of FP NPs on reversing cisplatin drug resistance.
8. This manuscript is not well prepared and it's hard to follow. There are many typos and grammatic errors in the entire manuscript (e.g., line 42). The lack of logical reasoning and the lack of logical information flow in writing are of concern. For example, it is not clear what kind of negative effect of cisplatin on TP53 mutations is referred to (line 46-47). Another example, the figures are not cited in the text in a logical order (e.g., Figure 1c was cited first and before Fig 1 a and b in the text, suppl Fig 16 was cited before suppl Fig. 8).
9. The methods section is not well referenced. Sex as a biological variable is not considered in the animal study design.
10. The discussion section is not well prepared; the mechanism of action, the clinical relevance of the models used and the results obtained are not well discussed. The presented results do not support the authors' conclusions.

Reviewer #3 - Drug delivery, nanotherapy - (Remarks to the Author):

Authors in their manuscript entitled "Mutant-p53-targeting-therapeutic nanoparticles resolves chemoresistance and tumor recurrence in non-small cell lung cancer" have prepared a prodrug of fluvastatin (inhibitor for mutant p53) and cisplatin and encapsulated in self-assembling nanoparticles. These nanoparticles were tested in different animal models and shown to inhibit the tumor growth of NSCLC promisingly. The idea is that nanoparticles enter tumors via passive delivery and taken up by tumor cells and inhibit mutant p53 while cisplatin induces apoptosis more effectively. The study has been well thought and experiments are performed in detail. The anti-tumor effects looks attractive, but the study lacks novelty in terms of chosen drugs and the concept of combination.

Fluvastatin is proposed as a drug to degrade mutant p53 in this study. However, this drugs is not specific for mutant p53 but has many other off-target effects which are responsible for inducing apoptosis.

Previous studies have reported the combination of fluvastatin and cisplatin (<https://pubmed.ncbi.nlm.nih.gov/20837358/>) which shows lack of new idea of using this combination. Fluvastatin was withdrawn from the market and therefore future translation is questionable.

Technically, the study has many shortcomings. for examples

1. stability studies in Fig 3e were only performed in buffers but not in plasma. There was a substantial release of cis-pt at pH7.4.
2. No nanoparticles were made without prodrug proving the additional value of the prodrug in nanoparticles. This is important to show that the prodrug as NPs has advantage over the simple encapsulation of these two drugs in polymeric NPs.
3. In fig 4, it is not clear what do they mean with 4uM different formulations. Fluplatin includes 2 molecules of Fluvastatin which means there could be 2x drug compared to free fluvastatin and this could lead to higher effect.
4. In vivo: The used animal models are not optimal. They used subcutaneous tumor for NSCLC and orthotopic tumor model should be used as these tumors lack the microenvironment. They
5. Also, tumor were allowed to grow to the size of about 3000 mm³ which are ethically not acceptable mostly nowadays.
6. One major problem is the lack of control NPs including both Flu + cis in nanoparticles. Why other control Fluplatin without NP not included in vivo?
7. Fig 7 misses quantitative analysis of staining and Western blot.

Responses to the comments of the Reviewers

Reviewer #1 - cisplatin prodrugs, nanoparticles - (Remarks to the Author):

The present research manuscript titled “Mutant-p53-targeting-therapeutic nanoparticles resolves chemoresistance and tumor recurrence in non-small cell lung cancer” by Bi et al is a well-written manuscript with apparent logic. The majority of conclusions are supported by a significant amount of data. The authors think out of the box and synthesized a novel prodrug of fluvastatin with cisplatin by taking p53 mutation, a major problem of cisplatin resistance, as an entry point. At the same time, the prodrug can also self-assemble into homogeneous nanoparticles, and the encapsulation by PEG-PE can cleverly combine the endoplasmic reticulum (ER) targeting ability of the prodrug to further amplify the ER stress which is independent of p53. Notably, this formulation breaks the restriction of p53 mutation and remains apoptogenic via p53 in wild-type cells, while once mutated, it can effectively degrade mutated p53 and cause apoptosis through a p53-independent pathway. In addition, this therapeutic strategy, not only achieved superior anti-tumor effects in vitro and in vivo, but also improved the poor prognosis of NSCLC, which is also of great clinical significance. As a first-line drug for NSCLC, chemoresistance is a major limitation in its application, and it is meaningful that this article can significantly address this issue by targeting its key action on p53. I would like to recommend this remarkable manuscript to be published in Nature Communications with minor revisions. My comments are as follows.

Response: Thank you very much for your endorsement on our work. Please find our responses to the comments point-by-point below.

1) It would be more informative to give a more specific experimental procedure for the induction of p53 mutation after cisplatin treatment presented in Fig. 2c.

Response: Thank you for your kind comments. Based on the recommendations of the reviewers, we have added more detailed and specific experimental procedure schematically in Fig. 1b (Fig. 2c of the original manuscript), and the description is updated in the revised manuscript.

Fig. 1 Mechanistic validation of the existence of a vicious cycle between cisplatin and p53. **a** Kaplan-Meier plot of the correlation between the mutation of p53 and the survival of patients with NSCLC (Log-rank Mantel–Cox test). Cisplatin (Cis), fluvastatin sodium (Flu). **b** Outline of the three assays for evaluating TP53 mutations, ROS levels, Cho levels and DNA damage before and after 30 days of cisplatin treatment (low dose, 10 μM ; moderate dose, 25 μM ; high dose, 50 μM). Heatmap of the number of mutations and RNA-seq analysis (**c**), ROS levels (**d**), Cho levels (**e**) and DNA damage (**f**) for the wtp53-expressing cells with cisplatin treatment for 30 days. **g** High-throughput sequencing before and after fluvastatin sodium (4

μM) treatment in A549 cells. **h** Genome-wide analysis of differential expression of fluvastatin sodium treated and untreated A549 cells after cisplatin induced. The volcano plot depicts the significance (false discovery rate adjusted p-value) and magnitude of difference (Fold Change). The dashed line indicates the threshold of the Fold Change > 2 and adjusted p-value < 0.05 . Some of the cancer related genes are labeled by dark colors. **i** GO enrichment analysis of differentially expressed genes (DEGs). The advanced bubble chart shows GO enrichment of DEGs in signaling pathways. The x-axis label represents the gene ratio, and the y-axis label represents GO terms. **j** GSEA shows how Flu affects the gene signatures of cholesterol metabolism. The normalized enrichment scores (NES) and p-values are indicated in each plot. **k** Heat map analysis of 17 mevalonate pathway genes and the SREBP2 gene from RNA-seq data using fluvastatin sodium treated and untreated A549 cells after cisplatin induced. The color scale indicates the expression fold change of genes. After Pifithrin- α and STI571 treatment for 24 h, the cytotoxicity and IC_{50} of cisplatin treatment on in H1299 cells transfected with expression constructs containing the wtp53 (**l**), vector (**m**), and expression constructs containing the R273H variant (**n**) were measured by MTT method, and then the enzymatic activity of caspas3/9 was detected ($n = 3$ independent samples; two-tailed unpaired t test for **l**(i, ii), **m**(i, ii) and **n**(i, ii); one-way ANOVA followed by Tukey's HSD post hoc test for **l**(iv), **m**(iv) and **n**(iv)). **o** Dot plots showing the $\text{IC}_{50}(\text{Cis})/\text{IC}_{50}(\text{Cis} + \text{Flu})$ in mutant and wild-type cells. **p** Schematic illustration of the treatment with cisplatin in mutp53 or wtp53 cells. Data were shown as the mean \pm s.d. Source data are provided as a Source Data file.

2) The yield of Fluplatin is 64% as mentioned in the manuscript, can a more precise value be given here? And can the exact yield at which step be reflected in the synthesis method?

Response: Thank you for your comments. The yield of Fluplatin was 64.18% (Synthetic yield coefficient = Actual product mass \div Theoretical product mass $\times 100\%$). We calculated the weight after purification and drying of the organic solvent and updated the contents of this part in the revised manuscript.

3) In Fig. 4i, the platinum content of different organelles was measured by ICP-MS. Is it possible to directly show the ratio of the FP NPs to the cisplatin content of cisplatin and Fluplatin to better reflect the advantage of FP NPs?

Response: Considering the reviewer's suggestions, we have directly shown the ratio of the FP NPs to the cisplatin content of cisplatin (48.8-fold in the endoplasmic reticulum) and Fluplatin (4.8-fold in the endoplasmic reticulum), and updated the contents of this part in the revised manuscript as shown in Fig. 2m (Fig. 4i of the original manuscript).

Fig. 2 Design and characterization of FP NPs. **a** CI value in different ratios of physical mixture between cisplatin and fluvastatin sodium, and the chemical structure and synthesis route of Fluplatin. **b** Particle size, zeta potential, PDI, and TEM image of the Fluplatin self-assembled nanoparticles. Scale bars, 50 nm. **c** Particle size, PDI, and encapsulation efficiency of nanoparticles formed by different proportions of Fluplatin and PEG-PE ($n = 3$ independent samples). **d** Particle size, zeta potential, PDI, and TEM image of the FP NPs. Scale bars, 200 nm. **e** Release profile of the Fluplatin self-assembled nanoparticles (i) and FP NPs (ii) in phosphate solution of 10% plasma at pH 7.4. **f** Release profile of the Fluplatin

self-assembled nanoparticles (i) and FP NPs (ii) in phosphate solution of 1% Tween 80 at pH 7.4, pH 6.5, and pH 4.5 ($n = 3$ independent samples). **g** Mechanism for the Fluplatin self-assembled nanoparticles (i) and FP NPs (ii) ($n = 3$ independent samples). **h** SEM image and EDS element mapping of FP NPs. Scale bars, 3 μm . **i** Confocal images of Dil@FP NPs in H1975 cells after treatment with 2 μM Dil@FP NPs for 1 h, 2 h, 4 h, 8 h, respectively. Scale bars, 20 μm . **j** Uptake efficiency of 2 μM FP NPs treatment for 4 h in H1975 cells in the presence of various endocytosis inhibitors was detected by ICP-MS ($n = 3$ independent samples; one-way ANOVA followed by Tukey's HSD post hoc test). **k** Confocal images of Dil@FP NPs and PEG-PE in H1975 cells after treatment with 2 μM Dil@FP NPs for 1 h, 2 h, 4 h, respectively. Scale bars, 5 μm . **l** Uptake Pt in H1975 cells after treatment with 2 μM different formulations (dosage based on Fluplatin) for 1 h, 2 h, 4 h, 8 h detected by ICP-MS, respectively ($n = 3$ independent samples; one-way ANOVA followed by Tukey's HSD post hoc test). **m** Uptake Pt in nuclear, lysosome, Golgi apparatus, endoplasmic reticulum, mitochondria after 8 h treatment of 2 μM different formulations (dosage based on Fluplatin; $n = 3$ independent samples; one-way ANOVA followed by Tukey's HSD post hoc test). **n** Major intracellular distribution of FP NPs. Data were shown as the mean \pm s.d. Source data are provided as a Source Data file.

4) It should have been clearer in the presentation in lines 207-209. It should directly reflect that PEG-PE is labeled by FITC, while Dil is assembled in Fluplatin self-assembled nanoparticles.

Response: Thank you for the suggestion. According to reviewer's suggestions, we clarified the language to more clearly illustrate the corresponding components of FITC and Dil in the revised manuscript, the sentence was rewritten as "To verify the mechanism of PEG-PE membrane insertion, we constructed FITC-Dil@FP NPs (PEG-PE is labeled by FITC, while Dil is assembled in F NPs) and observed their distribution in cells using CLSM."

5) Notice that Fig. 5c needs to be added to the scale. I think the labeling of the positions in the confocal images corresponding to the fluorescence intensities of Dil and ER/Mito-tracker in Fig. 5b,d should be more clear.

Response: We are grateful for the kind suggestion. We have added a scale to Fig. 4c (Fig. 5c of the original manuscript). According to the reviewer's comments we relabeled the positional annotations in the confocal images corresponding to the fluorescence intensity in the revised manuscript as shown in Fig. 4b, d (Fig. 5b, d of the original manuscript).

Fig. 4 Mechanism of p53-independent antitumor activity of FP NPs in vitro. **a** Schematic summary of the p53-independent antitumor mechanism of FP NPs. **b** Confocal images of Dil and ER-Tracker in H1975 cells treated with 2 μ M Dil@FP NPs. Their colocalization determined by Pearson's correlation coefficient were quantified ($n = 3$ independent samples; one-way ANOVA followed by Tukey's HSD post hoc test). Scale bars, 10 μ m. **c** Confocal images of Dil and Fluo-4 in H1975 cells treated with 2 μ M Dil@FP NPs. And their fluorescence intensity was quantified ($n = 3$ independent samples; one-way ANOVA followed by Tukey's HSD post hoc test). Scale bars, 10 μ m. **d** Confocal images of Dil and

Mito-Tracker in H1975 cells treated with 2 μ M Dil@FP NPs, and their colocalization determined by Pearson's correlation coefficient were quantified ($n = 3$ independent samples). Scale bars, 10 μ m. **e** Confocal images of JC-1 in H1975 cells treated with 4 μ M different formulations (dosage based on Fluplatin) for 6 h, and their fluorescence intensity were quantified (**j**), ($n = 3$ independent samples; one-way ANOVA followed by Tukey's HSD post hoc test). Scale bars, 10 μ m. Western blotting analysis of p-eIF2 α , eIF2 α , CHOP, ATF4 in H1975 cells (**f**) and A549 cells (**g**) after treatment with 4 μ M of different formulations (dosage based on Fluplatin; $n = 3$ independent samples). **h** Western blotting analysis of p-eIF2 α , eIF2 α , CHOP, ATF4 in H1975 cells after treatment with different concentration of FP NPs for 12 h ($n = 3$ independent samples). **i** Confocal images of IF staining against γ -H2AX in H1975 cells treated with 4 μ M FP NPs for 6 h, and their fluorescence intensity were quantified (**k**), ($n = 3$ independent samples; one-way ANOVA followed by Tukey's HSD post hoc test). Scale bars, 10 μ m. Source data are provided as a Source Data file.

6) In Supplementary Fig. 27 you provide the fluorescence intensity of ROS at different times, in addition to that, the values of the coefficient of ROS and ER need to be provided.

Response: Thank you for your precious comments and advice. Considering the reviewer's suggestions, we analyzed the values of the coefficient of ROS and ER using Image-J and supplement them in Supplementary Fig. 27.

Supplementary Fig. 27 Co-localization results of ROS and ER in H1975 cells. a Confocal images of ROS and ER-Tracker in H1975 cells treated with 4 μ M FP NPs for 1 h, 4 h, 8 h, respectively. Their colocalization determined by Pearson's correlation coefficient was quantified. **b** ROS fluorescence intensity were quantified. Scale bars, 10 μ m. Data were shown as the mean \pm s.d. Statistical analysis was performed using one-way ANOVA followed by Tukey's HSD post hoc test; n.s. = no significance.

7) Regarding your examination of recurrence in mice, could you give more specific steps to the procedure? In addition, I would also like to know some links between the examined indicators in recurrence and p53 mutations and the link with recurrence?

Response: Thank you for these thoughtful comments. Detailed steps in the experimental manipulation of recurrence have been added to the revised manuscript as suggested by the reviewers.

P53 mutations are common in tumorigenesis, disrupting cell cycle regulation and affecting susceptibility to recurrence¹. Aggressive genetic markers synergize with p53 gene mutations to enhance oncogenic pathways. Elevated cell proliferation rates characteristic of

malignant tumors acts synergistically with p53 gene mutations to increase genomic instability and lead to recurrence. Microenvironmental factors, including immune regulation and nutritional supply, interact with p53 mutations to create an environment conducive to recurrence²⁻⁴. A comprehensive understanding of these associations can facilitate targeted interventions to curb p53-related relapse pathways. Therefore, it is of strong clinical relevance to examine FP NPs from the perspective of recurrence in highly p53-mutated non-small cell lung cancer. Based on the reviewer's comments, we have enhanced the discussion of this section in the revised manuscript.

References

1. Jeong, Y. et al. Role of KEAP1/NRF2 and TP53 mutations in lung squamous cell carcinoma development and radiation resistance. *Cancer Discov.* **7**, 86–101 (2017).
2. Tsao, M. S. et al. Prognostic and predictive importance of p53 and RAS for adjuvant chemotherapy in non small-cell lung cancer. *J. Clin. Oncol.* **25**, 5240–5247 (2007).
3. Ganesh, K. et al. Targeting metastatic cancer. *Nat. Med.* **27**, 34–44 (2021).
4. Jin, X. et al. A metastasis map of human cancer cell lines. *Nature* **588**, 331–336 (2020).

Thank you very much!

Reviewer #2 - targeting P53, therapy (Remarks to the Author):

This is an interesting study aiming at the development of nanoparticles targeting p53 mutations to reverse cisplatin resistance. Although this project is not particularly innovative, it is of significance and has a high translational potential.

Response: Thank you very much for your recognition of our work. Please find our responses to the comments point-by-point below.

1) The lack of novelty of this project is of concern. It is not true that no strategies to target mutated p53 for reversing cisplatin resistance. A thorough literature search and review is needed to make such a statement (Line 49-50). For example, similar findings shown in Figure 2 have been well reported. No evidence for increase in p53 mutations is provided (Fig 2).

Response: Thank you very much for your comment, we would still like to illustrate the novelty of our study in terms of:

1. A new starting point for alleviating cisplatin resistance: induction of p53 mutations by cisplatin. As reviewer mentioned, there are a lot of studies that have been done to address the issue of cisplatin resistance, with various structural modifications or multidrug co-delivery of cisplatin¹, but have yet focused more on this key point of p53 mutations. Although p53 mutations have been extensively studied as a cause of resistance to cisplatin chemotherapy, this has not been a broad entry point in the design of cisplatin prodrugs.

2. Novel mechanisms of FP NPs in targeting p53 mutations. First, we first synthesized a prodrug Fluplatin. Fluplatin self-assembled nanoparticles were further coated with PEG-PE to obtain the final formulation of FP NPs (Fig. 2b-d). **FP NPs degrade cisplatin-induced/existing mutp53, upgrade wtp53, activate potent ERS, and eliminate "GOF"** (Fig. 3-5). **FP NPs reverse cisplatin resistance due to p53 mutations, as well as achieving highly effective antitumor effects** (Fig. 3). Although both cisplatin and fluvastatin are old drugs that have been used in antitumor research, our starting point is emerging (**proposed from the limitations of cisplatin in the p53 pathway**), the combinations are different in approach and purpose (**On the one hand, degrades mutp53 and mitigates "GOF"; on the other hand, activates ERS and induces p53-independent apoptosis**), and the focus on the p53 mutations is different (**if the problem posed by p53 cannot be solved comprehensively, it is used to achieve a more efficient antitumor effect**).

3. The ability of FP NPs to inhibit recurrence was examined. Non-small cell lung cancer (NSCLC) with p53 mutations has a worse prognostic status and shorter survival compared to wild type (Fig. 1a). Whereas, FP NPs significantly reversed the poor prognostic status of

p53-mutated NSCLC due to their unique mechanism of action.

We appreciate your mention of “strategies to target mutated p53 for reversing cisplatin resistance” and the fact that some of the backgrounds we have laid out in Fig. 2 (Fig. 1 of the revised manuscript) have been well done. As we described in the background, cisplatin is an indispensable drug in the treatment of NSCLC and it is inactivated for various reasons². The link between cisplatin and p53 mutations can be summarized as: the importance of p53 in the antitumor effect of cisplatin³; the negative impact of p53 mutations on the action of cisplatin⁴; the high mutation rate of p53 in NSCLC⁵; and the fact that cisplatin or cisplatin-produced ROS can directly cause mutations^{3,6} (no clear evidence that cisplatin induced *TP53* mutations in authoritative journals). Therefore, such a process is undoubtedly a malignant accumulation process for NSCLC, the resistance to cisplatin was alleviated by modulating mutp53. However, there are no studies that start with cisplatin to break this vicious circle, hence the statement in the manuscript “Despite all the basic research performed to date, no studies have been conducted on strategies used to target mutp53 to reverse the treatment bottleneck caused by cisplatin therapy failure”.⁷ In the revised manuscript we have improved the presentation to better understand the studies we referenced. In addition, we added experiments based on the reviewer’s comments, and deepened the degree of mechanistic studies in Fig. 1 by starting with the therapeutic bottleneck of cisplatin in NSCLC.

Thank you for your comment “No evidence for increase in p53 mutations is provided (Fig. 2 (Fig. 1 of the revised manuscript))”. There is established clinical evidence that patients resistant to cisplatin chemotherapy tend to have more detectable p53 mutations⁸ and that cisplatin as a chemotherapeutic agent carries a risk of inducing mutations^{3,6}. In our original manuscript, we treated wtp53-expressing A549 cells with a single dose of cisplatin for one month to mimic the course of chemotherapy and ultimately detected mutations in *TP53*. Following the reviewer’s comments, we set up three dose groups of high, moderate and low when examining the effect of cisplatin on p53 mutations in Fig. 1b, and directly examined the p53 mutation status and whether it can still function as wild p53 by Sanger sequencing and RNA sequencing (RNA-seq) analysis. In addition, the mutation status of p53 was examined indirectly by three indicators closely related to the mutation status of p53, which are ROS content, DNA damage and cholesterol content. As shown in Fig. 1c-f, cisplatin-induced p53 mutations increased in a concentration-dependent manner, while the ROS content, DNA damage, and cholesterol content showed a time-dependent increase at the same time, which can demonstrate that cisplatin increases p53 mutations.

Fig. 1 Mechanistic validation of the existence of a vicious cycle between cisplatin and p53. **a** Kaplan-Meier plot of the correlation between the mutation of p53 and the survival of patients with NSCLC (Log-rank Mantel–Cox test). Cisplatin (Cis), fluvastatin sodium (Flu). **b** Outline of the three assays for evaluating *TP53* mutations, ROS levels, Cho levels and DNA damage before and after 30 days of cisplatin treatment (low dose, 10 μM; moderate dose, 25 μM; high dose, 50 μM). Heatmap of the number of mutations and RNA-seq analysis (**c**), ROS levels (**d**), Cho levels (**e**) and DNA damage (**f**) for the *wtp53*-expressing cells with cisplatin treatment for 30 days. **g** High-throughput sequencing before and after fluvastatin sodium (4

μM) treatment in A549 cells. **h** Genome-wide analysis of differential expression of fluvastatin sodium treated and untreated A549 cells after cisplatin induced. The volcano plot depicts the significance (false discovery rate adjusted p-value) and magnitude of difference (Fold Change). The dashed line indicates the threshold of the Fold Change > 2 and adjusted p-value <0.05. Some of the cancer related genes are labeled by dark colors. **i** GO enrichment analysis of differentially expressed genes (DEGs). The advanced bubble chart shows GO enrichment of DEGs in signaling pathways. The x-axis label represents the gene ratio, and the y-axis label represents GO terms. **j** GSEA shows how Flu affects the gene signatures of cholesterol metabolism. The normalized enrichment scores (NES) and p-values are indicated in each plot. **k** Heat map analysis of 17 mevalonate pathway genes and the SREBP2 gene from RNA-seq data using fluvastatin sodium treated and untreated A549 cells after cisplatin induced. The color scale indicates the expression fold change of genes. After Pifithrin-α and STI571 treatment for 24 h, the cytotoxicity and IC₅₀ of cisplatin treatment on in H1299 cells transfected with expression constructs containing the wtp53 (**l**), vector (**m**), and expression constructs containing the R273H variant (**n**) were measured by MTT method, and then the enzymatic activity of caspas3/9 was detected ($n = 3$ independent samples; two-tailed unpaired t test for **l**(i, ii), **m**(i, ii) and **n**(i, ii); one-way ANOVA followed by Tukey's HSD post hoc test for **l**(iv), **m**(iv) and **n**(iv)). **o** Dot plots showing the IC₅₀(Cis)/IC₅₀(Cis + Flu) in mutant and wild-type cells. **p** Schematic illustration of the treatment with cisplatin in mutp53 or wtp53 cells. Data were shown as the mean ± s.d. Source data are provided as a Source Data file.

References

1. Peña, Q. et al. Metallo drugs in cancer nanomedicine. *Chem. Soc. Rev.* **51**, 2544–2582 (2022).
2. Rinaldi, M., Cauchi, C., & Gridelli, C. First line chemotherapy in advanced or metastatic NSCLC. *Ann. Oncol.* **17**, v64–v67 (2006).
3. Pisano, C. et al. Antitumor activity of the combination of synthetic retinoid ST1926 and cisplatin in ovarian carcinoma models. *Ann. Oncol.* **18**, 1500–1505 (2007).
4. Cao, X., Hou, J., An, Q., Assaraf, Y. G., & Wang, X. Towards the overcoming of anticancer drug resistance mediated by p53 mutations. *Drug Resist. Updat.* **49**, 100671 (2020).
5. Chaft, J. E. et al. Evolution of systemic therapy for stages I-III non-metastatic non-small-cell lung cancer. *Nat. Rev. Clin. Oncol.* **18**, 547–557 (2021).
6. Sabharwal, S. S., & Schumacker, P. T. Mitochondrial ROS in cancer: initiators, amplifiers or an Achilles' heel? *Nat. Rev. Cancer* **14**, 709–721 (2014).
7. Hassin, O., & Oren, M. Drugging p53 in cancer: one protein, many targets. *Nat. Rev. Drug Discov.* **22**, 127–144 (2023).
8. Bagrodia, A. et al. Genetic determinants of cisplatin resistance in patients with advanced germ cell tumors. *J. Clin. Oncol.* **34**, 4000–4007 (2016).

2) The clinical relevance of the mutations in A549 cells is not clear and clinical evidence is needed. The biological significance of the cisplatin-induced p53 mutations in A549 cells is

overly simplified by showing the changes in cholesterol levels in the cells. The clinical relevance of this type experiments is not clear (Figs 2c, d, e). The mechanisms responsible for p53 mutation, cholesterol production/metabolism and cell cytotoxicity are not demonstrated. The authors made their conclusions based on correlation analysis, not mechanistic studies. In addition, the study design in mechanistic studies is flawed by switching cell line among different assays (Figs 2g-k).

Response: We strongly agree with reviewer's comments that further clinical evidence is needed to demonstrate the clinical relevance of A549 cells in the study of p53 mutations. Firstly, we would like to clarify that our study was conducted in vitro using A549 cells as a model system because of A549 cells are a classic selection of cells widely used in preclinical studies¹. In addition, we are based on studies of previous cisplatin-induced other mutations and combined with this result of cisplatin-induced A549 cell mutations (Fig. 1c). It is important to note that cell lines in vitro studies such as A549 cells are widely used in biomedical research as a means of investigating cellular mechanisms and potential therapeutic targets. These studies are typically followed by in vivo studies using animal models and ultimately clinical trials in human subjects². Our study represents an important first step in this process, and we believe that it provides valuable information that can guide future research. That being said, we agree that clinical evidence is necessary to fully establish the clinical relevance of mutations observed in A549 cells. The clinical evidence we have collected so far suggests that p53 mutations are being detected more frequently in cisplatin-resistant patients. This also demonstrates the clinical relevance of our demonstration in A549 cells that cisplatin can induce mutations.

Your valuable suggestions for the biological significance of cisplatin-induced p53 mutations in A549 cells are greatly appreciated. Based on the reviewer's comments, we redesigned this part of the experiments. It is clear that DNA damage and ROS generated by cisplatin lead to mutations, as well as the association of cisplatin with mutp53 have been reported³. In a previous study, it is concluded that cisplatin dose-dependently generates ROS and damages DNA, and induces p53 mutations^{4,5}. P53 mutations lead to upregulation of the mevalonate pathway, we examined cholesterol levels as shown in original Fig. 1d-e. Cisplatin exerts its antitumor effect mainly through DNA damage to further activate the p53 pathway and the c-Abl pathway. Next, the weighting of the p53 pathway and the c-Abl pathway in the antitumor effect of cisplatin was investigated in wtp53-expressing A549 cells and mutp53-expressing H1975 cells, respectively, and it was found that p53 mutations not only inactivate the p53 pathway but also lead to inhibition of the c-Abl pathway (original Fig. 1h-i and Supplementary Fig. 5). In our revised manuscript, we have enhanced the depth of this

section of the examination. First, we increased the number of cell types and focused on mutation sites when examining cisplatin-induced mutations (Fig. 1b). In addition, the p53 target genes were examined by RNA-seq analysis to see whether the function of p53 changed after cisplatin treatment. The results all showed that cisplatin induced p53 mutations in a concentration-dependent manner and affected its downstream function (Fig. 1c). We also tracked ROS levels, cholesterol levels, and DNA damage during the induction of mutations and found that they accumulated over time and dose (Fig. 1d-f).

In clinical terms, the mutation of p53 is an important indicator that cancer patients need to be measured, and Sanger sequencing is a classic method that has been widely used in the past⁶. Therefore, we also demonstrated the mutations caused by the action mechanism of cisplatin by the adoption of Sanger sequencing (original Fig. 2c) as well as by the cholesterol levels in vitro and in vivo (original Fig. 2d, e). In addition, it was reported that basic studies on p53 found that **cholesterol metabolism was closely related to p53**. Studies suggest that increased cholesterol levels in multiple cancers are caused by p53 mutations, by upgrading the Mevalonate pathway⁷. And cholesterol levels are also significantly associated with cancer risk, which is a factor that needs attention in patients⁸. The clinical significance of this part of the experiment is emphasized in our revised manuscript.

Thank you for the reply “the responsible mechanisms for p53 mutations, cholesterol production/metabolism and cell cytotoxicity are not demonstrated”. Statins are a class of drugs that specifically degrade mutp53. In the validation of the mechanism by which subsequent FP NPs can exert fluvastatin-specific degradation of mutp53, we did not elaborate on this mechanism, leading to an incomplete understanding. The mechanism of fluvastatin in the regulation of p53 mutations has been well studied (including why it can be specifically degraded, the pathway of action of specific degradation, cholesterol production/metabolism, and cell cytotoxicity), we have discussed the mechanism of fluvastatin degradation combined with the content of the experiment in our revised manuscript. In addition, A549 cells induced by different doses (Fig. 1c) and cells with or without fluvastatin treatment after induction at moderate doses (Fig. 1h-k) were also analyzed by RNA-seq in the revised manuscript, and the results showed that after induction by cisplatin, the p53 target genes was down-regulated with the increase in the number of p53 mutations (Fig. 1c); whereas, after fluvastatin treatment, the downstream of the p53 pathway underwent a certain degree of restoration after fluvastatin treatment (Fig. 1h-i). Moreover, RNA-seq results also showed that the cholesterol production pathway was up-regulated after cisplatin induction, while this pathway was inhibited after fluvastatin treatment (Fig. 1j-k). The cytotoxicity of p53 wild/null/mutation cells has also been further investigated (Fig. 1l-n).

Thank you for the reply “The authors made their conclusions based on correlation analysis, not mechanistic studies”. Based on the potential applicability of some evidence, we performed a number of analyses. Even though the mechanical validation experiments in original Fig. 2 can support our findings, we still need to conduct more in-depth mechanical research in order to provide additional mechanical evidence supporting our findings. In the revised manuscript, we enhanced the mechanistic studies in response to reviewer comments to include changes in p53 function after cisplatin treatment, which focused in detail on the primary target of action of fluvastatin to highlight the limitations in our choice of fluvastatin to reverse cisplatin treatment (Fig. 1).

Thank you for your comments “the study design in mechanistic studies is flawed by switching cell line among different assays (Fig. 2g-k)”. We are very sorry for this misunderstanding due to the lack of precision in the presentation. In the original version of Fig. 2g-k, we did not switch the cell lines arbitrarily, **all the experiments were done in A549 cells**, and in order to compare the difference between wtp53 and mutp53, H1975 and A549/DDP were used as controls. Whereas the original Fig. 2k is a survival analysis based on whether p53 is mutated in the clinical database, not involving cell lines. We strengthened in the revised manuscript the selected samples in the description of the selected samples in our revised manuscript to make the conclusions we obtained and the experimental design more understandable. In addition, to further rule out differences between cells, we also replicated the p53 pathway and the c-Abl pathway affected by the p53 mutations **in H1299 cells transfected with vector, expression constructs containing the wtp53 and expression constructs containing the R273H variant (Fig. 1l-n).**

References

1. Li, J. Y. et al. Preclinical PK/PD model for combined administration of erlotinib and sunitinib in the treatment of A549 human NSCLC xenograft mice. *Acta Pharmacol. Sin.* **37**, 930–40 (2016).
2. Chargari, C. et al. Preclinical assessment of JNJ-26854165 (Serdemetan), a novel tryptamine compound with radiosensitizing activity in vitro and in tumor xenografts. *Cancer Lett.* **312**, 209–218 (2011).
3. Pisano, C. et al. Antitumor activity of the combination of synthetic retinoid ST1926 and cisplatin in ovarian carcinoma models. *Ann. Oncol.* **18**, 1500–1505 (2007).
4. Sabharwal, S. S., & Schumacker, P. T. Mitochondrial ROS in cancer: initiators, amplifiers or an Achilles’ heel? *Nat. Rev. Cancer* **14**, 709–721 (2014).
5. Cuella-Martin, R. et al. Functional interrogation of DNA damage response variants with base editing screens. *Cell* **184**, 1081–1097 (2021).

6. Hortal, A. M. et al. Overexpression of wild type RRAS2, without oncogenic mutations, drives chronic lymphocytic leukemia. *Mol. Cancer* **21**, 35 (2022).
7. Moon, S. H. et al. p53 Represses the Mevalonate Pathway to Mediate Tumor Suppression. *Cell* **176**, 564–580 (2019).
8. Huang, B. et al. Cholesterol metabolism in cancer: mechanisms and therapeutic opportunities. *Nat. Metab.* **2**, 132–141 (2020).

3) The selection of Fluvastatin is not well justified.

Response: Thank you very much for your valuable comment. Based on the reviewer's suggestions, we examined the following aspects. First, we performed RNA-seq on fluvastatin treated and untreated cells according to Fig. 1g. The results showed that the expression of p53 downstream genes as well as GO-enriched p53 downstream pathways were significantly changed in p53 mutant cells induced by moderate dose cisplatin, and that these changes would reflect the "GOF" effect. In the fluvastatin treated cells, these "GOF" effects were significantly reduced: the expression of genes and pathways related to chemoresistance, cell migration, and cell invasion were significantly reduced, which demonstrated the potential of fluvastatin to address the use of cisplatin in non-small cell lung cancer (Fig. 1h-i). In addition, our experimental results directly labeled the accumulation of intracellular cholesterol concentration after cisplatin treatment (Fig. 1e), which is consistent with previous studies that mutant p53 upregulates the mevalonate pathway. And the RNA-seq results of fluvastatin-treated and untreated cells can also conclude that the cholesterol production pathway is upregulated in cisplatin treated cells, and this can be downregulated by fluvastatin as cholesterol inhibitor (Fig. 1j-k), which can also be an important basis for fluvastatin being selected. Moreover, our results in Figure 1l-n also showed that the inhibition of apoptosis after the inhibition of p53 pathway and the c-Abl pathway could be restored by fluvastatin treatment, regardless of in p53 null, wild, and mutated cells. Finally, we also examined the IC₅₀ values of cisplatin with and without fluvastatin co-treatment in a large number of lung cancer cell lines, and showed that fluvastatin had the greatest synergistic effect on cisplatin in the conformationally mutated p53 cell lines (Fig. 1o). The results of these experiments in which fluvastatin was selected were revised in the manuscript based on the reviewer's comments.

4) The purity of FP NPs was not clearly demonstrated.

Response: Thank you very much for your kind comment. In response to the reviewer's comments, we have added to the revised manuscript the purity of the prodrug we synthesized

was 94.26%, while the encapsulation rate of the final formulation of FP NPs was $96.15 \pm 1.98\%$.

5) Fig. 4: The relative values of IC₅₀s (the ratios of cisplatin vs FP NPs) among the three cell lines are similar, indicating that there are no specific effects on drug resistant cells and there is no direct relation to p53 mutation.

Response: Thank you very much for your comment on the relative values of IC₅₀ (the ratios of cisplatin vs FP NPs) among the three cell lines in Fig. 4 (Fig. 3 of revised manuscript). We very much apologize due to the fact that we did not discuss the FI values in detail in the original manuscript. First, we need to clarify that **the FI values of these three cells are not similar, the ratios of cisplatin vs FP NPs among the three cell lines are 25.93 ± 4.33 (H1975), 2.85 ± 0.31 (A549), 25.12 ± 4.98 (A549/DDP).** It can be seen that the FI values of H1975 and A549/DDP, two cells with p53 mutations, are much larger than those of wild-type A549 cells (p53 mutations in A549/DDP was verified in Supplementary Fig. 25). Second, the FI values only indicate the fold change in IC₅₀ values of FP NPs relative to cisplatin, and the IC₅₀ values of FP NPs as well as the cell survival curves of each group should also be discussed together. **The similar IC₅₀ values** of the FP NPs in the three cells also corroborate the innovative mechanism of FP NPs, degradation of mutp53 while also achieving maximum antitumor effect through ERS that is not limited by the p53 mutations. Moreover, our results in Figure 3e-g, in cells expressing p53 wild/null/mutated, respectively, there was also little change in IC₅₀ values after inhibition of p53 or c-Abl, also suggesting that ERS reversed the restriction of p53 mutation or not. Finally, we also studied the FI values in more lung cancer cells in our revised manuscript, as shown in Fig. 3h, with slopes as FI values, and the conclusions are the same as the previous ones in the three cells: having higher FI values in mutant cells (especially p53 conformational mutations) proves that FP NPs resolve the therapeutic bottleneck of cisplatin to a greater extent in mutant cells, and break the limitation of p53 mutations. This section we have added in the revised manuscript.

Fig. 3 Antitumor activity of FP NPs in vitro. **a** Cytotoxicity and IC₅₀ of various formulations as determined by the MTT assay in H1975, A549, and A549/DDP cells after 24 h incubation ($n = 3$ independent samples; one-way ANOVA followed by Tukey's HSD post hoc test). FI (fold increase) is defined as IC₅₀ (Cis)/IC₅₀ (FP NPs) **b** Inverted microscopy images of H1975, A549, and A549/DDP cells calcein AM/PI double staining after 6 h treatment with 4 μM different formulations (dosage based on Fluplatin). Scale bars, 100 μm. **c** Apoptosis rate of H1975 cells after 6 h of 4 μM different formulations (dosage based on Fluplatin) incubation was determined by flow cytometry (FACS) analysis. **d** The fold rate of caspase3/9 enzymatic activities was detected after 6 h treatment with 4 μM different formulations (dosage based on Fluplatin; $n = 3$ independent samples; one-way ANOVA followed by Tukey's HSD post hoc test). After Pifithrin-α and STI571 treatment for 24 h, the

cytotoxicity and IC_{50} of cisplatin treatment on in H1299 cells transfected expression constructs containing the wtp53 (e), with vector (f), and expression constructs containing the R273H variant (g) were measured by MTT method, two-tailed unpaired t test for e, f, and g. h Dot plots showing the IC_{50} (Cis)/ IC_{50} (FP NPs) in mutant and wild-type cells. Data were shown as the mean \pm s.d. Source data are provided as a Source Data file.

6) Fig. 5: No direct evidence for FP NPs' specific effects on p53 mutations.

Response: Thank you very much for the valuable comments. According to the reviewer's suggestion, we have performed more experiments in the revised manuscript. Firstly, we investigated the ability of FP NPs to specifically degrade mutp53 and the mechanism of FP NPs specific degradation by Western Blot in several cell lines; secondly, we investigated the changes of downstream genes affected by the "GOF" of mutp53 by RNA-seq. We directly examined the levels of p53 in mutant and wild-type cells by Western Blot, the results are shown in Fig. 5a. In H1975, FP NPs could most effectively degrade mutp53 specifically, while in A549 it still exerted the apoptogenic effect of wtp53. According to previous studies, statins act as inhibitors of HMGR by modulating the mevalonate pathway and thus further degrading mutp53. Therefore, we demonstrated this specific degradation mechanism by assaying the enzymatic activity of HMGR (Fig. 5r). According to the reviewer's suggestion, we have added more experiments to increase the convincing power. First, we increased the number of cells to determine the different mechanisms of FP NPs in p53 mutant and wild-type cells. In multiple mutant cells, it was seen that FP NPs had the strongest effect on degrading conformationally mutated p53; whereas in wild-type cells, FP NPs up-regulated wtp53 (Fig. 5a-e, i-j). This course of action showed a dose-dependent effect in both mutant and wild-type cells (Fig. 5f-h). Next, we verified by immunoblotting that the effect of FP NPs on mutp53 is mediated by ubiquitination rather than autophagy, whereas wtp53 is not affected by the inhibitor and its expression rises upon treatment with FP NPs (Fig. 5k-l). Moreover, we determined the half-life of mutp53 by cycloheximide (CHX) after treatment with FP NPs. The results showed that FP NPs significantly decreased the half-life of mutp53 within 16 h; whereas wtp53 itself had a short half-life and did not change significantly after FP NPs treatment (Fig. 5m-p). Finally, we verified again by immunoprecipitation that FP NPs are mediated by ubiquitination to degrade mutp53 and do not degrade wtp53 (Fig. 5q). According to the reviewer's comments all these contents we have updated and discussed in the revised manuscript to directly demonstrate the specificity of FP NPs for p53 mutations.

Fig. 5 Mechanism of p53-related antitumor activity of FP NPs in vitro. **a** Western blotting analysis of p53 in H1975, H2087, H2342 and A549 cells after treatment with 4 μ M of different formulations (dosage based on Fluplatin) for 12 h. And their grayscale values were

quantified (**b-e**) ($n = 3$ independent samples; one-way ANOVA followed by Tukey's HSD post hoc test). **f** Western blotting analysis of p53 in H1975 and A549 cells after treatment with different concentrations of FP NPs for 12 h. And their grayscale values were quantified (**g-h**) ($n = 3$ independent samples; one-way ANOVA followed by Tukey's HSD post hoc test). **i** Western blotting analysis of p53 in H1793, H2196, H748, H1770, H1581, and H520 cells after treatment with 4 μ M of FP NPs for 12 h. And their grayscale values were quantified (**j**) ($n = 3$ independent samples; one-way ANOVA followed by Tukey's HSD post hoc test). **k** Western blot of H1975, H2087, H2342, and A549 cells treated with 4 μ M FP NPs or not for 12 h and cultured in medium containing 10 μ M MG132 or 10 mM 3-MA. And their grayscale values were quantified (**l**) ($n = 3$ independent samples; one-way ANOVA followed by Tukey's HSD post hoc test). Western blot of H1975 (**m**), H2087 (**n**), H2342 (**o**), and A549 (**p**) cells treated with 2 μ M FP NPs or not for 12 h and then treated with 100 μ g/mL CHX at the indicated time points. $n = 3$ biological independent samples, relative p53/actin ratios are shown. **q** Western blotting analysis of total ubiquitination (Ub) of the immunoprecipitated (IP) p53 in H1975 and A549 cells after the 2 μ M FP NPs treatment with out or with MG132. **r** HMGR inhibition assay under different concentration of fluvastatin sodium, Fluplatin, FP NPs, respectively ($n = 3$ independent samples; one-way ANOVA followed by Tukey's HSD post hoc test). **s** Genome-wide analysis of differential expression of FP NPs treated and untreated H1975 cells. **t** Schematic summary of the p53-related antitumor mechanism of FP NPs. Source data are provided as a Source Data file.

7) Figs. 6,7, and 8: Only one mutated cell line is used. P53 wt and p53 mt and drug resistant cells should be used to demonstrate the specific effects of FP NPs on reversing cisplatin drug resistance.

Response: Thank you for the kind suggestion. We believe that it is necessary to establish mutant, wild-type and drug-resistant cell lines in vivo to verify the specific effect of FP NPs on cisplatin-induced p53 resistance. According to the reviewer's suggestion, we have also complemented this part of the experiment to better confirm our conclusions in revised manuscript.

Supplementary Fig. 42 Antitumor efficacy of intravenously injected FP NPs in A549 and A549/DDP tumor-bearing mice model. a, h Schematic illustration of the experimental design. **b, I** Images of tumor and tumor weight/ volume on day 19 of each group ($n = 5$ mice per group; one-way ANOVA followed by Tukey's HSD post hoc test). **c, j** Tumor volume changes for each mouse in each group over 21 days of treatment ($n = 5$ mice per group). **d, k** The tumor volumes of mice during treatments with different formulations ($n = 5$ mice per group; two-way ANOVA followed by Tukey's multiple comparisons post test). **e, l** The body weight of mice during treatments with different formulations ($n = 5$ mice per group; two-way ANOVA followed by Tukey's multiple comparisons post test). Data were shown as the mean \pm s.d. Source data are provided as a Source Data file. **f, m** Kaplan-Meier survival curve of

mice treated with different formulations over 70 days ($n = 8$ mice per group; Log-rank Mantel–Cox test). **g, n** The body weight of mice during treatments with different formulations over 70 days ($n = 8$ mice per group; two-way ANOVA followed by Tukey’s multiple comparisons post test).

Fig. 9 The tumor recurrence and metastasis inhibition of FP NPs in H1975-luc tumor-bearing mice. **a** Schematic illustration of the experimental design. **b** The tumor volumes of mice during treatments with each group ($n = 7$ mice per group; two-way ANOVA followed by Tukey’s multiple comparisons post test). **c** The body weight of mice during

treatments with each group ($n = 7$ mice per group; two-way ANOVA followed by Tukey's multiple comparisons post test). **d** Tumor volume changes for each mouse in each group ($n = 7$ mice per group). **e** In vivo bioluminescence imaging of tumor-bearing mice receiving various treatments after surgery. Three representative mice in each treatment group were shown. Images of day 0 were taken on the day of surgery. **f** The final anatomical picture and the H&E staining of the lungs. **g** IHC of p53, E-cadherin, MMP-2, MMP-9, Vimentin in the primary tumor tissues of cisplatin or FP NPs treatment, recurrent and metastases tumor tissue of cisplatin treatment. Scale bars, 20 μm . **h** The recurrence tumor weights of different groups on day 42 after surgery ($n = 7$ mice per group; one-way ANOVA followed by Tukey's HSD post hoc test). **i** Kaplan-Meier survival curve of mice treated with each group over 65 days ($n = 7$ mice per group; Log-rank Mantel-Cox test). **j** Final weights of the heart, liver, spleen, lungs, kidneys ($n = 7$ mice per group; two-tailed unpaired t test). **k** Heatmap of TNF- α , VEGF, IL-10 and IL-6 expression profiles in serum ($n = 3$ mice per group). **l** Indicators of blood routine examination ($n = 3$ mice per group; two-tailed unpaired t test) of mice. Data were shown as the mean \pm s.d. Source data are provided as a Source Data file.

Fig. 10 The tumor recurrence and metastasis inhibition of FP NPs in A549-luc and

A549/DDP-luc tumor-bearing mice. a, l Schematic illustration of the experimental design. **b, m** The tumor volumes of mice during treatments with each group ($n = 7$ mice per group; two-way ANOVA followed by Tukey's multiple comparisons post test). **c, n** The body weight of mice during treatments with each group ($n = 7$ mice per group; two-way ANOVA followed by Tukey's multiple comparisons post test). **d, o** Tumor volume changes for each mouse in each group ($n = 7$ mice per group). **e, p** In vivo bioluminescence imaging of tumor-bearing mice receiving various treatments after surgery. Three representative mice in each treatment group were shown. Images of day 0 were taken on the day of surgery. **f, q** The final anatomical picture and the H&E staining of the lungs. **g, r** Final weights of the heart, liver, spleen, lungs, kidneys ($n = 7$ mice per group; two-tailed unpaired t test). **h, s** The recurrence tumor weights of different groups on day 42 after surgery ($n = 7$ mice per group; one-way ANOVA followed by Tukey's HSD post hoc test). **i, t** Kaplan-Meier survival curve of mice treated with each group over 65 days ($n = 7$ mice per group; Log-rank Mantel-Cox test). **j, u** Heatmap of TNF- α , VEGF, IL-10 and IL-6 expression profiles in serum ($n = 3$ mice per group). **k, v** Indicators of blood routine examination ($n = 3$ mice per group; two-tailed unpaired t test) of mice. Data were shown as the mean \pm s.d. Source data are provided as a Source Data file.

8) This manuscript is not well prepared and it's hard to follow. There are many typos and grammatic errors in the entire manuscript (e.g., line 42). The lack of logical reasoning and the lack of logical information flow in writing are of concern. For example, it is not clear what kind of negative effect of cisplatin on TP53 mutations is referred to (line 46-47). Another example, the figures are not cited in the text in a logical order (e.g., Figure 1c was cited first and before Fig 1 a and b in the text, suppl Fig 16 was cited before suppl Fig. 8).

Response: Thank you for the suggestion. Based on the recommendations of the reviewers, we have improved logic and readability in revised manuscript. And also, the manuscript was carefully revised before submission by Nature Research Editing Service to improve the grammar and readability.

December 08, 2022

Dear Hulin Jiang,

Thank you for choosing Springer Nature Author Services. This manuscript describing a novel nanoparticle-based NSCLC treatment that overcomes cisplatin resistance and prevents tumor recurrence is very interesting. The paper was edited for grammar, phrasing, and punctuation. In addition, many edits were made to further improve the flow and readability of the text. Below, we highlight the areas of this paper that we focused on in our edit.

Certain edits were made to remove redundant, repetitive or unnecessary phrasing and to present the information in a more straightforward manner.

Articles are an important aspect of the English language, including the definite article "the" and the indefinite articles "a" and "an." Our edits focused on improving article use, which is often strongly dependent on context and field conventions.

In cases where the meaning of the text was not clear, revisions were made to convey the information with increased clarity and reduced ambiguity.

Comments were left if further clarification would be helpful or confirmation of the meaning of the text was necessary. Please review these comments and all our changes carefully for more detailed suggestions, as well as to ensure that the final version of the manuscript is fully accurate.

Thank you again for using our editing services; we wish you the best of luck with your submission.

Best regards,

Nancy V.
Senior Editor
Springer Nature Author Services

We have adjusted the order of figures to make them more logically understandable in revised manuscript.

9) The methods section is not well referenced. Sex as a biological variable is not considered in the animal study design.

Response: Thank you for this thoughtful comment. We are sorry that we did not state directly in the method the reasons for the choice of models for animal experiments. In our study of NSCLC, female BALB/c nude mice were selected for in vivo studies in many of the studies we referred to, and **in order to avoid the biases associated with sex differences** in immune system and tumor growth, **only female mice were used**¹. We have improved the methods section to address such issues and have added gender-specific groups to the supplementary information to exclude this variable.

Supplementary Fig. 38 Antitumor efficacy of intravenously injected FP NPs in H1975 tumor-bearing mice model of different genders.

References

1. Limagne, E. et al. MEK inhibition overcomes chemoimmunotherapy resistance by inducing CXCL10 in cancer cells. *Cancer Cell* **40**,136–152 (2022).

10) The discussion section is not well prepared; the mechanism of action, the clinical relevance of the models used and the results obtained are not well discussed. The presented results do not support the authors' conclusions.

Response: Thank you very much for this comment. We are very sorry that we have not presented our advantages in the discussion better than previous research, neglecting a detailed examination of the new structure of our synthesis and the mechanisms of action of its preparations. Based on the reviewer's suggestions, these issues have been added and enhanced respectively in the revised manuscript.

Thank you very much!

Reviewer #3 - Drug delivery, nanotherapy - (Remarks to the Author):

Authors in their manuscript entitled “Mutant-p53-targeting-therapeutic nanoparticles resolves chemoresistance and tumor recurrence in non-small cell lung cancer” have prepared a prodrug of fluvastatin (inhibitor for mutp53) and cisplatin and encapsulated in self-assembling nanoparticles. These nanoparticles were tested in different animal models and shown to inhibit the tumor growth of NSCLC promisingly. The idea is that nanoparticles enter tumors via passive delivery and taken up by tumor cells and inhibit mutp53 while cisplatin induces apoptosis more effectively. The study has been well thought and experiments are performed in detail. The anti-tumor effects looks attractive, but the study lacks novelty in terms of chosen drugs and the concept of combination.

Response: Thank you very much for recognition of our work. Please find our responses to the comments point-by-point below.

1) Fluvastatin is proposed as a drug to degrade mutp53 in this study. However, this drugs is not specific for mutp53 but has many other off-target effects which are responsible for inducing apoptosis.

Response: Thank you for your precious comments. We are sorry for our inadequate description of statins, which led to ambiguity. Fluvastatin is specific to mup53 and only degrades mutp53¹. The off-target effect about leading to apoptosis is a side effect of statins as a traditional lipid-lowering drug². On the one hand, we make the drug primarily act on tumor cells through nano-delivery. On the other hand, this side effect is greatly reduced in FP NPs due to our final dosage being lower than that of statin as a lipid-lowering drug. These descriptions of fluvastatin are added in the revised manuscript based on the reviewer’s comments.

References

1. Ingallina, E. et al. Mechanical cues control mutant p53 stability through a mevalonate-RhoA axis. *Nat. Cell Biol.* **20**, 28–35 (2018).
2. Huang, Y. et al. A framework for identification of on- and off-target transcriptional responses to drug treatment. *Sci. Rep.* **9**, 7603 (2019).

2) Previous studies have reported the combination of fluvastatin and cisplatin (<https://pubmed.ncbi.nlm.nih.gov/20837358/>) which shows lack of new idea of using this combination. Fluvastatin was withdrawn from the marker and therefore future translation is questionable.

Response: Thank you for your comments. The article you mentioned about combining fluvastatin with cisplatin, which we have noticed in the past, does not affect our creativity in using this combination. In our study, firstly, we are addressing the bottleneck of cisplatin in the treatment of NSCLC from an emerging issue. Although many studies have been devoted to structural modification of cisplatin to diminish drug resistance, few have approached it from the perspective of p53 mutations, which is a major contributor to cisplatin chemoresistance¹⁻². Second, the mechanism of targeting mutp53 and breaking the mutp53 restriction of the FP NPs we designed is emerging. We synthesized a novel small molecule, Fluplatin, and Fluplatin self-assembled nanoparticles encapsulated in PEG-PE in only one step. This design achieves ER targeting. Fluplatin, which eventually reaches the endoplasmic reticulum, can target mutp53 degradation, upregulate wtp53, activate potent ERS and eliminate “GOF”, and ultimately achieve excellent antitumor effects. Finally, we also examined the ability of FP NPs to inhibit relapse, and the strongest relapse inhibitory ability was highlighted in mutant/wild/resistant models.

According to the latest developments, the FDA has not reported any withdrawal of fluvastatin. The FDA recently reported some New Potential side effects, but they are still assessing the need for regulatory action (January - March 2022 | Potential Signals of Serious Risks/New Safety Information Identified by the FDA Adverse Event Reporting System (FAERS) | FDA). Only cilastatin was withdrawn from the market in 2001 (<https://www.drugs.com/baycol.html>). Lovastatin, pravastatin, simvastatin, atorvastatin, and **fluvastatin** are **safe**. Rhabdomyolysis occurs only in rare cases and the health benefits outweigh any risks. We need to reinforce such relevant descriptions in the discussion to reduce the misconception that fluvastatin is associated with clinical risk. In addition, cisplatin and fluvastatin are both widely used drugs in the clinic, and it is also very clinically significant that we constructed this therapy to greatly enhance their antitumor effects alone or in physical mix.

References

1. Peña, Q. et al. Metallodrugs in cancer nanomedicine. *Chem. Soc. Rev.* **51**, 2544–2582 (2022).
2. Rinaldi, M., Cauchi, C., & Gridelli, C. First line chemotherapy in advanced or metastatic NSCLC. *Ann. Oncol.* **17**, v64–v67 (2006).

3) Technically, the study has many shortcomings. for examples

1. stability studies in Fig 3e were only performed in buffers but not in plasma. There was a substantial release of cis-pt at pH7.4.

Response: Thank you for your kind comment in Fig 3e (Fig. 2 of the revised manuscript). We have added the release experiments in plasma in Fig. 2e. With regard to your reference to “substantial release of cis-pt at pH 7.4”, we apologize that it was due to our inadvertent omission of labeling our release medium as PBS solution containing 1% Tween (accelerated release speed), which we have corrected in the revised manuscript.

Fig. 2 Design and characterization of FP NPs. a CI value in different ratios of physical

mixture between cisplatin and fluvastatin sodium, and the chemical structure and synthesis route of Fluplatin. **b** Particle size, zeta potential, PDI, and TEM image of the Fluplatin self-assembled nanoparticles. Scale bars, 50 nm. **c** Particle size, PDI, and encapsulation efficiency of nanoparticles formed by different proportions of Fluplatin and PEG-PE ($n = 3$ independent samples). **d** Particle size, zeta potential, PDI, and TEM image of the FP NPs. Scale bars, 200 nm. **e** Release profile of the Fluplatin self-assembled nanoparticles (i) and FP NPs (ii) in phosphate solution of 10% plasma at pH 7.4. **f** Release profile of the Fluplatin self-assembled nanoparticles (i) and FP NPs (ii) in phosphate solution of 1% Tween 80 at pH 7.4, pH 6.5, and pH 4.5 ($n = 3$ independent samples). **g** Mechanism for the Fluplatin self-assembled nanoparticles (i) and FP NPs (ii) ($n = 3$ independent samples). **h** SEM image and EDS element mapping of FP NPs. Scale bars, 3 μm . **i** Confocal images of Dil@FP NPs in H1975 cells after treatment with 2 μM Dil@FP NPs for 1 h, 2 h, 4 h, 8 h, respectively. Scale bars, 20 μm . **j** Uptake efficiency of 2 μM FP NPs treatment for 4 h in H1975 cells in the presence of various endocytosis inhibitors was detected by ICP-MS ($n = 3$ independent samples; one-way ANOVA followed by Tukey's HSD post hoc test). **k** Confocal images of Dil@FP NPs and PEG-PE in H1975 cells after treatment with 2 μM Dil@FP NPs for 1 h, 2 h, 4 h, respectively. Scale bars, 5 μm . **l** Uptake Pt in H1975 cells after treatment with 2 μM different formulations (dosage based on Fluplatin) for 1 h, 2 h, 4 h, 8 h detected by ICP-MS, respectively ($n = 3$ independent samples; one-way ANOVA followed by Tukey's HSD post hoc test). **m** Uptake Pt in nuclear, lysosome, Golgi apparatus, endoplasmic reticulum, mitochondria after 8 h treatment of 2 μM different formulations (dosage based on Fluplatin; $n = 3$ independent samples; one-way ANOVA followed by Tukey's HSD post hoc test). **n** Major intracellular distribution of FP NPs. Data were shown as the mean \pm s.d. Source data are provided as a Source Data file.

2. No nanoparticles were made without prodrug proving the additional value of the prodrug in nanoparticles. This is important to show that the prodrug as NPs has advantage over the simple encapsulation of these two drugs in polymeric NPs.

Response: Thank you for your comment. We strongly agreed with the need to add a simple encapsulation of these two drugs as a control group, so we added the F+C NPs group to the revised manuscript and also added the advantage of synthesizing the prodrug and wrapping it around PEG-PE in the revised manuscript as per the reviewer's comments.

3. In fig 4, it is not clear what do they mean with 4uM different formulations. Fluplatin includes 2 molecules of Fluvastatin which means there could be 2x drug compared to free fluvastatin and this could lead to higher effect.

Response: Thank you for your comment in Fig 4 (Fig. 3 of the revised manuscript). Four μM was based on the final concentration of Fluplatin, and the corresponding fluvastatin and fluvastatin in the Flu + Cis group was 8 μM (2x drug). We have corrected all descriptions to avoid ambiguity.

Fig. 3 Antitumor activity of FP NPs in vitro. **a** Cytotoxicity and IC_{50} of various formulations as determined by the MTT assay in H1975, A549, and A549/DDP cells after 24 h incubation ($n = 3$ independent samples; one-way ANOVA followed by Tukey's HSD post hoc test). FI (fold increase) is defined as $IC_{50}(\text{Cis})/IC_{50}(\text{FP NPs})$ **b** Inverted microscopy images of H1975, A549, and A549/DDP cells calcein AM/PI double staining after 6 h treatment with $4 \mu\text{M}$ different formulations (dosage based on Fluplatin). Scale bars, $100 \mu\text{m}$. **c** Apoptosis rate of H1975 cells after 6 h of $4 \mu\text{M}$ different formulations (dosage based on Fluplatin) incubation was determined by flow cytometry (FACS) analysis. **d** The fold rate of caspase3/9 enzymatic activities was detected after 6 h treatment with $4 \mu\text{M}$ different formulations (dosage based on Fluplatin; $n = 3$ independent samples; one-way ANOVA followed by Tukey's HSD post hoc test). After Pifithrin- α and STI571 treatment for 24 h, the

cytotoxicity and IC₅₀ of cisplatin treatment on in H1299 cells transfected expression constructs containing the wtp53 (e), with vector (f), and expression constructs containing the R273H variant (g) were measured by MTT method, two-tailed unpaired t test for e, f, and g. h Dot plots showing the IC₅₀ (Cis)/IC₅₀ (FP NPs) in mutant and wild-type cells. Data were shown as the mean ± s.d. Source data are provided as a Source Data file.

4. In vivo: The used animal models are not optimal. They used subcutaneous tumor for NSCLC and orthotropic tumor model should be used as these tumors lack the microenvironment.

Response: Thank you for your comment, the orthotropic tumor model would be more representative of the clinical situation. We apologize for not being able to add the orthotropic tumor model in the original study due to considerations of the number of animals used and the observability of the experiment. Usually, the subcutaneous tumor model is widely used to measure drug efficacy, comparison of preclinical model efficacy with Phase II clinical trial results supports the predictive value of cell lines^{1,2}. Therefore, we retained the section on subcutaneous tumors in the revised manuscript. Moreover, we have supplemented the experiment with what you refer to as the advantageous in orthotropic tumor model in revised manuscript.

Fig. 8 Antitumor efficacy of intravenously injected FP NPs in H1975-luc and A549-luc orthotopic lung tumors. a, g Schematic illustration of the experimental design. **b, h** The

progression of orthotopic lung tumors ($n = 5$ mice per group) in BALB/c nude mice. A representative bioluminescent image from each group was shown. **c, i** The final anatomical picture and the H&E staining of the lungs. **d, j** Average tumor nodule. Tumor nodules of 2-10 mm^3 in volume were counted using harvested lungs from the control and treated groups, and the average number of tumor nodules was determined. Each dot represents a tumor from an individual mouse. The tumor nodules in the lungs were indicated by arrows (one-way ANOVA followed by Tukey's HSD post hoc test). **e, k** Quantitative bioluminescence analysis in mice bearing orthotopic lung tumors; P/S = photons/second (two-way ANOVA followed by Tukey's multiple comparisons post test). **f, l** Mouse body weight in orthotopic lung tumor mouse model (two-way ANOVA followed by Tukey's multiple comparisons post test). Data were shown as the mean \pm s.d. Source data are provided as a Source Data file.

Supplementary Fig. 43 Antitumor efficacy of intravenously injected FP NPs in A549/DDP-luc orthotopic lung tumors. **a** Schematic illustration of the experimental design. **b** The progression orthotopic lung tumors ($n = 5$ mice per group) in BALB/c nude mice. A representative bioluminescent image from each group was shown. **c** The final anatomical picture and the H&E staining of the lungs. **d** Average tumor nodule. Tumor nodules of 2-10 mm^3 in volume were counted using harvested lungs from the control and treated groups, and the average number of tumor nodules was determined. Each dot represents a tumor from an individual mouse. The tumor nodules in the lungs were indicated by arrows (one-way ANOVA followed by Tukey's HSD post hoc test). **e** Quantitative bioluminescence analysis in mice bearing orthotopic lung tumors; P/S = photons/second (two-way ANOVA followed by Tukey's multiple comparisons post test). **f** Mouse body weight in orthotopic lung tumor mouse model (two-way ANOVA followed by Tukey's multiple comparisons post test). Data were shown as the mean \pm s.d. Source data are provided as a Source Data file.

References

1. Qiu, Q. et al. Drug-Polymer Hybrid Macromolecular Engineering: Degradable PEG Integrated by Platinum(IV) for Cancer Therapy. *Matter* **1**, 1618–1630 (2019).
2. Wang, L. et al. Systematic Strategy of Combinational Blow for Overcoming Cascade Drug Resistance via NIR-Light-Triggered Hyperthermia. *Adv. Mater.* **33**, 2100599 (2021).

5. Also, tumor were allowed to grow to the size of about 3000 mm³ which are ethically not acceptable mostly nowadays.

Response: Thank you for your comment. We are sorry that we overlooked the ethical end point of the institutional review in the methods section. Our institution based its ethics on the Guide to Euthanasia of Laboratory Animals (Standard of the Chinese Association for Laboratory Animal Sciences), stipulate a maximum tumor diameter not exceeding 20 mm as the experimental end point. Based on the reviewer's comments, a description of the ethical requirements was added to our revised manuscript. In addition, the dosing time was scaled down to control the maximum tumor volume in all in vivo experiments that were re-completed according to the reviewer's comments.

6. One major problem is the lack of control NPs including both Flu + cis in nanoparticles. Why other control Fluplatin without NP not included in vivo?

Response: Thank you for your kind comment. Based on the recommendations of the reviewers, we have supplemented this control group in revised manuscript (F+C NPs) as shown in Fig. 6-10 and Supplementary Fig. 42-43. Since Fluplatin is a hydrophobic agent in vivo, there may be safety concerns if administered intravenously as in other groups. Therefore, we have also supplemented the control of Fluplatin self-assembled nanoparticles without PEG-PE packaging (F NPs) as shown in Fig. 6-10 and Supplementary Fig. 42-43.

7. Fig 7 misses quantitative analysis of staining and Western blot.

Response: Thank you for your kind comment. We will add In response to the reviewer's comments, we have added quantitative analysis with staining and Western blot to the revised manuscript.

Fig. 7 Biodistribution and anticancer mechanism of FP NPs in H1975 tumor-bearing mice model. **a** Ex vivo biodistribution imaging of the main organs in mice bearing H1975 xenografts of the DiR@FP NPs at various time points post injection. **b** ICP-MS measurement of Pt accumulation in individual organs at 1, 6, 12 and 24 h after treatment with cisplatin (i) or FP NPs (ii) ($n = 3$ mice per group). **c** ICP-MS measurement of Pt accumulation in the tumor tissues at 1, 6, 12 and 24 h after treatment with cisplatin or FP NPs ($n = 3$ mice per group; one-way ANOVA followed by Tukey's HSD post hoc test). **d** TC levels in tumor tissues and TC, TG, HDL, LDL levels in serum after treatment with each group ($n = 3$ mice per group; one-way ANOVA followed by Tukey's HSD post hoc test). Confocal images of ATF4 (**e**) and γ -H2AX (**f**) in the tumor tissues of mice after treatment with each group. Scale bars, 100 μ m. **g** Western blotting analysis of p53^{R273H} in tumor tissues after treatment with each group. And their grayscale values were quantified (**h**) ($n = 3$ independent samples; one-way ANOVA

followed by Tukey’s HSD post hoc test). **i** IHC of p53 in the tumor tissues after treatment with each group. Scale bars, 20 μm. **j** Kaplan-Meier survival curve of mice treated with each group over 70 days ($n = 8$ mice per group; Log-rank Mantel–Cox test). **k** The body weight of mice during treatments with each group over 70 days ($n = 8$ mice per group; two-way ANOVA followed by Tukey’s multiple comparisons post test). **l** IHC of Ki67 in the tumor tissues after treatment with each group. Scale bars, 20 μm. Data were shown as the mean ± s.d. Source data are provided as a Source Data file.

Supplementary Fig. 41 **a** Quantitative of ATF4 fluoresce intensity after treatment with different formulations. **b** Quantitative of γ-H2AX fluoresce intensity after treatment with different formulations. **c** Quantitative of p53 positive cells after treatment with different formulations. **d** Quantitative of ki67 positive cells after treatment with different formulations.

Thank you very much!

REVIEWER COMMENTS

Reviewer #1 - Original (Remarks to the Author):

In the revised manuscript, the authors conducted research based on the possible problems in cisplatin treatment, and discussed the biological process of cisplatin and the expression level of the p53 target genes, which I think has certain research significance for attacking cisplatin resistance. In addition, a profound mechanistic study was conducted in the selection of fluvastatin, which elucidated the deficiencies in cisplatin treatment and the biological significance of the selection of fluvastatin. The cytotoxic mechanism of cisplatin in cells expressing wild-type p53 and cells expressing mutant p53, respectively, was also explored in sufficient detail to support the authors' view. Based on their background studies, the authors designed Fluplatin and the final formulation FP NPs, which can ultimately target the endoplasmic reticulum for efficient mutp53 degradation as well as endoplasmic reticulum stress, ultimately overcoming cisplatin resistance due to p53 mutations. The authors examined the mechanism by which FP NPs specifically act on mutp53 both in vivo and in vitro. The authors ultimately chose multiple models for their in vivo examination of the antitumor effects of FP NPs, including subcutaneous, in situ, and recurrent tumor models, and also in wild, mutant, and resistant cellular models at the same time. Overall, the authors' choice of entry point is meaningful, the mechanism study in the background study is detailed, and the whole study is full of data and has a certain potential for clinical application. Therefore, I am happy to recommend its acceptance for publication after minor revisions according to the following suggestions.

1. A control group of F NPs should be added to Fig. 5r.
2. The cells used in Fig. 9-10 should be labeled as luc cells.

Reviewer #3 - Original (Remarks to the Author):

There are no further comments from this reviewer.

Reviewer #4 - New Reviewer cisplatin resistance lung cancer, also asked to comment on Reviewer #2 (Remarks to the Author):

The article by Bi et al. "Mutant-p53-targeting-therapeutic nanoparticles resolves chemoresistance and tumor recurrence in non-small cell lung cancer" contains interesting findings such as the development of Fluplatin nanoparticles that are coated with PEG-PE. These particles show excellent in vitro and in vivo efficacy against cell line based xenografts

Unfortunately, the authors are using extremely bold statements such as "More importantly, FP NPs can also improve the poor prognosis, showing minimum recurrence." and based on the p53 transfection experiments (Fig 1m-n and Suppl. Fig 7) "This finding indicates a breakthrough in understanding the cause of the loss of cisplatin effectiveness in the clinic." In my opinion, the weakest part of this study is the relation between cisplatin-induced p53 mutations that would cause resistance to cisplatin according to the authors.

Furthermore, the article is difficult to read. There are many grammar errors and often abbreviations are not well explained or not all. The abstract is not well written for example: "Therefore, FP NPs were generated, comprising self-assembled Fluplatin to break the vicious cycle involving cisplatin and p53." and "Additionally, FP NPs relieve the mutp53-induced inhibition of cisplatin treatment and thus exert a profound antitumor effect." Similarly, the discussion is very difficult to follow.

Major comments:

1. Did the authors correct the patient data for stage etc? If cisplatin causes mutations in p53 then one would expect that NSCLC patient tumors had changed from wild type p53 before treatment to mutant p53 at recurrence. Is none of the other genes prognostic for overall survival? Which p53 mutations are common in NSCLC? Does it matter which type of mutations (nonsense vs missense mutation) for survival?
2. Did the authors really use 10 up to 50 μ M cisplatin continuously for 30 days for A549 and H460? That's an extremely high concentration. The authors show that there are mutations in p53 (Suppl Fig.4), which bases/ amino acids are affected or is it randomly? Did the authors sequence other genes? Why didn't the authors use classical western blotting to show p53 induction after cisplatin treatment or high expression in case it was a stabilized mutant p53 (Fig 1C/H).
3. In Fig 1O the IC50 of p53 mutant and wild type p53 cell lines are shown for cisplatin and cisplatin plus Fluvastatin. Which cell lines are used (or transfected p53 overexpressing ones)? There is a reduction in IC50 of cisplatin plus Fluvastatin in some of the p53 mutant ones, but they are overall still more resistant to cisplatin than the p53 wild type cell lines?
4. Figure 1P is not correct in my opinion: Fluvastatin is not affecting the appearance of mutant p53 but it is only effective on mutant p53. Or do the authors have data showing that disappearance of mutant p53 results in expression of wt p53 from the none mutated other allele(s).
5. Figure 3h is unreadable. Why is the x-axis not changed, so that the differences are more easily seen? Why didn't the authors analyze p53 expression levels when using FP NPs (Figure 4F,G,H). It's interesting that mutant p53 is degraded in the presence of FP NPs will wild type p53 is upregulated. Strangely, the induction of wild type p53 in A549 by FP NPs does not result in more apoptosis or reduced cell survival compared to A549/CCP cells that are mutant for p53 (Fig 3A). Which mutation is found in A549/CCP (Supp fig 25) and what is the p53 expression and response to FP NPs? What is indicated in Suppl Fig. 25A and B? There are many mutations in p53 in A549/CCP, are they also causing amino acid changes?
6. The authors use ERS as a mechanism of sensitivity induced by FP NPs. ESR is in my opinion more often referred to as ISR, integrated stress response.
7. Why are F NP and FP NP effective in vivo in H1975 mutant p53 tumors but also against A549 xenografts (wild type p53). What happened with p53 in these A549 xenografts?
8. The combination of Fluvastatin and cisplatin doesn't show any efficacy in H1975, A549 and A549/CCP, why not. Based on the in vitro experiments one would expect to see some antitumor activity in A549. The authors use F+C NPs but what is formula of these NPs? Why are they much less active than F NPs and FP NPs? Why are the F and FP NPs tumor selective and not inducing toxicity in mice, can the authors explain/discuss these results?

9. How did the authors determine the molar concentrations of Fluplatin NPs and can they be compared to free Fluvastatin and cisplatin?

10. What are the Y-axis indicating for the caspase 3 and 9 activity assays?

Responses to the comments of the Reviewers

Reviewer #1 - Original (Remarks to the Author):

In the revised manuscript, the authors conducted research based on the possible problems in cisplatin treatment, and discussed the biological process of cisplatin and the expression level of the p53 target genes, which I think has certain research significance for attacking cisplatin resistance. In addition, a profound mechanistic study was conducted in the selection of fluvastatin, which elucidated the deficiencies in cisplatin treatment and the biological significance of the selection of fluvastatin. The cytotoxic mechanism of cisplatin in cells expressing wild-type p53 and cells expressing mutant p53, respectively, was also explored in sufficient detail to support the authors' view. Based on their background studies, the authors designed Fluplatin and the final formulation FP NPs, which can ultimately target the endoplasmic reticulum for efficient mutp53 degradation as well as endoplasmic reticulum stress, ultimately overcoming cisplatin resistance due to p53 mutations. The authors examined the mechanism by which FP NPs specifically act on mutp53 both in vivo and in vitro. The authors ultimately chose multiple models for their in vivo examination of the antitumor effects of FP NPs, including subcutaneous, in situ, and recurrent tumor models, and also in wild, mutant, and resistant cellular models at the same time. Overall, the authors' choice of entry point is meaningful, the mechanism study in the background study is detailed, and the whole study is full of data and has a certain potential for clinical application. Therefore, I am happy to recommend its acceptance for publication after minor revisions according to the following suggestions.

Response: Thank you very much for your recognition of our work. Please find our responses to the comments point-by-point below.

1. A control group of F NPs should be added to Fig. 5r.

Response: Thank you for your kind comment. According to the reviewer's comments, we have added the control group of F NPs into Fig. 5r and the related description into the revised manuscript.

Fig. 5r HMGR inhibition assay under different concentrations of fluvastatin, F+C NPs, Fluplatin, F NPs, and FP NPs ($n = 3$ independent samples; one-way ANOVA followed by Tukey's HSD post hoc test).

2. The cells used in Fig. 9-10 should be labeled as luc cells.

Response: Thank you for your kind comment. According to the reviewer's comment we have relabeled the cells as luc cells in Fig. 9-10.

Fig. 9 The tumor recurrence and metastasis inhibition of FP NPs in H1975-luc tumor-bearing mice. **a** Schematic illustration of the experimental design. **b** The tumor volumes of mice during treatments with each group ($n = 7$ mice per group; two-way ANOVA followed by Tukey's multiple comparisons post test). **c** The body weight of mice during treatments with each group ($n = 7$ mice per group; two-way ANOVA followed by Tukey's multiple

comparisons post test). **d** Tumor volume changes for each mouse in each group ($n = 7$ mice per group). **e** In vivo bioluminescence imaging of tumor-bearing mice receiving various treatments after surgery. Three representative mice in each treatment group are shown. Images of day 0 were taken on the day of surgery. **f** The final anatomical picture and H&E staining of the lungs. **g** IHC of p53, E-cadherin, MMP-2, MMP-9, Vimentin in the primary tumor tissues of cisplatin or FP NPs treatment, and recurrent and metastatic tumor tissues of cisplatin treatment. Scale bars, 20 μm . **h** Recurrence tumor weights of different groups on day 42 after surgery ($n = 7$ mice per group; one-way ANOVA followed by Tukey's HSD post hoc test). **i** Kaplan-Meier survival curve of mice treated with each group over 65 days ($n = 7$ mice per group; Log-rank Mantel-Cox test). **j** Final weights of the heart, liver, spleen, lungs, and kidneys ($n = 7$ mice per group; two-tailed unpaired t test). **k** Heatmap of TNF- α , VEGF, IL-10 and IL-6 expression profiles in serum ($n = 3$ mice per group). **l** Indicators of routine blood examination ($n = 3$ mice per group; two-tailed unpaired t test) of mice. Data are shown as the mean \pm s.d. Source data are provided as a Source Data file.

Fig. 10 The tumor recurrence and metastasis inhibition of FP NPs in A549-luc and A549/DDP-luc tumor-bearing mice. a, l Schematic illustration of the experimental design. **b, m** The tumor volumes of mice during treatments with each group ($n = 7$ mice per group; two-way ANOVA followed by Tukey's multiple comparisons post test). **c, n** The body weight of mice during treatments with each group ($n = 7$ mice per group; two-way ANOVA followed by Tukey's multiple comparisons post test). **d, o** Tumor volume changes for each mouse in each group ($n = 7$ mice per group). **e, p** In vivo bioluminescence imaging of tumor-bearing mice receiving various treatments after surgery. Three representative mice in each treatment group are shown. Images of day 0 were taken on the day of surgery. **f, q** The final anatomical picture and H&E staining of the lungs. **g, r** Final weights of the heart, liver, spleen, lungs, and kidneys ($n = 7$ mice per group; two-tailed unpaired t test). **h, s** The recurrence tumor weights of different groups on day 42 after surgery ($n = 7$ mice per group; one-way ANOVA followed by Tukey's HSD post hoc test). **i, t** Kaplan-Meier survival curve of mice treated with each group over 65 days ($n = 7$ mice per group; Log-rank Mantel-Cox test). **j, u** Heatmap of TNF- α , VEGF, IL-10 and IL-6 expression profiles in serum ($n = 3$ mice per group). **k, v** Indicators of routine blood examination ($n = 3$ mice per group; two-tailed unpaired t test) of mice. Data are shown as the mean \pm s.d. Source data are provided as a Source Data file.

Thank you very much!

Reviewer #3 - Original (Remarks to the Author):

There are no further comments from this reviewer.

Response: We appreciate the reviewers' recognition of our revised manuscript.

Thank you very much!

Reviewer #4 - New Reviewer cisplatin resistance lung cancer, also asked to comment on Reviewer #2 (Remarks to the Author):

The article by Bi et al. "Mutant-p53-targeting-therapeutic nanoparticles resolves chemoresistance and tumor recurrence in non-small cell lung cancer" contains interesting findings such as the development of Fluplatin nanoparticles that are coated with PEG-PE. These particles show excellent in vitro and in vivo efficacy against cell line based xenografts

Response: Thank you very much for your recognition of our work.

Unfortunately, the authors are using extremely bold statements such as "More importantly, FP NPs can also improve the poor prognosis, showing minimum recurrence." and based on the p53 transfection experiments (Fig 1m-n and Suppl. Fig 7) "This finding indicates a breakthrough in understanding the cause of the loss of cisplatin effectiveness in the clinic." In my opinion, the weakest part of this study is the relation between cisplatin-induced p53 mutations that would cause resistance to cisplatin according to the authors.

Response: Thank you for your valuable comments and advice. We recognize that the discussion of the relationship between cisplatin-induced p53 mutations and resistance to cisplatin is inadequate, and based on the reviewer's suggestions we have strengthened this part of the conclusion and added experiments.

The relationship between cisplatin-induced p53 mutations that cause resistance to cisplatin has been reported¹⁻⁴. P53 is very important in the antitumor mechanism of cisplatin¹, and the mutation of p53 is one of the important factors of cisplatin resistance²; p53 has a high mutation rate in NSCLC³, and cisplatin further enhances the risk of gene mutations⁴. Additionally, our results showed that they do have a correlation. We verified the biological mechanism and final mutation status of some cisplatin-induced p53 wild-type NSCLC cell lines, and further investigated the expression of p53 target genes and the changes in the p53 downstream pathway using RNA sequencing (RNA-seq). The results showed that with increasing induction dose and time, the mutating factors accumulated and showed a higher number of mutations, and the apoptosis pathway was further inactivated by activating p53 (Fig. 1c). Moreover, pathway enrichment analysis showed a "gain of function (GOF)" effect of cisplatin-induced mutant p53 (mutp53), which could further lead to cisplatin resistance (Fig. 1h-i). To further strengthen the relation between cisplatin-induced p53 mutations that

would cause resistance to cisplatin, according to the reviewer's suggestions, we have added the Methyl thiazolyl tetrazolium (MTT) and caspase activity experiments, and the results also directly demonstrate the correlation between cisplatin resistance and cisplatin-induced p53 mutation (Supplementary Fig. 8).

Supplementary Fig. 8 The cytotoxicity and Caspase enzymatic activity of A549 cells and H460 cells after cisplatin treatment. The cytotoxicity and IC_{50} of cisplatin treatment on A549 cells (**a, b**) and H460 cells (**e, f**) were measured by the MTT method, and then the enzymatic activity of Caspase-3 and Caspase-9 was detected (**c, d, g, h**) ($n = 3$ independent samples). Data were shown as the mean \pm s.d. Statistical analysis was performed using two-tailed unpaired t test for **c, d, g, and h**; and one-way ANOVA followed by Tukey's HSD post hoc test for **a, b, e, and f**.

References

1. Pisano, C. et al. Antitumor activity of the combination of synthetic retinoid ST1926 and cisplatin in ovarian carcinoma models. *Ann. Oncol.* **18**, 1500–1505 (2007).
2. Cao, X., Hou, J., An, Q., Assaraf, Y. G., & Wang, X. Towards the overcoming of anticancer drug resistance mediated by p53 mutations. *Drug Resist. Updat.* **49**, 100671 (2020).
3. Chaft, J. E. et al. Evolution of systemic therapy for stages I-III non-metastatic non-small-cell lung cancer. *Nat. Rev. Clin. Oncol.* **18**, 547–557 (2021).
4. Bagrodia, A. et al. Genetic determinants of cisplatin resistance in patients with advanced germ cell tumors. *J. Clin. Oncol.* **34**, 4000–4007 (2016).

Furthermore, the article is difficult to read. There are many grammar errors and often

abbreviations are not well explained or not all. The abstract is not well written for example: “Therefore, FP NPs were generated, comprising self-assembled Fluplatin to break the vicious cycle involving cisplatin and p53.” and “Additionally, FP NPs relieve the mutp53-induced inhibition of cisplatin treatment and thus exert a profound antitumor effect.” Similarly, the discussion is very difficult to follow.

Response: Thank you for your kind comments. We apologize for the grammar errors present in the article. We have thoroughly reviewed and proofread the entire paper to rectify these issues. Additionally, we have sought assistance from a professional proofreader to ensure the higher quality of writing. We have revised the paper to expand the initial usage of abbreviations. Finally, we also revised the Abstract, Introduction, and Discussion sections to make the presentation of information more coherent and to ensure a logical flow from one point to another.

Major comments:

1. Did the authors correct the patient data for stage etc? If cisplatin causes mutations in p53 then one would expect that NSCLC patient tumors had changed from wild type p53 before treatment to mutant p53 at recurrence. Is none of the other genes prognostic for overall survival? Which p53 mutations are common in NSCLC? Does it matter which type of mutations (nonsense vs missense mutation) for survival?

1.1 Did the authors correct the patient data for stage etc?

Response: Thank you for the suggestion about the stage of the patient data. We considered the stage of the patient when analyzing the data. In addition, we have used the statistical analyses to account for any potential confounders associated with patients survival¹ (Supplementary Table 2).

Variable	Univariate analysis			Multivariate analysis		
Factor	HR	95% CI	p -value	HR	95% CI	p -value
Sex						
Male	1			1		
Female	0.741	0.600-0.916	0.006	0.898	0.720-1.120	0.339
Stage						
I	1			1		
II	1.575	1.210-2.048	0.001	1.594	1.223-2.079	0.001
III	2.135	1.654-2.754	0.000	2.276	1.758-2.947	0.000
IV	3.744	2.487-5.636	0.000	4.067	2.692-6.144	0.000
Person cigarette pack year value						
≤28	1			1		
>28	1.333	1.085-1.637	0.006	1.183	0.949-1.476	0.135

Supplementary Table 2. Univariate and multivariate Cox regression analysis for overall survival in all patients. CI, confidence interval; HR, hazard ratio.

1.2 If cisplatin causes mutations in p53 then one would expect that NSCLC patient tumors had changed from wild type p53 before treatment to mutant p53 at recurrence.

Response: Thank you very much for your question about whether p53 can be mutated before treatment. Mutations in Tumor Protein p53 (*TP53*) can occur at different stages of NSCLC development, and the timing of the mutations varies from person to person². The development of *TP53* mutations in NSCLC is a complicated process influenced by a variety of factors, including exposure to oncogenes, genetic susceptibility, and other environmental factors². Initial *TP53* mutations usually occur early in the tumorigenic process, which is the early stage of tumor development². Therefore, most patients during this period have not yet been able to detect the disease and participate in treatment³. NSCLC patients have a high rate of p53 mutations³, and it is possible that patients may already have mutations prior to cisplatin

treatment, and treatment with cisplatin further increases the risk of p53 mutations.

1.3 Is none of the other genes prognostic for overall survival?

Response: We acknowledge the reviewer's concern about the potential impact of other genes on overall survival. In NSCLC, mutations in genes such as epidermal growth factor receptor (*EGFR*) and V-Ki-ras2 Kirsten rat sarcoma viral oncogene homologue (*KRAS*) are also associated with overall survival⁴.

1.4 Which p53 mutations are common in NSCLC?

Response: In NSCLC, p53 mutations are relatively common, and there is a wide range of mutations observed in *TP53*⁵, and the most common mutation types generally recognized are still dominated by missense mutations⁵. The specific p53 mutations can vary between patients, and common p53 mutations in NSCLC can include: missense mutations, some common missense mutations in NSCLC such as R175H, R273H, V175F, and others; nonsense mutations; deletion or insertion mutations.

1.5 Does it matter which type of mutations (nonsense vs missense mutation) for survival?

Response: The impact of specific p53 mutations on patient survival can vary. In general, missense mutations may still leave some p53 function, although this function may be altered or diminished⁶. The degree of loss of p53 function caused by a missense mutation affects its impact on survival⁵. In contrast, nonsense mutations typically result in loss of p53 protein function, which may have a more severe impact on survival⁵. The relationship between the type of p53 mutation and its impact on patient survival is complex and may depend on a number of factors, including the patient's overall genetic and clinical characteristics, and the stage of the cancer⁷.

References

1. Hassin O. et al. Different hotspot p53 mutants exert distinct phenotypes and predict outcome of colorectal cancer patients. *Nat Commun.* **13**, 2800 (2022).
2. Chen X. et al. Mutant p53 in cancer: from molecular mechanism to therapeutic modulation. *Cell Death Dis.* **13**, 974 (2022).
3. Chaft, J. E. et al. Evolution of systemic therapy for stages I-III non-metastatic non-small-cell lung cancer. *Nat. Rev. Clin. Oncol.* **18**, 547–557 (2021).
4. Saleh MM. et al. Comprehensive Analysis of TP53 and KEAP1 Mutations and Their Impact on Survival in Localized- and Advanced-Stage NSCLC. *J Thorac Oncol.* **17**, 76–

88 (2022).

5. Provencio M. et al. Overall Survival and Biomarker Analysis of Neoadjuvant Nivolumab Plus Chemotherapy in Operable Stage IIIA Non-Small-Cell Lung Cancer (NADIM phase II trial). *J Clin Oncol.* **40**, 2924–2933 (2022).
6. abapathy K. et al. Therapeutic targeting of p53: all mutants are equal, but some mutants are more equal than others. *Nat Rev Clin Oncol.* **15**, 13–30 (2018).
7. Steels E. et al. Role of p53 as a prognostic factor for survival in lung cancer: a systematic review of the literature with a meta-analysis. *Eur Respir J.* **18**, 705–719 (2001).

2. Did the authors really use 10 up to 50 μM cisplatin continuously for 30 days for A549 and H460? That's an extremely high concentration. The authors show that there are mutations in p53 (Suppl Fig.4), which bases/amino acids are affected or is it randomly? Did the authors sequence other genes? Why didn't the authors use classical western blotting to show p53 induction after cisplatin treatment or high expression in case it was a stabilized mutant p53 (Fig 1C/H).

2.1 Did the authors really use 10 up to 50 μM cisplatin continuously for 30 days for A549 and H460? That's an extremely high concentration.

Response: We did use 10-50 μM cisplatin to treat A549 cells and H460 cells for one month. Our experimental protocol refers to the cisplatin-resistant cell line constructs, which usually use fixed concentration induction or gradually increase the concentration, and the generally chosen dose is usually the IC_{50} value or IC_{30} value¹⁻². Under serum-free conditions, the IC_{50} and IC_{30} values were $15.62 \pm 1.09 \mu\text{M}$ and $24.81 \pm 1.34 \mu\text{M}$ in A549 cells and $12.94 \pm 0.63 \mu\text{M}$ and $21.08 \pm 1.88 \mu\text{M}$ in H460 cells, respectively; under serum-containing conditions, the IC_{50} and IC_{30} values were $38.92 \pm 1.45 \mu\text{M}$ and $72.73 \pm 2.03 \mu\text{M}$ in A549 cells and $33.34 \pm 1.46 \mu\text{M}$ and $67.08 \pm 2.69 \mu\text{M}$ in H460 cells, respectively (Supplementary Fig. 6). Although there was a loss of cells in the high-dose group during the induction process, the number of cells in the high-dose group was still sufficient for all aspects of the study because we had prepared a sufficient number of cells for induction, and the mortality rate of the cells was reduced due to the accumulation of drug resistance during the induction process. We will elaborate on the rationale for this choice in the revised manuscript to provide a clearer context for our experimental design.

	IC ₅₀ (μM)	IC ₃₀ (μM)
A549(-)	15.62 ± 1.09	24.81 ± 1.34
A549(+)	38.92 ± 1.45	72.73 ± 2.03
H460(-)	12.94 ± 0.63	21.08 ± 1.88
H460(+)	33.34 ± 1.46	67.08 ± 2.69

Supplementary Fig. 6 Cytotoxicity of cisplatin in A549 cells and H460 cells with and without serum. With serum (-), without serum (+). ($n = 3$ independent samples; one-way ANOVA followed by Tukey's HSD post hoc test).

2.2 The authors show that there are mutations in p53 (Suppl Fig.4), which bases/amino acids are affected or is it randomly?

Response: The mutations in p53 are not random, and they correspond to specific amino acid changes. We have sequenced exons 5-9 in Supplementary Fig. 4c; they have been identified and we have added specific mutation information to the source data file in the revised manuscript.

2.3 Did the authors sequence other genes?

Response: First, in NSCLC, p53 mutations are the most common type of mutation and are closely associated with conditions such as patient prognostic status and chemotherapy resistance³. Second, our study aimed to address the issue of cisplatin in the treatment of NSCLC compared with *TP53*. Other genes with higher mutation rates in NSCLC, such as EGFR, and KRAS, have less impact on cisplatin and are more related to immunotherapy resistance⁴⁻⁵. Therefore, we did not test for other gene mutations. Of course, the examination of mutations in other genes also has some clinical significance; however, we would prefer to further strengthen the significance of our established study of targeting p53 mutations for the treatment of NSCLC by further validating the link between cisplatin and p53 mutations.

2.4 Why didn't the authors use classical western blotting to show p53 induction after cisplatin treatment or high expression in case it was a stabilized mutant p53 (Fig 1C/H).

Response: We appreciate this suggestion. In the revised manuscript, we have added Western

blot data to demonstrate p53 induction after cisplatin treatment and high expression in the case of stabilized mutp53 (Supplementary Fig. 7). These additional data provide a more comprehensive view of p53 regulation and expression in response to cisplatin treatment and its implications for our study. Thank you.

Supplementary Fig. 7 Western blotting analysis of p53 in cisplatin-induced cells after treatment with 4 μM fluvastatin for 12 h. **a** Western blotting analysis, and their fluorescence intensity were quantified (**b-c**). ($n = 3$ independent samples). Data are shown as the mean \pm s.d. Statistical analysis was performed using two-tailed unpaired t test for **b, c**.

References

1. Chung YM. et al. Establishment and characterization of 5-fluorouracil-resistant gastric cancer cells. *Cancer Lett.* **159**, 95–101 (2000).
2. Benavente S. et al. Establishment and characterization of a model of acquired resistance to epidermal growth factor receptor targeting agents in human cancer cells. *Clin Cancer Res.* **15**, 1585–1592 (2009).
3. Chaft, J. E. et al. Evolution of systemic therapy for stages I-III non-metastatic non-small-cell lung cancer. *Nat. Rev. Clin. Oncol.* **18**, 547–557 (2021).
4. Azuma K. et al. Association of PD-L1 overexpression with activating EGFR mutations in surgically resected nonsmall-cell lung cancer. *Ann Oncol.* **25**, 1935–1940 (2014).
5. Dong ZY. et al. Potential Predictive Value of *TP53* and *KRAS* Mutation Status for Response to PD-1 Blockade Immunotherapy in Lung Adenocarcinoma. *Clin Cancer Res.* **23**, 3012–3024 (2017).

3. In Fig 10 the IC50 of p53 mutant and wild type p53 cell lines are shown for cisplatin and cisplatin plus Fluvastatin. Which cell lines are used (or transfected p53 overexpressing ones)? There is a reduction in IC50 of cisplatin plus Fluvastatin in some of the p53 mutant ones, but they are overall still more resistant to cisplatin than the p53 wild type cell lines?

3.1 In Fig 10 the IC50 of p53 mutant and wild type p53 cell lines are shown for cisplatin

and cisplatin plus Fluvastatin. Which cell lines are used (or transfected p53 overexpressing ones)?

Response: We appreciate the reviewer's interest in the data presented in Fig. 1o, which compares the IC₅₀ values of the p53 mutant and wild type p53 cell lines for cisplatin and cisplatin plus fluvastatin. We selected a series of lung cancer cells expressing mutp53/wild-type p53 (wtp53). These cell lines are not transfected with exogenous p53 and naturally carried mutp53/wtp53. The specific cell lines are labeled in detail in the Source Data file of Fig. 1o.

3.2 There is a reduction in IC₅₀ of cisplatin plus Fluvastatin in some of the p53 mutant ones, but they are overall still more resistant to cisplatin than the p53 wild type cell lines?

Response: As observed in the data, there was a reduction in the IC₅₀ when fluvastatin was combined with cisplatin for some of the p53 mutant cell lines. However, it is important to note that p53 mutant cell lines still exhibit greater resistance to cisplatin than p53 wild type cell lines¹, as shown in Fig. 1o. These results underscore the complex interplay between p53 status and cisplatin sensitivity. While fluvastatin may enhance cisplatin sensitivity in certain p53 mutant cell lines, p53 mutations often confer inherent resistance to cisplatin². Moreover, as shown by our experimental results, although fluvastatin can specifically degrade mutp53, it is not completely eliminated but reduces the level of mutp53 to a certain extent (Supplementary Fig. 7, 9). The combination of fluvastatin can alleviate cisplatin resistance in mutant cell lines, but it is still limited compared to wild-type cell lines. Therefore, we designed a therapeutic strategy from targeting p53 mutations to overcoming them. This complexity is an important aspect of our study, and we have provided a detailed discussion of these findings in the revised manuscript to address the reviewer's concerns and to enhance the understanding of the role of p53 mutations in cisplatin resistance.

References

1. Skowron MA. et al. The developmental origin of cancers defines basic principles of cisplatin resistance. *Cancer Lett.* **519**, 199–210 (2021).
2. Cao X, Hou J. et al. Towards the overcoming of anticancer drug resistance mediated by p53 mutations. *Drug Resist Updat.* **49**, 100671 (2020).
4. Figure 1P is not correct in my opinion: Fluvastatin is not affecting the appearance of mutant

p53 but it is only effective on mutant p53. Or do the authors have data showing that disappearance of mutant p53 results in expression of wt p53 from the none mutated other allele(s).

4.1 Figure 1P is not correct in my opinion: Fluvastatin is not affecting the appearance of mutant p53 but it is only effective on mutant p53.

Response: Thank you for your kind comments. We appreciate the reviewer's attention to the data presented in Fig. 1p and the interpretation of the results. We have made corrections to Fig. 1p based on the reviewer's comments.

Fig. 1p Schematic illustration of treatment with cisplatin in mutp53 or wtp53 cells.

4.2 Or do the authors have data showing that disappearance of mutant p53 results in expression of wt p53 from the none mutated other allele(s).

Response: We do not have data showing that the disappearance of mutp53 results in the expression of wt p53 from the nonmutated allele(s). The effects of fluvastatin on p53 in our study primarily relate to the regulation and potential modulation of mutp53, and we have made this interpretation clearer in the revised manuscript. We thank the reviewer for highlighting this aspect, and we ensure the accurate representation of our findings in the revised figure and manuscript.

5. Figure 3h is unreadable. Why is the x-axis not changed, so that the differences are more easily seen? Why didn't the authors analyze p53 expression levels when using FP NPs (Figure 4F,G,H). It's interesting that mutant p53 is degraded in the presence of FP NPs will wild type p53 is upregulated. Strangely, the induction of wild type p53 in A549 by FP NPs does not result in more apoptosis or reduced cell survival compared to A549/CCP cells that are mutant for p53 (Fig 3A). Which mutation is found in A549/CCP (Supp fig 25) and what is the p53 expression and response to FP NPs? What is indicated in Suppl Fig. 25A and B? There are

many mutations in p53 in A549/CCP, are they also causing amino acid changes?

5.1 Figure 3h is unreadable. Why is the x-axis not changed, so that the differences are more easily seen?

Response: We appreciate the feedback regarding the readability of Fig. 3h. In the original version, we kept the scales of the X and Y axes uniform to be able to observe the degree of change in the IC₅₀ values through the slope. We make the necessary adjustments in the revised charts to improve the clarity and visualization of the data. Specifically, we have added an additional image to the original figure to enhance the presentation of differences between groups and make it easier to interpret.

Fig. 3h Dot plots showing the IC₅₀(Cis)/IC₅₀(FP NPs) in mutant and wild-type cells.

5.2 Why didn't the authors analyze p53 expression levels when using FP NPs (Figure 4F,G,H).

Response: We have examined and discussed the regulation of p53 by FP NPs in cells expressing mutp53/wtp53 in Fig. 5a-h. The different effects of FP NPs on cells expressing mutp53/wtp53 are because fluvastatin specifically degrades mutp53 and has no effect on wtp53. Therefore, in mutp53-expressing cells, FP NPs degraded mutp53 more efficiently due to higher cellular uptake and endoplasmic reticulum (ER) targeting capacity, whereas in wtp53-expressing cells, cisplatin released from FP NPs still exerted the same upregulation of the proapoptotic mechanism of p53. According to the reviewer's comment, we analyzed the p53 expression levels of mutp53/wtp53-expressing cells in more detail when exposed to FP NPs in the revised manuscript.

Fig. 5 Mechanism of p53-related antitumor activity of FP NPs in vitro. **a** Western blotting analysis of p53 in H1975, H2087, H2342 and A549 cells after treatment with 4 μ M of different formulations (dosage based on Fluplatin) for 12 h. Their grayscale values were quantified (**b-e**) ($n = 3$ independent samples; one-way ANOVA followed by Tukey's HSD

post hoc test). **f** Western blotting analysis of p53 in H1975 and A549 cells after treatment with different concentrations of FP NPs for 12 h. Their grayscale values were quantified (**g-h**) ($n = 3$ independent samples; one-way ANOVA followed by Tukey's HSD post hoc test). **i** Western blotting analysis of p53 in H1793, H2196, H748, H1770, H1581, and H520 cells after treatment with 4 μ M of FP NPs for 12 h. And their grayscale values were quantified (**j**) ($n = 3$ independent samples; one-way ANOVA followed by Tukey's HSD post hoc test). **k** Western blot of H1975, H2087, H2342, and A549 cells treated with 4 μ M FP NPs or not for 12 h and cultured in medium containing 10 μ M MG132 or 10 mM 3-MA. Their grayscale values were quantified (**l**) ($n = 3$ independent samples; one-way ANOVA followed by Tukey's HSD post hoc test). Western blot of H1975 (**m**), H2087 (**n**), H2342 (**o**), and A549 (**p**) cells treated with 2 μ M FP NPs or not for 12 h and then treated with 100 μ g/mL CHX at the indicated time points. $n = 3$ biologically independent samples, relative p53/actin ratios are shown. **q** Western blotting analysis of total ubiquitination (Ub) of immunoprecipitated (IP) p53 in H1975 and A549 cells after treatment with 2 μ M FP NPs with or without MG132. **r** HMGR inhibition assay under different concentrations of fluvastatin, F+C NPs, Fluplatin, F NPs, and FP NPs ($n = 3$ independent samples; one-way ANOVA followed by Tukey's HSD post hoc test). **s** Genome-wide analysis of differential expression of FP NPs treated and untreated H1975 cells. **t** Schematic summary of the p53-related antitumor mechanism of FP NPs. Source data are provided as a Source Data file.

5.3 It's interesting that mutant p53 is degraded in the presence of FP NPs will wild type p53 is upregulated. Strangely, the induction of wild type p53 in A549 by FP NPs does not result in more apoptosis or reduced cell survival compared to A549/CCP cells that are mutant for p53 (Fig 3A).

Response: Thank you for these thoughtful comments. Not only between A549/DDP cells and A549 cells, as shown in Fig. 3h, but also between the cell lines that expressing mutp53/wtp53, the difference in IC_{50} values was not significant due to the unique mechanism of action of FP NPs. This result was attributed to the fact that FP NPs specifically degraded mutp53 in mutp53-expressing cells, and still upregulated p53 in wtp53-expressing cells. On the other hand, due to the structure of fluvastatin in the prodrug and the encapsulation of poly-(ethylene glycol)-phosphoethanolamine (PEG-PE) in the FP NPs, the ER targeting ability of FP NPs can ultimately induce ER stress (ERS) in both p53 wild/mutant NSCLC cells, which results in the potent proapoptotic ability of FP NPs independent of the p53 pathway. In addition, our results in Fig. 3e-g show that the IC_{50} values after inhibition of p53 or c-Abl also did not change much in cells expressing p53 wild/null/mutation, which also suggests that ERS exerts a major proapoptotic effect. In the revised manuscript, we have discussed these interesting findings in more detail, exploring potential explanations and implications for the study.

5.4 Which mutation is found in A549/CCP (Supp fig 25) and what is the p53 expression

and response to FP NPs? What is indicated in Suppl Fig. 25A and B? There are many mutations in p53 in A549/CCP, are they also causing amino acid changes?

Response: We appreciate your interest in the A549/DDP cell line and the specific p53 mutations it carries. In the sequencing results of exons 5-9 of A549/DDP cells, point mutations were detected, as described in original Supplementary Fig. 25b (Supplementary Fig. 28b in revised manuscript) and in the Source Data file. According to the reviewer's comment about p53 expression and response to FP NPs, we supplemented the experiments. The results showed that the p53 expression of A549/DDP cells was higher than that of A549 cells expressing wtp53, which was also due to the p53 mutation in A549/DDP cells; FP NPs could upregulate p53 in A549 cells expressing wtp53 and degrade p53 in A549/DDP cells expressing mutp53 (Supplementary Fig. 29). Fragments 1, 2, and 3 labeled in original Supplementary Fig. 25a are fragments from our *TP53* sequencing of A549/DDP cells and are exons 5-9, which are more mutated in *TP53*. Original Supplementary Fig. 25b shows the mutations we detected in exons 5-9 in A549/DDP cells, and we have labeled the specific mutation sites in detail in the Source Data file, and they correspond to specific amino acid changes.

Supplementary Fig. 29 Western blotting analysis of p53 in A549 cells and A549/DDP cells after treatment with 4 μ M FP NPs for 12 h. a Western blotting analysis, and their fluorescence intensity was quantified (**b**). ($n = 3$ independent samples). Data were shown as the mean \pm s.d. Statistical analysis was performed using two-tailed unpaired t test for **b**.

6. The authors use ERS as a mechanism of sensitivity induced by FP NPs. ESR is in my opinion more often referred to as ISR, integrated stress response.

Response: We appreciate the reviewer's comment. As the reviewer suggested, ESR is more often referred to as ISR, integrated stress response. In this work, we intended to amplify the ER damage by cisplatin through the ER-targeting ability of FP NPs, which further triggers a

stronger ERS, thus making tumor cells apoptotic independently of the p53 pathway. In the examination of the antitumor mechanism of action of FP NPs, we also verified that it exerts its proapoptotic effect through the Protein kinase R-like endoplasmic reticulum kinase (PERK) pathway among the three ERS pathways (Fig. 4f-h). Whereas ISR is a broader definition, ERS is also one of the stimuli to which ISR responds. ERS is more consistent with the mechanism of action of FP NPs, and we ultimately chose ERS to describe the pro-apoptotic biological process that responds to stress and further damages mitochondria through activation of the PERK pathway and release of Ca^{2+} (Fig. 4a).

7. Why are F NP and FP NP effective in vivo in H1975 mutant p53 tumors but also against A549 xenografts (wild type p53). What happened with p53 in these A549 xenografts?

Response: In both in vitro and in vivo experiments, F NPs and FP NPs were highly effective against tumors in both p53 mutant/wild-type cells, which was attributed to their mechanism of action of targeting mutp53 and overcoming p53 mutations (Fig. 4-5). In short, we synthesized the prodrug Fluplatin, which can exert the mechanism of fluvastatin and cisplatin, and enhances certain ER targeting abilities due to the formation of the prodrug, which also enhances the ERS of cisplatin (Fig. 2, 4). F NPs are nanoparticles formed by self-assembled Fluplatin, which have stronger intracellular uptake than Fluplatin (Fig. 2j). The final preparation FP NPs further enhanced cellular uptake and ER targeting through PEG-PE encapsulation (Fig. 2m). Therefore, Fluplatin, F NPs, and FP NPs all exerted specific degradation of mutp53 by fluvastatin in p53 mutant cells, and the degradation effect was enhanced by the uptake ability (Fig. 5a-e). In addition, with the gradual enhancement of ER targeting ability, the ERS that can be triggered by cisplatin is also continuously enhanced, which can eventually exert highly efficient p53-independent apoptosis (Fig. 4). At the same time, cisplatin in the prodrug can also exert other p53-independent proapoptotic pathways (Fig. 4). In p53 wild-type cells, wtp53 is not subjected to degradation, and the ability of cisplatin to upregulate wtp53 can still be exerted in the presence of Fluplatin, F NPs, and FP NPs (Fig. 5a-e). In addition, the targeting ability of ER is not affected by whether it is mutated or not, and it can still exert efficient ERS in p53 wild-type cells (Fig. 4 and Supplementary Fig. 30-37). Furthermore, we can conclude from the results in Fig. 3e-g that the IC50 values of FP NPs after inhibition of the p53 or c-Abl pathway did not change much in H1299 cells

expressing wild-type/null/mutant p53, proving that ERS exerted a major proapoptotic effect. Therefore, we believe that FP NPs can maximize ERS triggering and thus overcome the limitation of p53 mutation on cisplatin treatment, regardless of whether the mutation occurs or not or a new mutation occurs during the treatment process, and ultimately can be effective in the treatment of NSCLC.

8. The combination of Fluvastatin and cisplatin doesn't show any efficacy in H1975, A549 and A549/CCP, why not. Based on the in vitro experiments one would expect to see some antitumor activity in A549. The authors use F+C NPs but what is formula of these NPs? Why are they much less active than F NPs and FP NPs? Why are the F and FP NPs tumor selective and not inducing toxicity in mice, can the authors explain/discuss these results?

8.1 The combination of Fluvastatin and cisplatin doesn't show any efficacy in H1975, A549 and A549/CCP, why not. Based on the in vitro experiments one would expect to see some antitumor activity in A549.

Response: In the in vivo experiments of H1975, A549, and A549/DDP cells, the combination of fluvastatin and cisplatin had a significant difference relative to the PBS group and the cisplatin group (Fig. 6d, Fig. 8e, k, Supplementary Fig. 46d, k and Supplementary Fig 47e), which proved that the combination of fluvastatin could restore cisplatin resistance but was still limited.

In the subcutaneous and orthotopic NSCLC models of A549 (Fig. 8 and Supplementary Fig. 47), cisplatin resistance was relatively low because A549 is a p53 wild-type cell, and cisplatin treatment alone showed a significant difference, while the combination of fluvastatin further enhanced the antitumor effect. In the revised manuscript, we have put the data into more detailed intergroup comparisons as well as corresponding explorations.

Fig. 6 Antitumor efficacy of intravenously injected FP NPs in an H1975 tumor-bearing mouse model. **a** Schematic illustration of the experimental design. **b** Images of tumor and tumor weight/volume on day 19 of each group ($n = 5$ mice per group; one-way ANOVA followed by Tukey's HSD post hoc test). **c** Tumor volume changes for each mouse in each group over 21 days of treatment ($n = 5$ mice per group). **d** The tumor volumes of mice during treatments with each group ($n = 5$ mice per group; two-way ANOVA followed by Tukey's multiple comparisons post test). **e** The body weight of mice during treatments with each group ($n = 5$ mice per group; two-way ANOVA followed by Tukey's multiple comparisons post test). Confocal images of ROS (**f**) and TUNEL (**g**) in the tumor tissues of mice after treatment with each group. Scale bars, 100 μm . IHC of Cleaved Caspase-3 (**h**) and H&E staining (**i**) in

the tumor tissues after treatment with each group. Scale bars of c-caspase3, 20 μm ; Scale bars of H&E, 100 μm . Data are shown as the mean \pm s.d Source data are provided as a Source Data file.

Supplementary Fig. 46 Antitumor efficacy of intravenously injected FP NPs in A549 and A549/DDP tumor-bearing mouse models. a, h Schematic illustration of the experimental design. **b, i** Images of tumor and tumor weight/volume on day 19 of each group ($n = 5$ mice per group; one-way ANOVA followed by Tukey's HSD post hoc test). **c, j** Tumor volume changes for each mouse in each group over 21 days of treatment ($n = 5$ mice per group). **d, k** The tumor volumes of mice during treatments with different formulations ($n = 5$ mice per

group; two-way ANOVA followed by Tukey's multiple comparisons post test). **e, l** The body weight of mice during treatments with different formulations ($n = 5$ mice per group; two-way ANOVA followed by Tukey's multiple comparisons post test). Data are shown as the mean \pm s.d Source data are provided as a Source Data file. **f, m** Kaplan-Meier survival curve of mice treated with different formulations over 70 days ($n = 8$ mice per group; Log-rank Mantel-Cox test). **g, n** The body weight of mice during treatments with different formulations over 70 days ($n = 8$ mice per group; two-way ANOVA followed by Tukey's multiple comparisons post test).

Fig. 8 Antitumor efficacy of intravenously injected FP NPs in H1975-luc and A549-luc orthotopic lung tumors. **a, g** Schematic illustration of the experimental design. **b, h** The

progression of orthotopic lung tumors ($n = 5$ mice per group) in BALB/c nude mice. A representative bioluminescent image from each group is shown. **c, i** The final anatomical picture and H&E staining of the lungs. **d, j** Average tumor nodule. Tumor nodules of 2-10 mm³ in volume were counted using harvested lungs from the control and treated groups, and the average number of tumor nodules was determined. Each dot represents a tumor from an individual mouse. The tumor nodules in the lungs are indicated by arrows (one-way ANOVA followed by Tukey's HSD post hoc test). **e, k** Quantitative bioluminescence analysis in mice bearing orthotopic lung tumors; P/S = photons/second (two-way ANOVA followed by Tukey's multiple comparisons post test). **f, l** Mouse body weight in the orthotopic lung tumor mouse model (two-way ANOVA followed by Tukey's multiple comparisons post test). Data are shown as the mean \pm s.d. Source data are provided as a Source Data file.

Supplementary Fig. 47 Antitumor efficacy of intravenously injected FP NPs in A549/DDP-luc orthotopic lung tumors. a Schematic illustration of the experimental design. **b** The progression of orthotopic lung tumors ($n = 5$ mice per group) in BALB/c nude mice. A representative bioluminescent image from each group is shown. **c** The final anatomical picture and the H&E staining of the lungs. **d** Average tumor nodule. Tumor nodules of 2-10 mm³ in volume were counted using harvested lungs from the control and treated groups, and the average number of tumor nodules was determined. Each dot represents a tumor from an individual mouse. The tumor nodules in the lungs are indicated by arrows (one-way ANOVA followed by Tukey's HSD post hoc test). **e** Quantitative bioluminescence analysis in mice bearing orthotopic lung tumors; P/S = photons/second (two-way ANOVA followed by Tukey's multiple comparisons posttest). **f** Mouse body weight in the orthotopic lung tumor

mouse model (two-way ANOVA followed by Tukey's multiple comparisons posttest). Data are shown as the mean \pm s.d. Source data are provided as a Source Data file.

8.2 The authors use F+C NPs but what is formula of these NPs? Why are they much less active than F NPs and FP NPs?

Response: F+C NPs are nanoparticles of free cisplatin and fluvastatin encapsulated by PEG-PE. Because what was encapsulated in the F+C NPs was not a prodrug, the eventual membrane insertion by the PEG-PE micelles only increased the cellular uptake capacity, with limited enhancement of ER targeting, and did not induce ERS in the FP NPs that made the tumor cells apoptotic in large numbers.

Nanoparticles	Formulation
F+C NPs	cisplatin, fluvastatin, PEG-PE
F NPs	Fluplatin
FP NPs	Fluplatin, PEG-PE

8.3 Why are the F and FP NPs tumor selective and not inducing toxicity in mice, can the authors explain/discuss these results?

Response: F+C NPs, F NPs, and FP NPs all have lower toxicity compared to the cisplatin-containing free drug because the nanomedicine is more enriched in the tumor tissue. The capillary plexus in the tumor region is sparser and the lymphatic system is dysregulated in the region. In contrast, in normal tissues, the tight junctions between vascular epithelial cells make it more difficult for the drug to pass through the blood vessels. Due to the size effect of the nanoparticles, there is ultimately more aggregation at the tumor site¹. We have added more discussion of the potential reasons for these results in the revised manuscript.

References

1. Yu W. et al. Size-Tunable Strategies for a Tumor Targeted Drug Delivery System. *ACS Cent Sci.* **6**, 100–116 (2020).

9. How did the authors determine the molar concentrations of Fluplatin NPs and can they be compared to free Fluvastatin and cisplatin?

Response: While we prepared the final preparation of FP NPs, we also calculated its actual encapsulation rate (Fig. 2c), and ultimately our doses in both in vivo and in vitro experiments were calculated based on the actual concentration of Fluplatin in FP NPs. The molar

concentration of Fluplatin is directly convertible to the corresponding free fluvastatin and cisplatin (1 mol Fluplatin=1 mol cisplatin + 2 mol fluvastatin). We have enhanced the description of this section in the revised manuscript.

10. What are the Y-axis indicating for the caspase 3 and 9 activity assays?

Response: In these assays, the Y-axis represents the units of caspase 3/9 enzyme activity per unit weight of protein in a sample. In the revised manuscript, we have ensured that the Y-axis labels and units are clearly defined to improve the understanding of the results.

Thank you very much!

REVIEWERS' COMMENTS

Reviewer #4 (Remarks to the Author):

The authors have sufficiently answered the questions raised by the reviewers.

Responses to the comments of the Reviewers

Reviewer #4 (Remarks to the Author):

The authors have sufficiently answered the questions raised by the reviewers.

Response: We appreciate the reviewer's recognition of our revised manuscript.

Thank you very much!